
**An evaluation of new particle formation events in Helsinki during a Baltic Sea cyanobacterial**
**summer bloom**
Roseline C. Thakur[1], Lubna Dada[1,2,3], Lisa J. Beck[1], Lauriane L.J. Quéléver[1], Tommy Chan[1], Marjan
Marbouti[1], Xu-Cheng He[1], Carlton Xavier[1], Juha Sulo[1], Janne Lampilahti[1], Markus Lampimäki[1], Yee
Jun Tham[1,11], Nina Sarnela[1], Katrianne Lehtipalo[1,4], Alf Norkko[8,9], Markku Kulmala[1,5,6,7], Mikko
Sipilä[1], Tuija Jokinen[1,10]
[1]Institute for Atmospheric and Earth System Research/Physics, Faculty of Science, 00014 University
of Helsinki, Helsinki, Finland.
[2]School of Architecture, Civil and Environmental Engineering, École Polytechnique Fédérale de
Lausanne, Lausanne, Switzerland
[3]Laboratory of Atmospheric Chemistry, Paul Scherrer Institute, 5232 Villigen PSI, Switzerland
[4]Finnish Meteorological Institute, Helsinki, Finland.
[5]Aerosol and Haze Laboratory, Beijing Advanced Innovation Center for Soft Matter Science and
Engineering, Beijing University of Chemical Technology, 100089 Beijing, China.
[6]Joint International Research Laboratory of Atmospheric and Earth System Sciences, Nanjing
University, 210023 Nanjing, China.
[7]Lomonosov Moscow State University, Faculty of Geography, 119991, Moscow, GSP-1, 1
Leninskiye Gory.
[8]Tvärminne Zoological Station, University of Helsinki, J.A. Palméns väg 260, FI-10900 Hangö,
Finland
[9]Baltic Sea Centre, Stockholm University, Stockholm, Sweden
[10]Climate & Atmosphere Research Centre (CARE-C), The Cyprus Institute, P.O. Box 27456, Nicosia,
CY-1645, Cyprus.
[11]School of Marine Sciences, Sun Yat-Sen University, Zhuhai 519082, China
Correspondence to: roseline.thakur@helsinki.fi
**Abstract**
Several studies have investigated New Particle Formation (NPF) events from various sites ranging
from pristine locations, including (boreal) forest sites to urban areas. However, there is still a dearth
of studies investigating NPF processes and subsequent aerosol growth in coastal yet semi-urban sites,
where the tropospheric layer is a concoction of biogenic and anthropogenic gases and particles. The
investigation of factors leading to NPF becomes extremely complex due to the highly dynamic
meteorological conditions at the coastline especially when combined with both continental and
oceanic weather conditions. Herein, we engage a comprehensive study of particle number size
distributions and aerosol-forming precursor vapors at the coastal semi-urban site in Helsinki, Finland.
The measurement period, 25 June 2019–18 August 2019, was timed with the recurring cyanobacterial
summer bloom in the Baltic Sea region and coastal regions of Finland. Our study recorded several
regional/local NPF and aerosol burst events during this period. Although the overall anthropogenic
influence on Sulfuric Acid (SA) concentrations was low during the measurement period, we observed
that the regional or local NPF events, characterized by SA concentrations in the order of $10^7$



molecules per cm$^{-3}$ occurred mostly when the air mass travelled over the land areas. Interestingly,
when the air mass travelled over the Baltic Sea, an area enriched with Algae and cyanobacterial
blooms, high Iodic Acid (IA) concentration coincided with an aerosol burst or a spike event at the
measurement site. Further, SA-rich bursts were seen when the air mass travelled over the Gulf of
Bothnia, enriched with cyanobacterial blooms. The two most important factors affecting aerosol
precursor vapor concentrations, and thus the aerosol formation, were (1) the type of phytoplankton
species and intensity of bloom present in the coastal regions of Finland/ Baltic Sea and (2) the wind
direction. During the events, most of the growth of sub-3 nm particles was probably due to SA, rather
than IA or MSA, however much of the particle growth remained unexplained indicative of the strong
role of organics in the growth of particles, especially in the 3–7 nm particle size range. Further studies
are needed to explore the role of organics in NPF events and the potential influence of cyanobacterial
blooms in coastal locations.

Keywords: coastal environment, particle growth, methane sulfonic acid, cyanobacterial summer
bloom, sulfuric acid, iodic acid

**1 Introduction**
New particle formation (NPF) and growth of aerosols are regional processes occurring globally
introducing a substantial aerosol load into the atmosphere. NPF has been observed in different
environments, including pristine (Asmi et al., 2016; Jang et al., 2019; Jokinen et al., 2018), polluted
boundary layers and urban areas (Kulmala et al., 2021; Kulmala et al., 2017; Manninen et al., 2010;
Kulmala et al., 2016; Wang et al., 2017; Cai and Jiang, 2017; Deng et al., 2020; Yao et al., 2018; Du
et al., 2021; Yan et al., 2021), boreal forests (Buenrostro Mazon et al., 2016; Dada et al., 2017;
Kulmala et al., 2013; Kyrö et al., 2014; Leino et al., 2016; Nieminen et al., 2014; Rose et al., 2018),
tropical forests (Artaxo et al., 2013; Wimmer et al., 2018) and mountain tops (Bianchi et al., 2016,
2020). Few studies have investigated NPF processes in a coastal environment although the coastal
NPF research started quite early. The investigation of coastal aerosol events dates back to 1978, when
the measurements of total aerosol  number concentration  were carried out at the Tasmanian coast
(Bigg and Turvey, 1978). After that atmospheric nucleation was observed in the Southern hemisphere
around the Antarctic coastline (O'Dowd et al., 1997), in Mace Head (Flanagan et al., 2005; McFiggans
et al., 2004; O'Dowd et al., 2002), in coastal regions of China and Spain (Yu et al., 2019; Mc Figgans
et al., 2010; Mahajan et al., 2011) and in open water regions of North East Greenland (Dall'Osto et
al., 2018).  Most of these studies have identified biogenic emissions from marine algae as the main
precursors driving the new particle formation.



It is well documented that sulfuric acid (henceforth SA) is an important precursor to
NPF in most environments (Almeida et al., 2013; Kulmala et al., 2013; Croft et al., 2016; Jokinen et
al., 2017; Kirkby et al., 2011; Sipilä et al., 2010). The advancement in aerosol research, revealed  that
a binary system of SA and water is not sufficient to produce particles in ambient atmospheric
conditions without stabilizing compounds (Benson et al., 2008; Duplissy et al., 2016; Kirkby et al.,
2011). More recently, it has been found that a ternary system involving SA-ammonia-water or SA-
amines-water yield much higher nucleation rates as compared to the binary system (Kulmala et al.,
2000; Benson et al., 2008; Almeida et al., 2013; Glasoe et al., 2015; Kürten et al., 2016). In addition
to these systems, organic compounds which are highly oxygenated - thus less volatile- have been
found to contribute to secondary organic aerosol (SOA) mass in forested areas, mountain tops and
anthropogenically influenced field sites (Ehn et al., 2014; Pierce et al., 2011; Riipinen et al., 2012;
Zhang et al., 2009; Heikkinen et al. 2020;  et al., 2020) and laboratory experiments  have shown that
they can contribute also to the first steps of NPF  (Simon et al., 2020; Lehtipalo et al., 2018; Kirkby
et al., 2016; Tröstl et al., 2016) .
Furthermore, another important molecular class, iodine as well as its related oxidized
species play a crucial role in NPF especially in coastal areas (Allan et al., 2015; Mahajan et al., 2009;
Raso et al., 2017; Sipilä et al., 2016) and in pristine marine locations (Baccarini et al., 2020; Beck et
al., 2021; He et al., 2021). Some previous studies have reported the emissions of $I_2$ from the
macroalgae at coastal sites (Huang et al., 2010; Peters et al., 2005; Saiz-Lopez and Plane, 2004).
Several studies from coastal sites like Roscoff, France (Mahajan et al., 2009; McFiggans et al., 2010),
Mace Head, Ireland (O'Dowd et al., 2002) and other European coastlines (Mahajan et al., 2011; Saiz-
Lopez et al., 2012) have reported iodine species initiating NPF. The reported events can be considered
as aerosol burst events with high aerosol concentration and having exceptionally high initial growth
rates (GR) (O'Dowd et al., 2002; McFiggans et al., 2004; Mahajan, et al., 2011). The study from the
Roscoff coast suggests that the daytime emissions of $I_2$ (produced by macroalgae) during low tides
drives the particle formation (McFiggans et al., 2010). The iodine oxides and/or oxoacids formed by
the biogenic emissions from the micro- and macroalgae near the coastal regions are capable of self-
clustering, which could form new particles with a diameter <3 nm and sometimes with a high gas
concentration reaching up to $10^6$ $cm^{-3}$ or even more. Recent studies have shown that ion-induced iodic
acid nucleation proceeds at the kinetic limit and the overall nucleation rates (ion-induced nucleation
+ neutral nucleation) driven by iodine oxoacids (iodic acid, $HIO_3$ and iodous acid, $HIO_2$) are high,
even exceeding the rates of well-known precursors of NPF (He et al., 2021b, 2021a): sulfuric acid
with roughly 100 pptv ammonia under similar conditions (Sipilä et al., 2010). The rapid photolysis
of $I_2$, (< 10 s), produces I atoms above the ocean surface and can be detected in high concentrations



close to the source region (McFiggans et al., 2010). However, the compounds with longer lifetimes
such as $CH_3I$ (two days) provide a source of iodine throughout the troposphere (Saiz-Lopez et al.

2012).

Another important gaseous precursor of NPF, SA, could have different sources in

Helsinki (Dada et al., 2020b; Väkevä et al., 2000). Dimethyl sulfide (DMS) oxidation by OH radical
in the daytime and by nitrate radical in the nighttime yields other aerosol precursor gases, such as
methane sulfonic acid (henceforth, MSA) and SA (Barnes et al., 2006), which play a crucial role in
the NPF processes. In a marine coastal environment, MSA concentrations, which  are typically lower
than those of SA, could be as low as 10% of SA concentration and could maximally reach 100% of
SA concentration (Eisele and Tanner, 1993), yet MSA is a potential candidate to participate in the
atmospheric nucleation and growth processes (Beck et al., 2021).  The stability of heterogeneous
MSA clusters have been studied in laboratory and modelling studies (Chen et al., 2020, 2015, 2016)
but no study has yet documented MSA clusters in the field. The limited NPF studies in the coastal
regions and the dynamic coastal atmospheric chemistry drives the motivation of this research. No
detailed studies of NPF events were done before taking into account biogenic precursor gases near
the coast of Finland despite the fact that extensive cyanobacteria blooms occur every year in the Baltic
Sea region and neighboring water bodies (including Finnish lakes) (Kahru and Elmgren 2014), which
could be a significant source of iodine species, SA and MSA. In addition, there is a lack of studies
reporting the MSA concentrations in the atmosphere of Finland. Thus, this study was undertaken to
understand particle formation processes, when the air plume is a mixture of anthropogenic as well as
biogenic gases and particles as in the coastal semi-urban location in Helsinki, Finland.
Investigating the origin and chemistry of NPF events in an urban coastal setting could be quite
challenging since precursor vapors of nucleation are likely a mixture of both anthropogenic and
biogenic vapors from different sources. Further, pre-existing particles in the atmosphere affect the
occurrence of NPF events by acting as sink for precursor gases and freshly formed particles
preventing the latter from further growth. These parameters, in turn, are influenced by the local
meteorological parameters such as wind direction, wind speed, (air mass) turbulences especially at
the surface layer of the lower atmosphere. Coastal locations are dynamic environments with rapid
changes in meteorological parameters, also making the study of NPF more challenging. The
meteorological condition could likely govern the removal of particles from the air stream preventing
the growth of newly formed particles.

In this study, we aim at a thorough evaluation of aerosol precursor molecules with a

detailed (NPF events) analysis during the cyanobacterial bloom period, in the coastal-city of Helsinki,



Finland, from June to August (summer) 2019. This work evaluates the role of phytoplankton blooms
and meteorological parameters in the NPF events observed during the measurement period. We also
identify the major precursor vapor(s) and molecular clusters found during the aerosol events. Here,
we formulate the hypothesis that gaseous precursors formed from the biogenic emissions from the
surrounding marine areas could play an important role in the nucleation processes in Helsinki.
Although Helsinki is a coastal area yet the role of marine emissions on New Particle Formation
processes has not been studied before.

**2 Measurement Site and Methodology**

To understand the chemical composition of the precursor vapors emitted from various sources
around the site, the Chemical ionization Atmospheric Pressure interface-Time Of Flight mass
spectrometer (CI-APi-TOF) was operated from the $4^{th}$ floor laboratory of the Physicum building,
Kumpula campus, University of Helsinki (60° 12' N, 24° 58' E ; 49m , a.m.sl). The other aerosol and
trace gases instruments were operated at the SMEAR III station which is 180 m away from the mass
spectrometric measurement site (Station for Measuring Ecosystem-Atmosphere Relation (SMEAR
III), 60.20° N, 24.96° E; 25 m a.s.l.).

The measurement sites are surrounded by coastal water bodies (<4km,
Vanhankaupunginselkä), forests (<3km) and road connecting to the main city (<300m) as seen in
figure 1. Overall Helsinki is located on a relatively flat land on the coast of the Gulf of Finland. The
Helsinki Metropolitan area is about 765 km$^2$ with approximately one million inhabitants, counting
together the city of Helsinki and the neighboring cities of Espoo, Vantaa, and Kauniainen. The climate
in southern Finland can be classified as either marine or continental depending on the air-flows and
pressure systems. Either way, the weather is milder than typically at the same latitude (60°N) mainly
due to the Atlantic Ocean and the warm Gulf Stream.

The site and measurement period (25 June 2019–18 August 2019) selected for this
particular study are unique. We hypothesize that the biogenic emissions from summertime
cyanobacterial blooms in the Baltic Sea and the neighboring water bodies could influence the new
particle formation processes at this semi-urban location. The blooms in the Baltic Sea region are
recurring phenomena during the summer. Increasing temperatures and the excessive nutrient load in
the Baltic Sea promote algal growth (Kuosa et al., 2017; Suikkanen et al., 2007, 2013). According to
HELCOM (Baltic Marine Environment Protection Commission), the Baltic Sea has warmed 0.3° C
per decade, however after 1990 significantly faster at 0.6° C per decade and in Finnish coastal areas
the warming is even faster with a 2° C increase since 1990 (Humborg et al. 2019). The amount of
blue-green algae (i.e. cyanobacteria) has shown a statistically significant increase in open sea areas



in the Gulf of Finland, Sea of Åland and the Sea of Bothnia in the last 40 years (Kahru and Elmgren,
2014). Although nutrient pollution has showed a decreasing trend (Andersen et al., 2017), growing
oxygen deficient waters recirculate nutrients and perpetuate cyanobacterial blooms (Funkey et al.,
2014). The increase in frequency and intensity of cyanobacterial blooms would increase the potential
emission of biogenic gases changing the composition of the overlying atmosphere and the atmosphere
of the neighboring sites, depending on the meteorological conditions.

### 2.1 Main Instruments

The Atmospheric Pressure interface-Time Of Flight (APi-TOF) mass spectrometer is

the state-of-the-art instrument for gas phase chemical composition investigations including aerosol
precursor characterizations. Here the instrument is coupled with a chemical ionization (CI) inlet in
order to measure neutral gas-phase molecules that are clustered and charged with a reagent ion. The
Time Of Flight (TOF) mass analyzer can detect molecules with masses up to 2000 Th with a mass
resolution of 3600 Th/Th. More details on the working principle of the instrument and calibrations
can be found in earlier studies (Junninen et al., 2010, Jokinen et al., 2012; Kürten et al., 2014). The
sampled air was drawn in through a 1 m-long, "¾" diameter stainless steel tube with an average flow
rate of 10 lpm. In this study, the chemical ionization was done via nitrate ions ($NO_3^-$) through X-ray
exposure of nitric acid ($HNO_3$, flow rate: 3 mlpm), saturating the sheath air flow entering the CI (flow
rate: 30 lpm), the inlet flow of 10 lpm was reached by using a 40 lpm total flow. The instrument was
calibrated prior to the experiment according to (Kürten et al., 2012) resulting in a calibration factor
of $1.45 \times 10^9$ molecule per normalized unit signal including the diffusion losses in the inlet line. The
resulting data (i.e. obtained signals) were averaged to 60 min before the mass calibration step
performed through the MATLAB based program tofTools (Junninen et al., 2010). The final
concentration of the gases were derived using the equation mentioned in Jokinen et al., 2012.
Uncertainties of absolute concentration measured by CI-APi-TOF are estimated to be in the order of
±50%, while the uncertainties of relative changes in the concentration are smaller than 10% (Ehn et
al., 2014). SA, MSA, IA concentrations and normalized signals of specific HOMs (all figures
presented in SI) found in the study are calculated using high resolution peak fitting data. Please note
that all HOM sum (monomers and dimers) concentrations were calculated from the Unit Mass
Resolution (UMR) data. The

Neutral cluster and Air Ion spectrometer (NAIS, Airel Ltd., Estonia, Manninen et al.,

2010; Mirme and Mirme, 2013) was used to measure the number size distribution of both positive
and negative ions between 0.8 nm and 42.0 nm (electric mobility diameter). The NAIS also measures



the number size distribution of total particles (neutral and naturally charged) between 2.5–42.0 nm.
It uses two identical differential mobility analyzers (DMA, (Knutson and Whitby, 1975)) for
simultaneous measurement of positive and negative ions. The flow rate of the instrument is 60 lpm
which is split into 30 lpm for each DMA. The instrument was installed in the SMEAR III station. The
data was recorded every 2 s.

Larger particles of 6–820 nm were measured using a twin differential mobility particle

sizer (DMPS) (Aalto et al., 2001). The instrument was installed in the SMEAR III station. The time
resolution of data is 10 minutes.

The size distribution of 1–3 nm particles was measured by a Particle Size Magnifier

(PSM, Airmodus Ltd., Finland; Vanhanen et al., 2011) in series with a condensation particle counter
(Airmodus Ltd., Finland). The PSM was operated by scanning the flow 0.1–1.3 lpm (continuously
changing the saturator flow rate) which allows determining the 1–3 nm particle concentration and
calculation of particle size distribution. The data was recorded for each second and the duration of
each scan was fixed to 240 s. The raw data inversion was carried out through the  kernel method
(Chan et al., 2020; Lehtipalo et al., 2014). The raw data of the PSM employed a pretreatment filter
that calculates the correlation between the observed particle concentration and the saturator flow rate
of a single scan and discards scans with significant non-correlation or negative correlation (Chan et
al., 2020).

**2.2 Back trajectory calculations**
Back trajectories of the different NPF event  days were calculated using the data from the Global data
Assimilation System (GDAS) as input into the NOAA Hybrid Single-Particle Lagrangian Integrated
Trajectory (HYSPLIT) model (http://www.arl.noaa.gov/ready/, Rolph et al., 2017; Stein et al., 2015).
We used the isentropic trajectories as they incorporate vertical transport components. The 24 h back
trajectories were calculated at an arrival height of 100 m a.g.l. The new trajectory starts every 6 hours.
The frequency (%) of trajectory was calculated with the following equation (Eq. (1)).

$$\text{Traj. Freq.} = \frac{100 \times number\ of\ trajectories\ passing\ through\ each\ grid\ square}{number\ of\ trajectories} \qquad (1)$$

The trajectory analysis was also performed using the Lagrangian particle dispersion model Flexpart
v10.4  (Pisso et al., 2019; Stohl et al., 2005) mainly to assess the residence times of the air masses.
Flexpart is a stochastic model used to compute trajectories of hypothetical particles, based on mean
as well as turbulent and diffusive flow (Pisso et al., 2019). We have used Flexpart along with ECMWF
ERA-Interim wind-fields which has a spatial resolution of 1°×1° at three hour temporal resolution





244 (Pisso et al., 2019). Flexpart was used to simulate 3-day backward trajectories starting from the

245 particle release point located at SMEAR III (24.5° E, 60.1° N) for the event days. The residence times

246 were normalized for clarity in the all the figures and is shown on a scale of 0 to 1 (Results are included

247 in the supplementary information).

249 **2.3 Meteorological and other supporting data**

250 The meteorological data such as wind speed, wind direction, temperature, pressure, relative humidity

251 and other supporting datasets e.g chlorophyll (Chl-*a*), $SO_2$, $O_3$ concentration and sea level information

252 was additionally used to interpret the NPF events and support the observations of this work (See table

253 S1 for details). The Chl-*a* satellite images were mapped through the GlobColour level-3. The

254 GlobColour level-3 mapped products present merged data from SeaWIFS, MERIS, MODIS AQUA,

255 VIIRS (0'Reily et al., 2000) sensors to provide robust and high coverage data for Chl-*a*

256 measurements. The merging processes are described in Mangin, 2017. In this study, weighted average

257 method (AVW) for retrieving daily Chl-*a* concentration (mg m$^{-3}$) for latitude: 45 °N to 80 °N and

258 longitude: 20 °W to 60 °E was used. The GlobColour level-3 binned products have a resolution of

259 1/24° at the equator (i.e. around 4.63 km) for global products (Mangin, 2017).The details of these

260 additional supporting data given in SI (Table S1).

262 **2.4 Formation and growth rate calculations**

263 The growth rates (GR) were calculated based on the 50% appearance time method using the NAIS

264 ion data from both polarities, depending on the better quality polarity (Dada et al., 2020a; Dal Maso

265 et al., 2016; Lehtipalo et al., 2014). This method uses particle number concentration at different size

266 bins (D*p*), which are recorded as a function of time. The "appearance time" of particles of size D*p* is

267 the time when their number concentration reaches 50% of its maximum value during the NPF event.

268 To estimate the maximum GR (kinetic) that can be explained by the condensation of certain vapors,

269 two parametrization methods were used, first by Nieminen et al., 2010 for IA and MSA and the

270 second by Stolzenburg et al., 2020 for SA. The growth estimation from SA condensation recently

271 provided by Stolzenburg et al., 2020 also takes into account the hydration of SA particles and dipole-

272 dipole enhancement which is responsible for increasing the collision rate between neutral molecules

273 and neutral particles. As these parameters were not known for IA and MSA, we used the method by

274 Nieminen et al. (2010) for them. The growth  due to MSA could be  slightly overestimation by this

275 method (Beck et al., 2021) since the parameterization is based on the assumption of irreversible

276 condensation, but MSA rapidly partitions between gas and particle phases if suitable meteorological



conditions prevail. The calculated kinetic GR was compared with the total measured particle GR to
determine the contribution of each vapor to the growth process (discussed in further sections).

The formation rate of the total particles of diameter 1.5 nm is calculated using the time

derivative of the particle number concentration measured using the PSM in the size range 1.5– 3 nm.
The formation rate was corrected for the coagulation losses and growth out of the bin following the
method explained by Kulmala et al. 2012. The formation rate of the charged particles was calculated
from the time derivative of ions measured using the NAIS in ion mode in size range 1.5–3 nm from
both polarities. The formation rate of ions was corrected for coagulation sink, growth outside of the
bin, ion-ion recombination and ion-neutral attachment as previously discussed in Kulmala et al. 2012.

**2.5 Condensation sink**
The condensation sink (CS) plays an important role in understanding aerosol dynamics. This
parameter determines  how fast gas molecules will condense on the pre-existing particles (Dal Maso
et al., 2002; Kulmala et al., 2005, 2012). In this study, *CS* has been calculated by using the DMPS
data, according to Pirjola et al., 1999.

**3. Results and discussions**
**3.1 Meteorological parameters and cyanobacterial bloom during the study.**
**3.1.1 Meteorological Parameters**
The meteorological parameters, especially the wind speed, wind direction and ambient temperature,
varied significantly during the study period. The time format in the entire study is UTC+02:00 h. This
study period includes the hottest summer days of Finland in year 2019. The average temperature
during 17–28 July (the warmest period) was 21.6° C with a maximum temperature of 31.6° C recorded
on the 28 July (Fig. 2). Temperature starts to decrease after 29 July. The average temperature in
August was 16.5° C with a maximum temperature of 21.9° C recorded on 5 August 2019.

The wind direction was highly variable during June-July period. The wind direction in

July was mostly from the sectors 270°–320° (West-Northwest) and 90°–150° (East-South East). In
August, the wind gained more stability and was dominantly blowing from 180°–270° (South-West)
(Fig. 2). The wind speed also showed high variability in June-July. The wind speeds during June and
early weeks of July were mostly >6.5 m s$^{-1}$, followed by a bit calmer mid-July (mostly <=4 m s$^{-1}$)
with preceding high winds in end of July until mid-August (gusts of winds > 5.2 m s$^{-1}$) (Fig. 2).
However, the average wind speeds in both the months was 3 m s$^{-1}$. The average daylight hours in July
were 17-18 hours with the daytime hours between 04:00–22:00 h which starts to decrease in August
to 15–16 hours of daylight per day (05:00 h – 21:00 h) as per the Global radiation data obtained from





SMEAR III station for the study period. Therefore, the actual nighttime hours in our measurement
site can considered from 23:00 h–03:00 h during Finnish summers.

### 3.1.2 Cyanobacterial bloom conditions during the study

The Baltic Sea (defined from 53° N to 66° N latitude and from 10° E to 30° E longitude inclusive of
Gulf of Bothnia, Gulf of Finland and Gulf of Riga) is characterized by usually two algal blooms
occurring in early Spring (mostly diatoms) and a Summer bloom increasingly dominated by
cyanobacteria (blue green algae). The summer bloom period selected for this study was typically
characterized by cyanobacteria. When these microscopic cyanobacteria multiply and aggregate, they
are seen as blue-green patches or scum-like layers over the surface of lakes and marine waters. The
warm early summer temperatures (during June) resulted in a cyanobacterial bloom (Finnish national
monitoring; SYKE, 2019). However, the weather conditions in July began changing with high winds
causing the cyanobacteria to be highly mixed in the water column, which reduced bloom intensity at
the sea surface to lower than normal in July and August. Subsequently, temperatures were lower in
August as compared to July and windier as compared to other summer months. These windy
conditions kept the lake cyanobacteria well mixed in the water. The northern Baltic Sea, including
the Gulf of Finland, the Southern parts of the Åland islands and even the Bothnian Sea occasionally
observed massive blooms of cyanobacteria during June-August 2019. However, the bloom intensity
of cyanobacteria at the coastal areas were intermittent and changed rapidly due to the spatial
complexity of the coastline and variable winds and currents.
These cyanobacterial blooms are generally dominated by three taxa, *Nodularia*
*spumigena* , *Aphanizomenon* sp. and *Dolichospermum* sp. (Knutson et al., 2016; Kownacka et al.,
2020). In the Baltic Sea, these cyanobacteria actually contribute the most to the total pelagic nitrogen
fixation (Klawonn et al., 2016). Other potential primary producers emitting vapors are the littoral
macroalgae growing along the shallow coastline. For example, the perennial macroalgae, *Fucus*
*vesiculosus* covers large areas of the coastal areas of Baltic Sea, where they support very high biomass
and high productivity (Attard et al., 2019). Low sea levels (0.2–0.8 m) were recorded in mid-July (11
July 2019–27 July 2019) during the period when high temperatures (20° C and above) prevailed
(Fig.2) in our study region. During these conditions, contributors to emissions might be a mix of both
coastal macroalgae and open sea microalgae. There is a possibility that reasonably, large extents of
coastal macroalgae, including *F. vesiculosus*, were exposed to direct sunlight (in shallow waters or
low tide conditions) hence making this time window favorable for observing potentially high
emissions in gas phase from macroalgae, in addition to the emissions from cyanobacterial blooms.
However, in the semi-urban/coastal setting of this measurement site, there could be various other





parameters, which also could play a role in determining the concentrations of the biogenic emissions;
for example the wind speed and wind direction. The atmosphere in this semi-urban coastal location
is itself a cocktail of various vapors, oxidants and particles, which would affect the quantification,
source apportionment and characterization of the biogenic emissions.

**3.2 Precursor vapor concentrations and their sources**
The measured daytime precursor vapor concentrations showed a regular diurnal cycle consistent with
the photochemical production of SA and IA in 90% of the days in this study. SA, key precursor of
atmospheric NPF, is formed mainly by reaction of sulphur dioxide with OH-radicals, which is
predominantly controlled by the photochemical cycles (e.g. Sipilä et al., 2010; Jokinen et al., 2017).
The mean (whole day) concentration of SA in July and August was $2.98 \times 10^6$ molec. cm$^{-3}$ and 2.67
$\times 10^6$ molec. cm$^{-3}$ respectively. The mean concentration is slightly lower than compared to the
concentrations of SA reported by very recent study measured in a Helsinki street canyon, $1 \times 10^7$
molec. cm$^{-3}$ (Olin et al., 2020) but similar to the SA concentration measured at the SMEAR III station
in 2018 (Okuljar et al., 2021). In the study of Olin et al., 2020, SA concentrations were greatly
affected by vehicular traffic as the site is situated at a busy street canyon. The SMEAR III is
considered as a background site much less affected by vehicular traffic (Okuljar et al., 2021). In
comparison to other locations, the daytime SA concentration in pristine Antarctic region has been
reported from $10^5$ up to $10^7$ molec. cm$^{-3}$ (Mauldin et al., 2001, Jokinen et al., 2018), $10^6$ molec. cm$^{-3}$
in remote continental, remote marine and forest regions and $10^7$ molec. cm$^{-3}$ in urban and rural
agricultural lands using the same technique as in here (Berresheim et al., 2000; Kuang et al., 2008;
Petäjä et al., 2009; Kurtén et al., 2011; Zheng et al., 2011; Chen et al., 2012; Jokinen et al., 2012;
2017, Kürten et al., 2014; Bianchi et al., 2016; Baalbaki et al., 2021; Dada et al., 2020b). It has been
well documented that SA contributes to aerosol formation and growth processes (Boy et al., 2008;
Eisele et al., 2006; Fiedler et al., 2005; Iida et al., 2008; Sarnela et al., 2015; Jokinen et al., 2018;
Kürten et al., 2015, 2016; Mauldin et al., 2001; Paasonen et al., 2010; Wang et al., 2011; Weber et
al., 1998, 1999; Yao et al., 2018; Dada et al., 2020b). Most of these studies are conclusive that SA
concentration in the atmosphere depends on the anthropogenic and biogenic activities around the site.
In the coastal marine boundary layer, the MSA concentration is typically 10–100% of
that of SA (Berresheim et al., 2002; Eisele and Tanner, 1993). Until recently, no studies have been
found to report MSA and IA concentrations in coastal/urban setting of Finland. The mean (whole
day) concentration of MSA in July and August was almost similar, $4 \times 10^5$ molec. cm$^{-3}$. The mean
concentration of IA in July and August was $1.27 \times 10^6$ molec. cm$^{-3}$ and $2.69 \times 10^6$ molec. cm$^{-3}$,
respectively, showing two times increase in IA concentrations in August (Fig. S1). A similar increase



in IA concentrations from summer to autumn were observed in the Arctic Ocean, where the increase
in IA was attributed to the freezing onset of the pack ice and increase in ozone concentrations
(Baccarini et al., 2020). However, here the increase is mainly due to the change in the air mass arriving
at the experimental site, enriched with biogenic emissions from the blooms. For the same period, the
CI-APi-TOF data shows exceptionally high concentrations of highly oxygenated organic molecules
(HOMs), with monomer concentrations (300–450 amu) of $10^8$ molec. cm$^{-3}$ and HOM dimer
concentrations (450–600 amu) of $10^8$ molec. cm$^{-3}$ as well (Fig.S2).
In more details, the IA concentration rises one order of magnitude, from $10^6$ to $10^7$
during the 11–17 August, when the wind direction changes abruptly (from 280°–360°to 180°–230° ,
marine air mass, Fig. 3). We found that during the marine air (180°–230°, South Easterly, over Gulf
of Finland and South westerly, over Northern Baltic sea) influence over the study region the average
noontime maximum of SA, IA is on the order of $10^7$ molec. cm$^{-3}$ and MSA is around $10^6$ molec. cm$^{-3}$
(Fig. 3). This is one order of magnitude higher concentration than when the wind was from over
land (Fig. 3).
The highest concentration, $3.2 \times 10^7$ molec. cm$^{-3}$ of IA was observed when the wind is
coming from the Baltic sea sector, whereas the highest SA concentrations (~$3.0 \times 10^7$ molec. cm$^{-3}$)
we observe when air mass travelled over the countries of Estonia and Russia crossing Gulf of Finland
before entering the measurement site (land+sea region). The connection between the aerosol
precursors and the wind direction can be observed in the cases where the wind direction changes
rapidly. The highest IA concentration was recorded when the wind direction changes after the 4
August, 180°–230° (the Baltic Sea region). The change in wind direction was clearly reflected in a
reversal of the concentration trends of SA and IA (Fig. 3). It was observed that the winds coming
from 80°–180° or 250°–280° (land-sea region, Fig. 3) were SA rich air masses. This comprises of the
landmasses of south and northeastern Finland, Northern Russia, part of Gulf of Finland and Estonia
and North-North western part of Finland including a part of northernmost Gulf of Bothnia. The sector
0°–90° or 280°–360° (land, Fig. 3) consists mostly of urban cities.
During the entire study period, when the air plume passed over the northern Baltic Sea region and
the wind speed was high enough (> 4m s$^{-1}$) high concentrations of IA was observed. While IA can be
exclusively sourced from the marine and biogenic emissions (Mahajan et al., 2011; O'Dowd et al.,
2002; Sipilä et al., 2016; Carpenter et al., 2021), SA could be biogenic or /and anthropogenic. Further,
the temperatures prevailing during this period may have facilitated the DMS oxidation at a higher
rate, which forms the source of biogenic SA and MSA. However, this is not a very simple equation,
since this fractional yield of (biogenic) SA from DMS oxidation additionally also depends on the
atmospheric $NO_x$ ($NO + NO_2$) and $HO_x$ ($OH + HO_2$) levels and on the scavenging of $SO_2$ by sea salt



or cloud droplets (Hoffmann et al., 2016). The anthropogenic sources of SA for this site includes
vehicular or ship traffic especially considering that there is a city road just 250 m and a harbor 6 km
away from the measurement site. We explored the correlations of SA to a biogenic proxy, MSA and
correlation with $NO_x$ (anthropogenic proxy) to have a clear source apportionment of SA (Fig.4). $SO_2$
could not be treated entirely as anthropogenic proxy as it can be sourced from DMS oxidation as well.
The good correlations ($r_s$ >0.6, Fig. 4a and 4b) between SA and MSA during the study period
(June–August) could suggest that they were sourced from a common biogenic source, the DMS
emission from the cyanobacterial bloom. Good correlations of SA and MSA was also found in August
($r_s$ = 0.8, Fig.4b) when the air mass was mostly marine (and/or from the Finnish coastline, Fig. 3).
Another observation was that $SO_2$ also shows some correlations with SA in both June-July and August
study periods ($r_s$ = 0.4, Fig. 4c and 4d), but not as significant as SA and MSA correlations. $SO_2$ can
have different sources unlike MSA which is mostly biogenic, hence these observations could possibly
indicate SA was more biogenic than from other sources. But we cannot be very accurate in this
estimation only by analyzing the correlation coefficients since both MSA and SA can have a similar
daily cycles due to the oxidation pathways.
Both $SO_2$ and MSA are the oxidation products of DMS (produced by phytoplanktons, including
some cyanobacteria), oxidized through OH and $NO_3$ radical (Chen et al., 2000). Some of the previous
chamber studies have confirmed that $SO_2$ is the major intermediate products formed from DMS
oxidation (Sørensen et al., 1996; Berresheim et al., 1995). The $SO_2$ could be oxidized to SA ($OH/O_2$
oxidation) during the transport. Since our experimental site was surrounded by water bodies and the
summer season had enriched most of these freshwater and marine waters with abundant
cyanobacterial blooms, this biogenic SA contribution to the study site has to be accounted when
analyzing the sources of SA. However, $SO_2$ can also be sourced from various anthropogenic activities
and can be oxidized to SA. In Finland the major sources of anthropogenic $SO_2$ is the public power
industries contributing to almost 90% to the total $SO_2$ emissions in Finland in the year 2019, while
transport contributing to less than 1% according to the emission inventory prepared by Finnish
Environment Institute, SYKE (Finnish Air Pollution Inventory; ymparisto.fi/en-
US/Maps_and_statistics/Air_pollutant_emissions). Further the maximum data points of high
concentrations of $SO_2$ ($\sim 10^7$ molec. cm$^{-3}$) were not observed during the traffic hours in June-July-
August (Fig. 4c and 4d) another possible indication that biogenic sources could be contributing to the
$SO_2$ concentrations and thus SA concentrations near the study site.
The emission inventory of Finland for the year 2019 indicated that sources of $NO_x$ as $NO_2$ were
mainly the power industries (41.5%) and the transport sources (41%) (ymparisto.fi/en-
US/Maps_and_statistics/Air_pollutant_emissions). These sources are indeed the most significant





sources of $NO_x$ globally (Meixner and Yang, 2006) $NO_x$, definitive proxy of anthropogenic influence
shows a poor correlation with SA ($r_s$=0.28, Fig. 4e) during June-July also suggest insignificant effect
of traffic on the SA concentrations. Unfortunately, the $NO_x$ data from August was unavailable due to
instrument malfunction.
After carefully analyzing the data presented in Figure 3, where we observe high SA concentrations
even when the air mass was marine and the good correlations of SA-MSA (inclusive of insignificant
correlations of SA-$NO_x$) (Fig. 4) indicate towards a greater possibility of the influence of biogenic
emissions on the concentrations of SA as compared to the anthropogenic emissions.

**3.3 Types of nucleation events during the study**
During, 25 June 2019–19 August 2019, we observe a number of NPF events characterized by a short
appearance of ultrafine particles in the number size distribution lasting for less than one hour. These
so-called bursts /spikes appearing at small sizes (sub-3 nm) are indicative of local clustering and NPF
processes in contrast to regional events, where it is possible to follow the growing particle mode for
several hours (Dada et al., 2018; Dal Maso et al., 2005). We do observe transported events (events
with a growing particle mode, but no small particles forming at the site) and non-events days but they
are not included in the analysis. This section discusses the occurrence of local and regional new
particle formation events with the focus on: 1) trace gases variability during the event days, 2) the
evolution of different sized particles during these events, 3) the impact of meteorological parameters
and 4) the effect of cyanobacterial bloom on the events.
**3.3.1 Nucleation: Regional and Local events**
A regional NPF event was observed on 30 June 2019, which starts at 08:45 h and ends at 13:23 h (Fig
5a). The negative ion clusters start to increase in concentration first at 08:45 h (Fig. 5b) concurrent
with the increase in concentration of the smallest particles (<3nm) from $10^2$ to $10^3$ cm$^{-3}$ (Fig. 5c).
Subsequently, SA concentration doubles from $2\times10^6$ to $4\times10^6$ molec. cm$^{-3}$ (Fig. 5d), while the particle
formation rate at 1.5 nm ($J_{1.5}$) increasing from 0.3 cm$^{-3}$ s$^{-1}$ to 0.6 cm$^{-3}$ s$^{-1}$. $J_{1.5}$ was much higher than
either of $J^+_{1.5}$ and $J^-_{1.5}$, thus indicating a neutral formation pathway rather than an ion mediated one.
Further we also observe local clustering event at 15:00 h with simultaneous increase of concentration
of SA and HOMs along with increase in the smallest particle concentration. This possibly indicates
the role of SA and HOMs in the nucleation initiation. The work of Okuljar et al. (2021) also report
an increase in sub-3 nm particles with a simultaneous increase in SA concentration at the SMEAR III
site, supporting our observations. However, the role of HOMs in nucleation initiation has not been
explored at this site.



A clear increase in nucleation mode particles is seen during the event, starting at 08:45

h (234 cm$^{-3}$) and reaching its maximum at 12:30 h (4589 cm$^{-3}$). This increase in concentration of the
nucleation mode particles was followed by the increase in concentration of Aitken mode and
accumulation mode particles and continues for a couple of hours, indicating growth of particles (Fig.
5e), possibly reaching to CCN relevant sizes. The growth continues until the wind direction suddenly
changed after 12:00 h (Fig. 5d), that apparently discontinued the precursor vapor source to our site.
After the change in local wind direction, the observed SA and IA slightly increase, and we still
observe local clustering (formation of small ions and particles), but no continuous growth typical for
regional events. Figure 5f shows that >40% of the trajectories passes above the Swedish island of
Gotland towards southern part of Bothnian Sea. The MODIS data shows that the bloom was present
in the Bothnian Sea, but not quite dense as compared to the southern Baltic Sea (south of Gotland
island) and the northern part of the Gulf of Finland.  The majority of the trajectories did not pass over
the dense cyanobacterial bloom patch during this day (Fig. 5f). The calculated (normalized) residence
time was higher over the neighboring cities of Helsinki (Southwestern side) and parts of Bothnian
Sea during the event time (see Fig. S3). Thus the land based anthropogenic activities and biogenic
sources both can be contributing to SA concentrations for this event; here we cannot exactly quantify
the source types for SA.  However, the source of SA from the local sources such as vehicular traffic
around our measurement site is small (as discussed above) but cannot be completely ignored (Olin et
al., 2020).

The high signals (normalized) of DMA-SA cluster seen during the entire event (rising

from the start of NPF event) indicates SA clusters initiate the event (Fig.S4a). The increase of HOMs
is also clearly observed during the event Fig. S4b. Therefore we suggest that nucleation and growth
of particles was possibly due to SA-organics which ensures that particles reach the CCN and thus
climate relevant diameters.

The particle GR (7–25 nm) for this event was 16.5 nm h$^{-1}$, which is typical of a coastal

site. Even when several condensing vapors participate in the growth process, growth rates typically
do not exceed 20 nm h$^{-1}$ (Kulmala et al., 2004). The GR for organics was calculated after subtracting
the combined contribution of the GR of SA, IA and MSA from the measured particle GR. The GR
for organics should be treated as an estimation since no separate GR calculations and assumptions
were used.  The calculated growth rates (GR) shows that SA can explain maximum 41% of the growth
of sub-3 nm particles, while IA and MSA can explain only <1% of the GR in this size range. The GR
by SA in the bigger size fraction (7–25 nm) was only 0.51 nm h$^{-1}$ explaining only 3% of the measured
growth rate of particles. This means that vapors other than SA, IA and MSA were responsible for
96% of the measured particle growth. These other vapors could include different organics since



organics are known to contribute to growth of particles (Kulmala et al., 1998, 2004; Riipinen et al.,
2012; Zheng et al., 2020) and explain particle growth in the boreal forest (Ehn et al., 2014).

Another example of regional event (neutral nucleation) probably driven by SA and organics was

observed on 30 July 2019 (Fig. S5) which lasts for around four hours. The trajectory frequency plots
showed that most of the trajectories were from the northern land areas (including urban cities and
boreal forests) of Finland with highest residence times over these land regions (Fig. S6). Therefore,
the precursor gases from the biogenic origin, IA and MSA do not show a significant concentration
increase during this event and hence assumed to be contributing insignificantly to this event. The
greater residence times over the land areas clearly support the high SA and organic concentrations
seen during the event indicating a SA-HOMs driven local event (Fig. S6). In this case, the growth
due to SA explains 60% of growth of sub-3nm particles compared to 41% when the dominating
trajectories passed over the Gulf of Finland (Fig. 5, 30 June 2019). Still, as for the previous case, a
major fraction of the growth in the 3–7 nm range remains unexplained by the available acids (SA, IA,
MSA) and is expected to be related to organic material being abundant. The GRs explained by SA
in both sub-3 nm (1.93 nm h$^{-1}$) and 3–7 nm (1.46 nm h$^{-1}$) size ranges are 58-59% higher than on 30
June 2019 (0.79 nm h$^{-1}$ and 0.61 nm h$^{-1}$ for sub-3 nm and 3–7 nm, respectively) which could be
explained by the increase in SA by 52% on 30 July 2019. Thus, the events on 30 June and 30 July
possibly occur via the nucleation of sulfuric acid (possibly stabilized by bases eg. ammonia or amines)
and the HOMs contribute to growth of particles and possibly in nucleation as well.

**3.3.2 Nucleation: Burst events**
**Case 1: Biogenic IA nucleation- burst/spike events, 11 August.2019**
Intense burst events are frequently observed at coastal sites accompanied with high concentrations of
IA (O'Dowd et al., 2002; Rong et al., 2020; Sipilä et al., 2016). Two of such bursts or spike events
were observed on 11 August 2019 at 04:00 h and 13:00 h (Fig. 7a). Only the second spike event was
observed in the NAIS size distribution with a higher intensity in the negative mode at 13:00 h (Fig.
7b). During both these spike events we observe the formation of clusters (1.5 nm) and the formation
rate ($J_{1.5}$) increases from 0.2 to 3.7 cm$^{-3}$s$^{-1}$ during the event with a simultaneous significant increase
in the sub-3 nm particle concentrations from ~100 to >2000 cm$^{-3}$ (Fig. 7c). $J_{1.5}^+$ and $J_{1.5}^-$ remain lower
than the total formation rate indicating this event to be a case of neutral nucleation. At the same time,
IA shows increase in concentration from 9.2×10$^5$ molec. cm$^{-3}$ at 03:00 h to 1.2×10$^6$ molec. cm$^{-3}$ at
04:00 h. During this event the air masses changes from 160° to 140° i.e the direction of the airmass
is changed to the Gulf of Finland. In the second burst (at 13:00 h), the IA concentration increases
from 2.3 ×10$^6$ to 7.3 ×10$^6$ molec. cm$^{-3}$ from 13:00 h to 14:00 h (Fig. 7d) with a slight change in wind



direction from 151° to 166° Most of these air masses are from the Gulf of Finland. SA concentration
also increased but remained lower than IA during both the burst/spike events indicating a possibility
that iodine oxoacid formation initiates cluster formation (He et al., 2021). We observe a growth of
particles until 15:00 h in the particle modes (NAIS data, Fig. 7B). However the particles are seen
reaching sizes up to size 100 nm (DMPS data , Fig. 7A). The organics almost remain constant within
the range of $2.5–3.1 \times 10^8$ molec. cm$^{-3}$. A further increase in IA concentration, $3.18 \times 10^7$ molec. cm$^{-3}$
occurs at 15:00 h, and the concentration remains in the range of $10^7$ molec. cm$^{-3}$ for another two hours
(Fig. 7d). This was the highest observed IA concentration in the entire measurement period. A recent
study by He et al., 2021, indicate that HIO$_3$ concentrations above $1 \times 10^7$ molec. cm$^{-3}$ leads to rapid
new particle formation at +10° C. At such concentrations the efficacy of iodine oxoacids to form
new particles exceeds that of the H$_2$SO$_4$-NH$_3$ system at the same acid concentrations. Thus, the
concentration of IA found in this event is capable of initiating nucleation, especially since the
concentration of IA being two times higher than SA during the start of the event. In addition, a clear
increase in the normalized signal of deprotonated IO$_3^-$ with no significant increase in DMA-SA cluster
during the event at 13:00 h (Fig. S7a). However, HNO$_3$-IO$_3^-$ cluster was the most abundant followed
by the H$_2$O-IO$_3^-$ cluster indicating this event to be IA-driven nucleation. Further, between 14:00–
15:00 h, when we observe the highest IA concentrations a subsequent growth of particles is noted.
We also observe an increasing number concentration of nucleation mode particles from 13:40 h (~650
cm$^{-3}$) to 14:40 h (~1800 cm$^{-3}$). After this one hour of intense clustering, the Aitken mode particles
also begin to increase in concentration from ~1300 cm$^{-3}$ to ~4800 cm$^{-3}$ during 15:00 h–18:00 h (Fig.
7E). The total particle concentration increased from ~2400 cm$^{-3}$ to ~6400 cm$^{-3}$ within an hour during
this burst event. We suggest that this burst event was possibly capable of producing particles big
enough to act as CCN.

The global radiation and brightness parameter suggest that 11 August 2019 was an

overall a cloudy day until 12:30 h (Fig. S8). The weather starts to turn into clear-sky after 13:00 h
when the brightness parameter increases from <0.3 to ~0.7 (Fig. S8). Impact of brightness parameter
on NPF is also observed in a previous study (Dada et al. 2017). The clearing of the sky could explain
the intense spike at 13:00 h in the particle number size distribution as well as in the acid
concentrations. For this particular case, we investigated further the source of such high IA
concentrations and we found that during this day, the cyanobacterial bloom was observed in three
intense patches in the central Baltic sea, southern Gulf of Finland (ship transect route between
Helsinki and Tallinn) and Gulf of Riga (Fig. 7f). The trajectory frequency analysis clearly shows that
the maximum frequency of trajectories was observed over southern Gulf of Finland (inclusive of the
coastal waters of Suomenlinna island) however we do see the air masses coming in from the central



Baltic sea as well which was characterized by intense bloom during this day (Fig. 7f). The sea level
was also low as it was observed to be 0.8–0.9 m in the coastal waters in around the measurement site
(Suomenlinna and Gulf of Finland coastal measurements sites), supporting the exposure of the macro
algae to sunlight which can be a good source of iodine precursors.
The residence time of the airmasses coming from the Gulf of Finland and Northern
Baltic Sea were longer than the residence time of the airmasses coming from the neighboring land
areas (Fig. S9) clearly explaining the source of high IA observed during the event, which is through
the blooms. Further, the airmass was completely marine at 15:00 h when the highest IA is recorded
supportive of the marine biogenic source of IA and its transport to the measurement site. The distance
from the Gulf of Finland to the measurement site is approximately 5-10 km. With the wind speed of
5 m s$^{-1}$ recorded during the event, it takes less than one hour for the emission to transfer to our
measurement site. By the time the air mass reached our measurement site from the emission source,
all of the I$_2$ was oxidized to IA. However, at this point we cannot differentiate between the sources
of IA from neighboring coastal waters and the central Baltic Sea but can speculate that most of the
IA observed could be sourced from the nearest coastal locations of Gulf of Finland.
Another burst/spike event driven by IA occurred on 14 August 2019 (Fig. S10) when
the IA concentration was found to be $8.54 \times 10^6$ molec.cm$^{-3}$ which was 2-3 times higher than SA
concentration ($4.2 \times 10^6$ molec.cm$^{-3}$). The event did not last more than 30 minutes. The precursor
vapor concentration was not large enough for the event to continue or the particles to grow further.
The meteorological conditions were very much similar to this event (11 August 2019). For this event
also, the airmass was marine with maximum residence times over the Gulf of Finland and Baltic Sea
regions. Vicinity of the emissions to the measurement site enabled the detection of these fast-forming
clusters (from the emissions).
**Case 2: Biogenic SA nucleation –multiple bursts events**
Another kind of event was observed on 15 August 2019 (Fig. 8a) where multiple particle bursts are
observed and the particles grow to sizes > 50 nm.

The formation rates for the smallest clusters for both the polarities were the same ($J_{1.5}^+$ and $J_{1.5}^-$) (Fig.
8b and c). This was also a case of neutral nucleation as inferred from the relatively high (as compared
to ions) J$_{1.5}$ (neutrals).  On 15 August there was a sudden change of wind direction from the $180°$–
$215°$ (prominent wind direction during 11–14 August) to $280°$ and a series of bursts is triggered with
the intense formation of clusters (<3 nm) at each burst (Fig. 8d). The two most intense burst events
(marked as dashed rectangles in Fig. 8a, b, d and e) were associated with an increase in SA from 2.4



to $6.43 \times 10^6$ molec. cm$^{-3}$ at 06:00 h, and 5.3 to $7.03 \times 10^6$ molec. cm$^{-3}$ at 14:00 h (Fig. 8d). A third
burst at 09:00 h showed an increase in SA from 3.4 to $6.25 \times 10^6$ molec. cm$^{-3}$ at 09:00 h interestingly
with IA$_{max}$: $3.14 \times 10^6$ molec. cm$^{-3}$. In all the three bursts a simultaneous increase in IA and MSA
from 03:00 h to 12:00 h is observed, but the SA concentration was two to three times higher than IA
and four to five times higher than MSA concentrations. The most intensive burst was at 14:00 h (as
compared burst to 6:00 h) when the SA was 3 times higher than IA. This burst was associated with a
significant increase in Aitken mode particle concentration (from 1490 at 14:00 h to 4300 cm$^{-3}$ at 15:00
h). The increase in accumulation particle concentration was seen just after one hour from the start of
the bursts for both events (06:00 h and 14:00 h). However the increase in accumulation mode particle
concentration for these two events was not very significant (~100cm$^{-3}$) although particles reaching a
size more than 80 nm (CCN relevant sizes) was observed. We saw DMA-SA clusters during the event
(Fig. S11) which supports the observation that this a SA-driven NPF event.
During both these events (in fact, all the smaller burst events observed during this day), the
trajectories were originating from Sweden (24 h prior to arrival). However, before entering the
measurement site the trajectories passed over the Southern part of Gulf of Bothnia and the trajectory
frequency was >70% when the wind passed over the cyanobacterial bloom region (Fig. 8f).

## 632 3.4 Possible contributions of biogenic emissions to Precursor gaseous vapors

Assuming insignificant anthropogenic SA contribution as discussed in section 3.2, we investigated
the other possible sources of SA by evaluating the type of algae present in the water bodies from
where the air masses travelled during the events. The marine algae produces
dimethylsulfoniopropionate (DMSP), which is capable of forming DMS, which subsequently
oxidizes into SA and MSA. While very few cyanobacterial species are capable of producing DMSP
(Karsten et al., 1996; Jonkers et al., 1998), and its concentration can vary considerably from one
species to another (Keller et al., 1989). Moreover, blooms could be well-mixed with other algal
species (ESA report, 2000) which are capable of producing DMSP. A recent experiment identified
*Aphanizomenon* as the only cyanobacteria producing DMS (Steinke et al., 2018). The  Gulfs of
Bothnia and Riga are dominated by the genus *Aphanizomenon* (Kownacka et al., 2020). In addition,
the Bothnian Sea and Gulf of Finland were found to be rich in cyanobacterial genera of
*Aphanizomenon* along with *Nodularia* and *Dolichospermum* (SYKE 2020).

A recent study also indicated that the abundance of DMS producing cyanobacteria,

*Aphanizomenon* has increased in the Bothnian Sea due to decreasing salinity (Olofsson et al., 2020).
Moreover, marine waters themselves are a large source of DMS (Kettle and Andreae, 2000)



explaining the contribution of biogenic SA in the above-mentioned burst events (15 August 2019).
Hence to conclude the gulf regions surrounding the experimental site could be potential sources of
biogenic SA. Moreover, high iodine emissions could be expected over the Baltic Sea proper region
due to the presence of the macroalgal species which are well established and adapted in the Baltic
Sea despite its low salinity (Kautsky and Kautsky, 2000; Schagerström et al., 2014) (high IA on 11
August 2019 event day). The rocky shorelines of the northern Baltic Sea provide ample habitat for
several species of macroalgae, including *F. vesiculosus* (Kautsky & Kautsky 2000, Torn et al., 2006).
Previous studies have documented that certain macroalgae contain high levels of iodine (Ar Gall et
al., 2004), of which the kelp *Laminaria digitata* stores the highest amount (Ar Gall et al., 2004;
Küpper et al., 1998).

However recent chamber experiments comparing different species of brown algae
found that emission rate of $I_2$ was higher in the case of *F. vesiculosus* when compared to other species
like *L. digitata* (Huang et al., 2013). This could possibly explain the high IA concentration recorded
by the CI-APi-TOF when the air mass was coming from the Northern Baltic Sea region (11 August
2019 and 14 August 2019). High production of macroalgal species is common along the extensive
archipelago coastlines of the northern Baltic Sea, and particularly *F. vesiculosus* is likely to contribute
with high emission rates, especially when during peak production times when exposed to low sea-
levels and direct sunlight. However, partitioning the influence of macroalgae requires further
mechanistic studies. We conclude that marine and coastal regions surrounding the measurement site
are capable of producing SA and IA during bloom period, which can initiate NPF.

**4 Conclusions**
We studied the composition, concentrations and sources of precursor vapors forming aerosols in
Helsinki, Finland during the summer of 2019. The source of precursor gases causing new particle
formations were assessed by analyzing the meteorological parameters, situation of cyanobacterial
bloom in the Baltic Sea. Our study recorded several regional, local and burst events and we found
that they were connected to elevated concentrations of SA and IA. The burst /spike events occurred
simultaneously with high intensity cyanobacterial blooms in the Baltic Sea.

The study draws the following conclusions. 1) Constantly changing algal conditions in
Gulf of Bothnia, Gulf of Finland and Baltic Sea are a significant source for the emission of iodine
precursors and DMS. These emission further oxidize in the atmosphere to form IA and SA, which
can be detected by mass spectrometric methods. Interestingly, during marine air mass intrusion with
higher residence time over the algal blooms, the gaseous precursors formed from the biological
emissions possibly exceeded the gaseous precursors sourced from anthropogenic emissions at the



measurement site. In fact, an overall higher impact of biogenic emissions was noted in this semi-
urban site. 2) Moreover, the meteorological conditions like wind direction (biogenic and
anthropogenic source sectors) and wind speed were identified as the most important parameters
influencing the precursor vapor concentration reaching the measurement site and thus determining if
NPF occurred. These factors will become more important if the measurement site is distant from the
coast. Our study infers, that when the air mass travelled over the land with higher residence time of
the air mass over the urban areas, it was enriched with SA and organics from proximal-local sources
leading to the occurrence of regional and local events (30 June 2019 and 30 July 2019). In contrast,
when the air mass travelled over the water bodies, with higher residence times over the cyanobacterial
blooms, the air mass was enriched with biogenic IA and/or SA initiating a burst/spike event at the
measurement site (11, 14, 15 August 2019). This observation is comparable to other coastal sites like
Mace Head, although the NPF events are much stronger in Mace Head, since the measurement site is
just at the coast with intensive low tide high tide periods. 3) The formation rates of 1.5 nm particle
and ions suggest that both IA-driven and SA-driven NPF events were neutral nucleation events. 4)
The type of phytoplankton species, intensity of the bloom and distance of the bloom from the
experimental site plays a very important role in determining the concentrations of precursor gases
and thus influence the duration and type of NPF. The IA driven nucleation occurred when the air
mass travelled from over the Baltic Sea region, where the coasts are dominated by several species of
macroalgae, including *F. vesiculosus.* The SA rich burst events occurred when air mass travelled over
the Gulf of Bothnia which was mainly dominated by the cyanobacteria species *Aphanizomenon* 5)
Burst/spike events, connected to high IA concentrations, likely led to fast growth of particles
potentially to CCN sizes. The role of stabilizing the IA clusters by SA and ammonia in a semi-urban
coastal place needs to be further explored. The growth rate of particles was not fully explained by the
SA, IA and MSA alone, this applies especially for 3–7 nm or larger particles, indicating that organics
might be playing a critical role in the growth of particles in this semi-urban location. We have
significantly high ambient concentrations of HOMs in this study, although the detailed descriptions
is beyond the scope of this work.
The role of organics (HOM) in the growth of particles is an active research question.
Exploring the sources and characterizing them during a bloom period, when the emission of biogenic
volatile organics increase with temperature, is crucial to understand the climate linkages of aerosol
formation. Resolving these links require more quantitative studies linking of the quality and quantity
of cyanobacterial blooms to the strength of emissions and to production of aerosol precursors. More
studies partitioning the influence of pelagic cyanobacterial blooms and influence of coastal
macroalgae on new particle formations would need to be undertaken.




*Data availability*

Mass spectrometer and air ion spectrometer data related to this article are available upon request to the corresponding author. Rest of the data are available for download from https://avaa.tdata.fi/web/smart/smear.

*Supplement*

The supplement related to this article is available online at:

*Author contributions*

RCT and TJ, MS designed the experiment, MS, LB, NS, YJT, TC, YJ, JL, ML were involved in the instrument installations and performed calibrations, RCT, collected, processed, analyzed and interpreted the mass spectrometric data. TC, JS, JL, RCT and ML collected and processed the particle data. RCT, LD and KL interpreted the particle data. LD, LB, LQ and XCH preformed the calculations. MS, RCT, TJ and MK conceptualized the idea of connecting marine biology and atmospheric processes. AN improvised the marine biology section of the paper. CX carried out Flexpart analysis. MM contributed to the satellite data procurement and its interpretation. All authors contributed commented on the manuscript and improvised the data interpretation.

*Acknowledgements*

We thank the ACTRIS CiGAS-UHEL calibration center for providing facility for CI-APi-TOF calibration and INAR technical staff for support during the entire experiment. We acknowledge Finnish Meteorological Institute for providing open access to oceanographic data used in this study. Financial support: This work was supported by the European Research Council (ERC) under the European Union's Horizon 2020 research and innovation programme (GASPARCON, grant agreement no. 714621) and by the Finnish Academy (grant agreement no. 334514). We also acknowledge Jane and Aatos Erkko Foundation, ERC ATM-GTP, Flagship ACCC and Aerosols, clouds and trace gases infrastructure (ACTRIS) for funding support.

*Competing Interests:* The authors declare that there are no conflict of interests

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





Table 1:  Timing and maximum concentration of SA, MSA and IA during local and burst/spike
nucleation events during the study period

| Dates | Type of Event | time of NPF (UTC+02:00 h) | SA (max) molec. cm$^{-3}$ | MSA (max) molec. cm$^{-3}$ | IA(max) molec. cm$^{-3}$ |
|---|---|---|---|---|---|
| 30.06.2019 | Regional/ local | 8:45-13:23 14:00-16:30 | $7.9 \times 10^6$ | $5.6 \times 10^5$ | $2.3 \times 10^6$ |
| 30.07.2019 | Regional/ local | 7:45 -11:16 | $1.2 \times 10^7$ | $1.2 \times 10^6$ | $5.3 \times 10^6$ |
| 11.08.2019 | Ion Burst (Spike) | 13:40-14:32 | $1.0 \times 10^7$ | $1 \times 10^6$ | $3.2 \times 10^7$ |
| 14.08.2019 | Ion Burst (Spikes) | 8:00-8:20 | $4.2 \times 10^6$ | $5.3 \times 10^5$ | $8.5 \times 10^6$ |
| 15.08.2019 | Multiple Ion Bursts (Spikes) | 6:00, 8:58, 14:00-16:00 | $6.4 \times 10^6$ $6.3 \times 10^6$ $7.0 \times 10^6$ | $5.8 \times 10^5$ $4.6 \times 10^5$ $6.8 \times 10^5$ | $2.5 \times 10^6$ $3.1 \times 10^6$ $1.5 \times 10^6$ |




















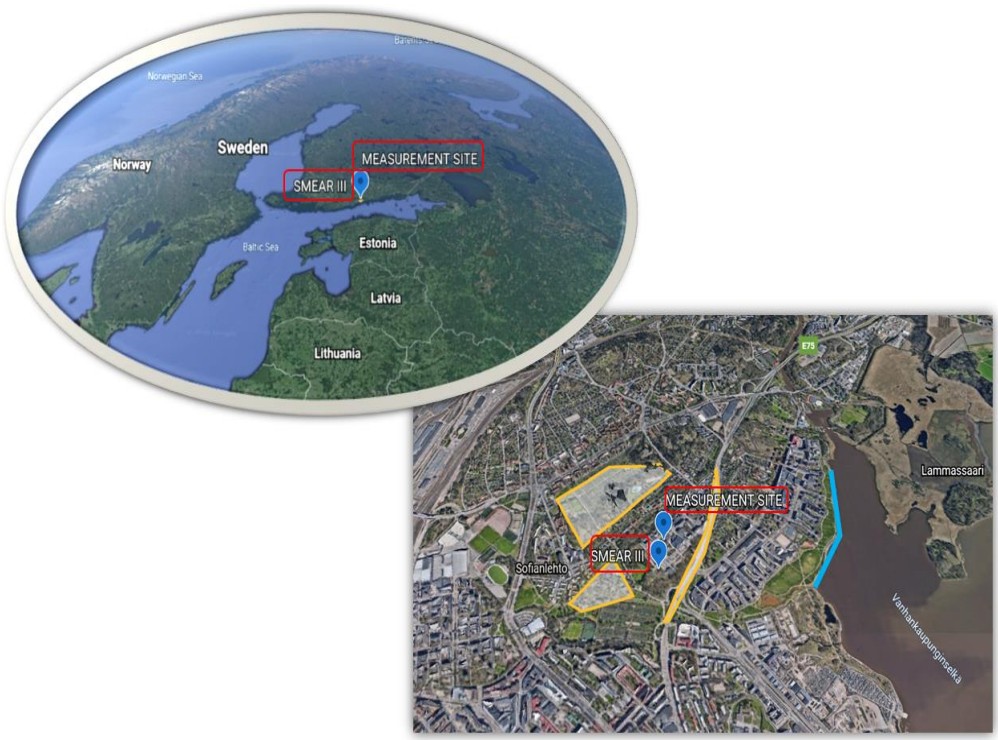


**Figure 1:** Map showing the two locations included in the study where instruments were operated (upper left panel). The yellow polygons on the left side of the measurement locations (on the lower right panel) shows forest/park with little or no traffic (West and Northwest, 300 m from the measurement site). The yellow double lines on the right of the measurement locations is the traffic area or the main road (E75) leading to the Helsinki city center (250 m east of the measurement site). The blue lines depict the coastline after which the lakes and coastal waters of Gulf of Finland start (1 km to the east from the measurement site) © Google Earth 2019.












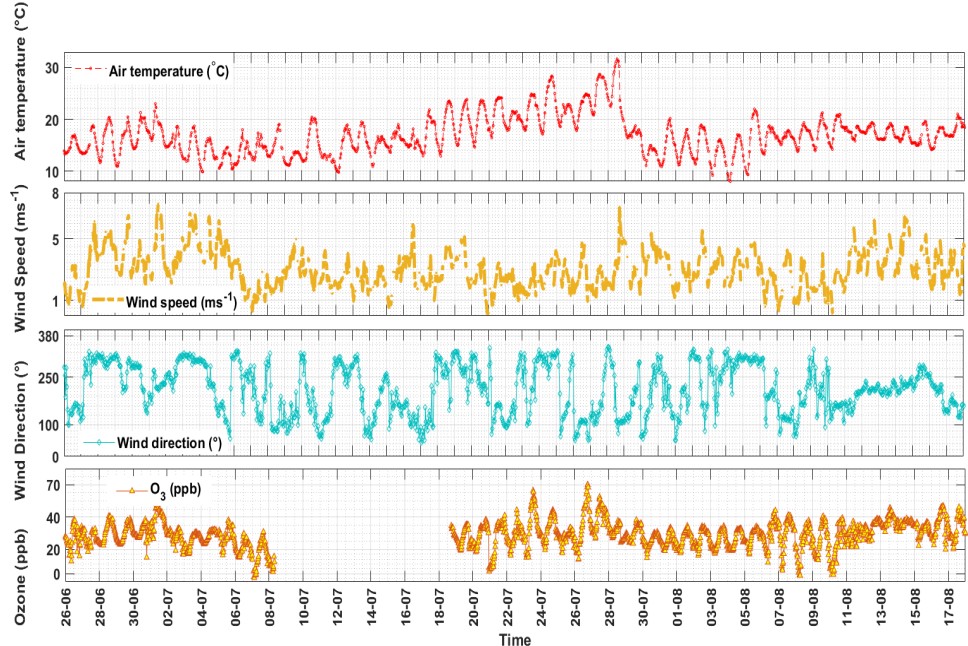



**Figure 2:** Time series of meteorological parameters and O₃ (data from SMEAR III station, 30-minute

averaged) during the study period.




















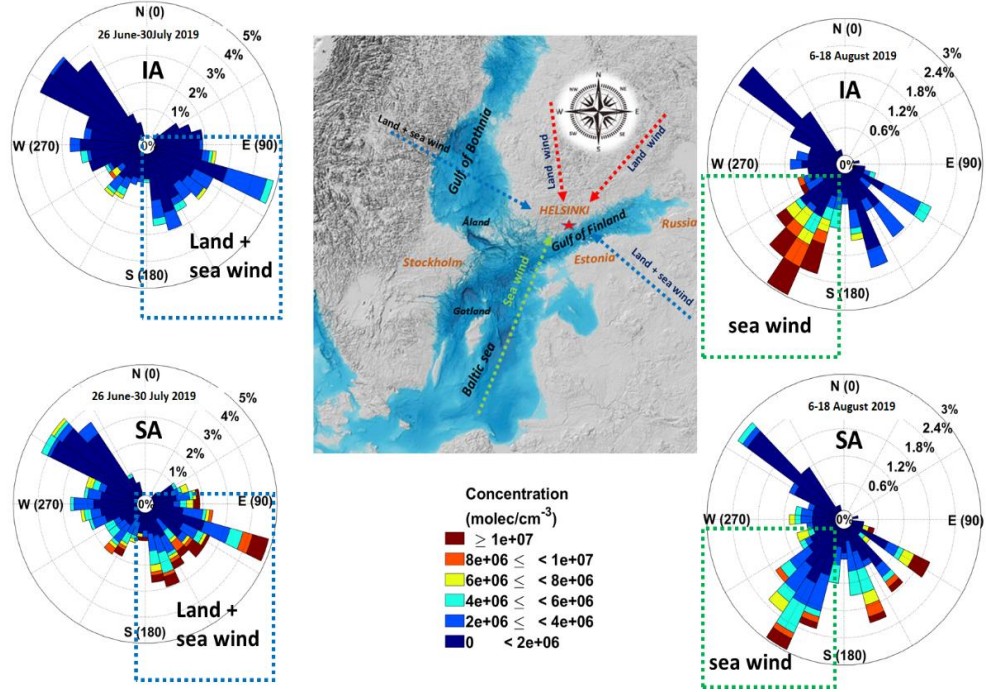



**Figure 3:** Windroses showing the variability in the concentration of gases with wind direction during
the study period. Percentages on the concentric circles denote the frequency of winds from different
directions. The spokes are color coded as per the concentration of the gas from the particular direction.
The numbers in the parenthesis within the windroses refer to the wind direction in degrees.











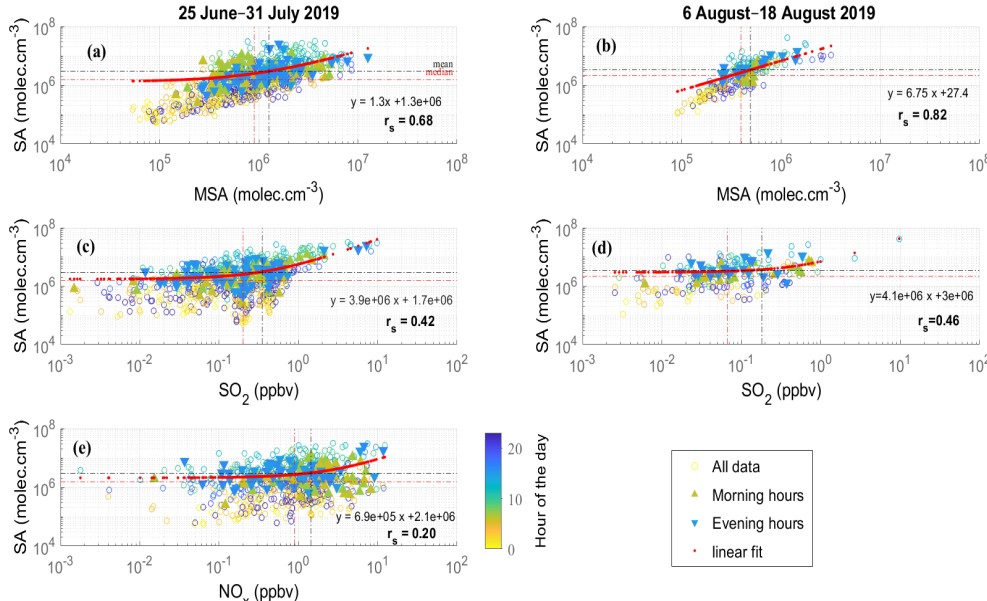



**Figure 4**: Correlation of SA with MSA (a,b), $SO_2$ (c,d) and $NO_x$ (e) for June–July. The black dashed lines for both axis represent the mean of the gas concentration, red dashed line represent the median value the gas concentrations and red solid line represents the linear fit. Spearmann's coefficient ($r_s$) was used to test the correlation, at significance level, 0.001. The circles represent data points at different hours of the day. The upward pointing green triangles represent the morning rush hours (6:00–8:00 h) and the downward pointing blue triangles represent the evening rush hours (15:00–17:00 h). The yellow hollow circles represent all data. $NO_x$ data unavailable of August.












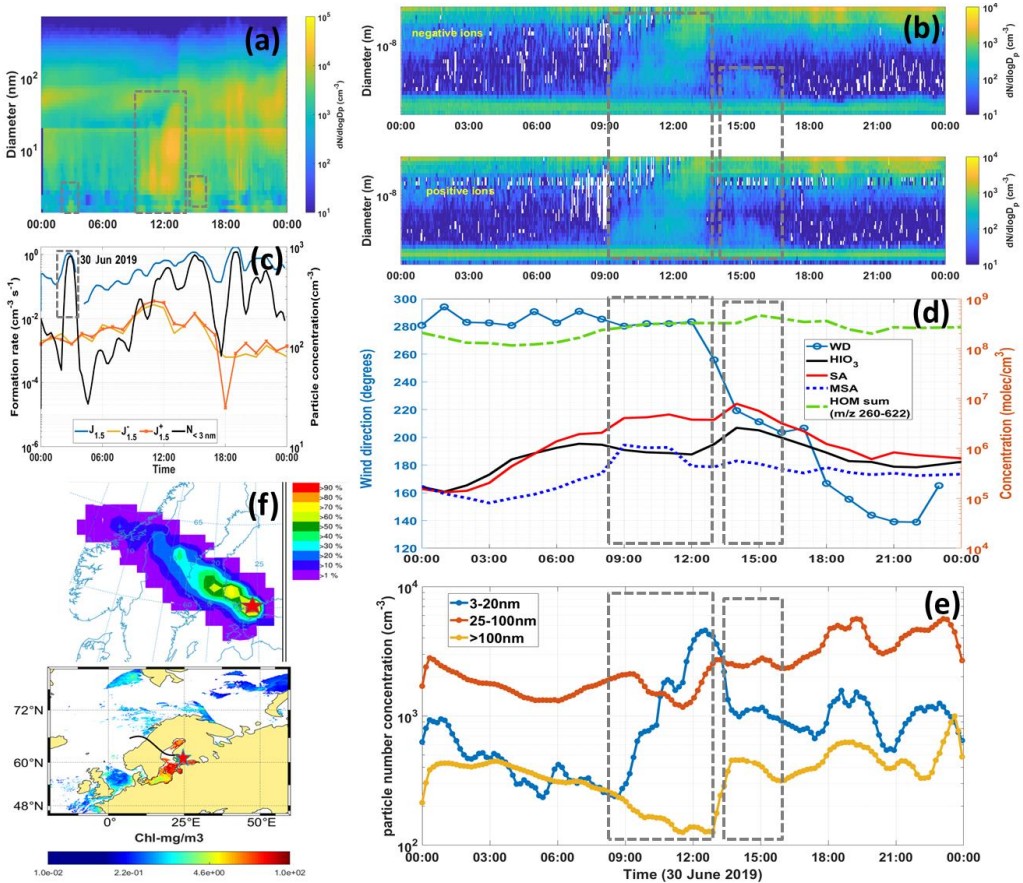



**Figure 5:** NPF Event (Regional and local events), 30 June 2019, the large dashed rectangle denotes the regional event, the small dashed rectangles show local cluster formation events. (a) Number size distribution of particles (data from PSM, NAIS and DMPS; size range: sub-3–100nm). (b) Charged particles number size distribution (negative: upper, positive: lower) obtained from the NAIS. (c) Diurnal variation of formation rates ($J_{1.5}$) of 1.5 nm particles and ions ($J^-_{1.5}$ and $J^+_{1.5}$) on the left axis and particle number concentrations (1.5–3 nm) on the right axis. (d) Diurnal variation of HOMs SA, IA and MSA with wind direction (WD). (e) The diurnal variation of particle concentration in nucleation:3–20 nm; aitken: 25–100 nm and accumulation: >100nm) mode particles during the event (Data from DMPS). (f) Trajectory frequency plot (100 a.g.l, arrival time of trajectories at the meaurement site: 20:00 h) for 24 h back trajectory using GDAS meterological input data (frequency grid resolution: $1.0° \times 1.0°$) and Chl-$a$ concentrations (MODIS); Black line shows the trajectory direction and the red-star point denotes the measurement site.







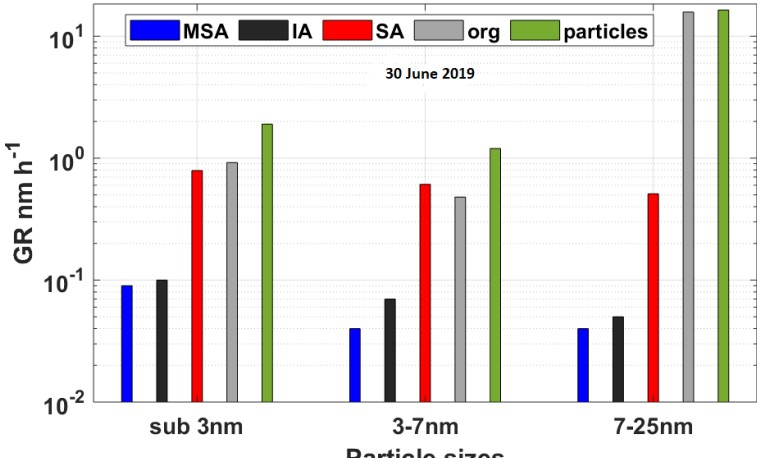



**Figure 6:** Particle growth rates calculated from the kinetic condensation of gases (data from CI-APi-
ToF) and the observed particle GRs (data from NAIS) in different size classes on 30 July 2019.




















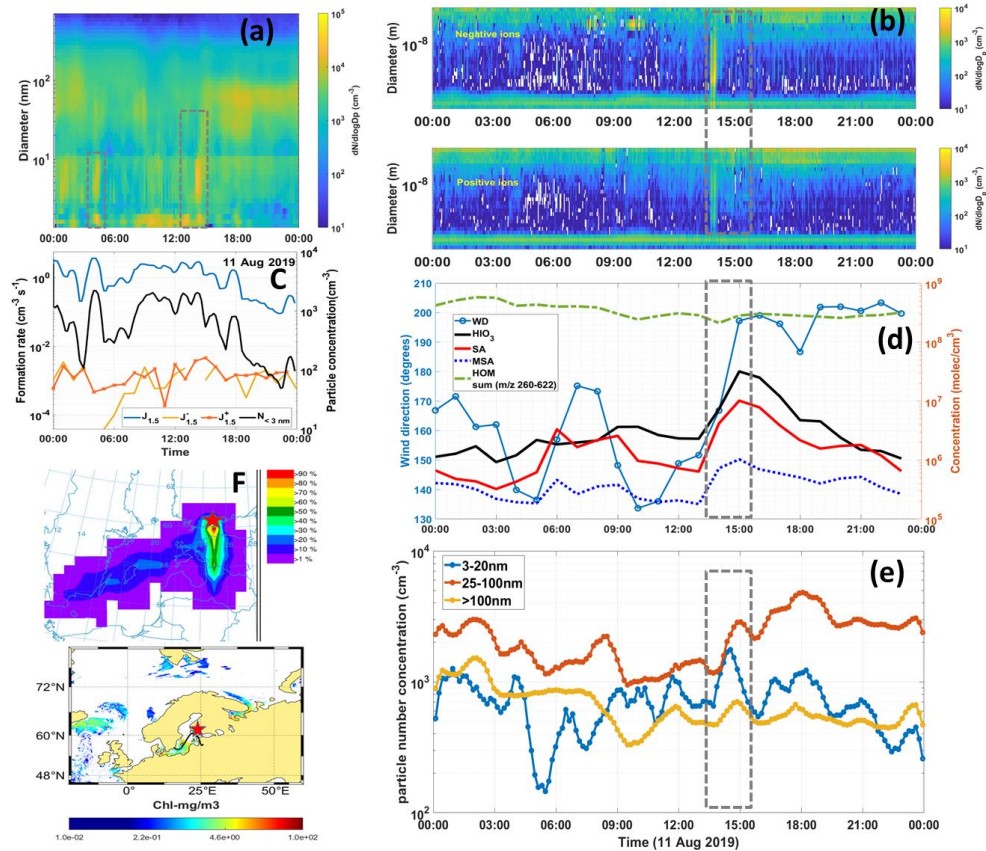


**Figure 7:** Burst/Event, 11 August 2019, The dashed grey rectangles denote the time stamp of the nucleation events. (a) Number size distribution of particles (data from PSM, NAIS and DMPS; size range: 1–100 nm). (b) Charged particles number size distribution (negative: upper, positive: lower) obtained from the NAIS. (c) Diurnal variation of formation rates ($J_{1.5}$) of 1.5 nm particles and ions ($J^-_{1.5}$ and $J^+_{1.5}$) and total number concentrations of particles (<3 nm, PSM). (d) Diurnal variation of HOMs, SA, IA and MSA with wind direction (WD). (e) The diurnal variation of particle concentration in nucleation (3–20 nm), Aitken (25–100 nm) and accumulation mode (>100 nm) particles (DMPS data). (f) Trajectory frequency plot (100 a.g.l, arrival time of trajectories at measurement site: 22:00 h) for 24 hour back trajectory using GDAS meteorological input data (frequency grid resolution: 1.0° × 1.0°) and Chl-*a* concentrations (MODIS); Black line shows the trajectory direction and the red star point denotes the measurement site.



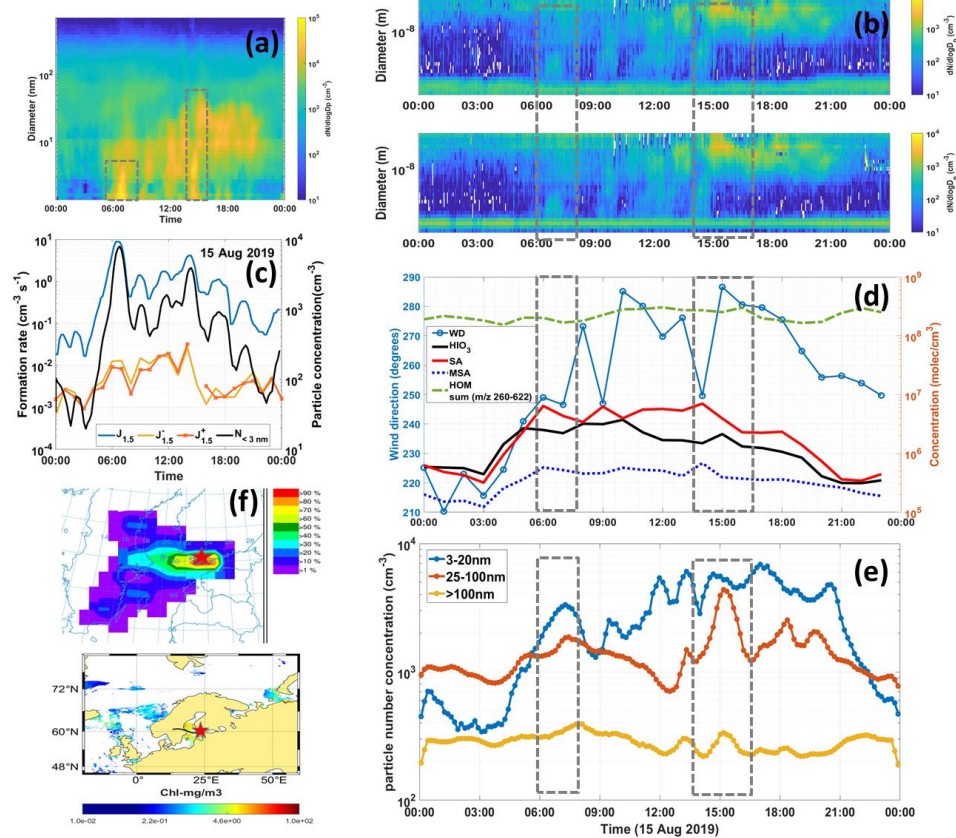



**Figure 8:** Multiple Burst/Spikes, 15 August 2019, The dashed grey rectangles denote the time stamp
of the nucleation events. (a) Number size distribution of particles (data from PSM, NAIS and DMPS;
size range: 1–100nm). (b) Charged particles number size distribution (negative: upper, positive:
lower) obtained from the NAIS. (c) Diurnal variation of formation rates ($J_{1.5}$) of 1.5 nm particles and
ions ($J^-_{1.5}$ and $J^+_{1.5}$) and total number concentrations of particles (<3 nm, PSM). (d) Diurnal variation
of HOMs SA, IA and MSA with wind direction (WD). (e) The diurnal variation of particle
concentration in nucleation (3–20 nm), Aitken (25–100 nm) and accumulation mode (>100 nm)
particles (DMPS data).(f) Trajectory frequency plot (100 a.g.l, arrival time of trajectories at the
measurement site: 22:00 h) for 24 h back trajectory using GDAS meterological input data (frequency
grid resolution: 1.0° × 1.0°) and Chl-*a* concentrations (MODIS); Black line shows the trajectory
direction and the red-star point denotes the measurement site.
