# Peer review of "An evaluation of new particle formation events in Helsinki during a Baltic Sea cyanobacterial"

_Atmospheric Chemistry and Physics, 2021_

## Author Comment (AC1)

**REVIEWER 1**

General Comments

This paper by Thakur et al. explores new particle formation events in Helsinki from gaseous **precursors of marine (iodic acid, sulfuric acid) and anthropogenic (sulfuric acid) origin. Importantly, this study highlights the complexity of nucleation in a semi-urban location with marine and anthropogenic influence. The authors use a wide range of ground-based instruments to monitor particle size and concentration, in addition to measurements of key gas-phase species. These measurements are paired with meteorological and satellite observations to identify the source of the precursors to NPF. This study fills a measurement gap of nucleation events in coastal urban areas.**

**This paper has some interesting results that are valuable to the NPF community. With that, I find it suitable for publication in ACP. However, this paper would benefit from clearer explanations of how the conclusions were reached, or perhaps some softening of their conclusions. Furthermore, I believe this work could use some editing for clarity.**

We thank the reviewer for appreciating the work and providing the specific and very valuable comments which has considerably improved the quality and clarity of the manuscript. We have answered the queries/comments for each point as detailed below. The corrections would be incorporated in the revised manuscript including softening of the conclusions reached in this work.

Specific Comments

• **In the abstract (L49), the authors indicate that the type of phytoplankton species and the intensity of the bloom was one of the most important factors affecting aerosol precursor vapor concentrations (IA and SA). How was this conclusion reached, when the only measurements made in this study to link their gas and particle phase measurements to biological activity were satellite measurements of Chl-a, which does not differentiate between species?**

We agree with the reviewer, that we did not make any actual measurements of the algal species neither did we do any species identification for this study. But we speculate that the emissions from the type of phytoplankton species found in a particular area (area selected based on the trajectory of air mass) might be influencing the gas concentrations in the atmosphere. Our interpretations are based on the residence time of air masses in a particular marine region. We made the best possible estimations on

the species present in that region based on Baltic-wide monitoring of cyanobacterial blooms from previous studies mentioned in the MS. As per these studies (Knutson et al., 2016; Attard et al., 2019; Kownacka et al., 2020) results show that bloom composition is fairly consistent for different regions and seasons from year to year, which makes it possible for us to make close estimations of the species present during our study in a particular region (from where the airmass travels). Importantly, however, the bloom composition during summer is different from spring time blooms, which we detail in our study and helps us interpret particle formation and their potential sources.

Accordingly we have added /modified the statements in the section 3.4

"The Gulfs of Bothnia and Riga are dominated by the genus *Aphanizomenon* (Kownacka et al., 2020). In addition, the Bothnian Sea and Gulf of Finland were found to be rich in cyanobacterial genera of *Aphanizomenon* along with *Nodularia* and *Dolichospermum* (Kownacka et al., 2020). As per the previous studies which carried out the Baltic-wide monitoring (Kowancka et al., 2020 and the references mentioned therein) that bloom composition is fairly consistent for different regions and seasons from year to year, which makes it possible for us to make close estimations of the species present during our study in a particular region (from where the airmass travels and the residence time over a particular region).

- **The authors also conclude on L696 that the type of phytoplankton species, bloom intensity, and distance from the bloom plays an important role.**

    1. **How does the phytoplankton species affect the gas-phase concentration in their measurements?**

        Not all plankton species emit DMS (a precursor for biogenic SA). There are only very few specific species found in some particular areas that may be relevant to account for regarding their contribution to biogenic SA in the atmosphere. Similarly, specific species of macroalgae are responsible for large emissions of $I_2$ which finally oxidizes to IA either at the source or during their transport to the study site. A detailed explanation of the species and their niche is explained in section 3.2.

    2. **Is there an instance where there was a sea wind with less intense phytoplankton bloom, and no NPF events?**

        Yes, we observed a few days when this occurred; for example, on August 13 and August 17 when there was no event, yet the there was a sea wind and the bloom was less intense as

compared to other event days. An example of such a day (August 17) is shown in the figure S13. The figure and this required explanation is incorporated in the supplementary information as reference.

[Figure]

**Figure S13:** No event day on 17 August 2019 (a) Chl-*a* concentrations (MODIS), the red star shows the experimental site (b) Trajectory frequency plot (100 a.g.l) for 24 h back trajectory using GDAS meterological input data (frequency grid resolution: 1.0° × 1.0°) (c) Charged particles number size distribution (negative: upper, positive: lower) obtained from the NAIS.

- **I'm also not convinced by the importance of the cyanobacterial blooms on the IA concentration, especially when compared to the other algae and marine sources. The authors timed their study to match with the cyanobacterial blooms that are expected in the Baltic Sea and coastal regions of Finland. In section 3.1.2 however, the authors emphasized that the cyanobacterial blooms were reduced below normal in July and August, which were the time periods in which they**

**observed the NPF events. The authors also point out that the low tide and high irradiance could be a source of macroalgae iodine, as was observed in McFiggans et al., 2010.**

- In section 3.3.2 (Case 1) we propose that the contribution of macroalgae to IA could be the dominant source when IA is speculated to play a dominant role in initiating the burst event on 11 August. However, we have based our conclusions only on the high Chl-*a* values in the region from where the air masses originated and where the air mass residence time was the highest. This section does not talk about the dominant contribution of IA from the Cyanobacterial blooms.

- In section 3.3.2 (Case 2): However, we speculate that when the wind direction (coinciding with high residence times) was over the bloom areas which are dominated by specific cyanobacteria producing DMS (a precursor to SA), we see an increase in SA followed by a burst event. The interpretation regarding particular species of cyanobacteria in the respective marine and coastal areas is based on ongoing yearly cyanobacteria monitoring coordinated by the Finnish Environment Institute (and cited in the text). No species identification was done for this study. However as stated above the bloom composition is relatively stable for particular seasons and areas between years, but their intensity may vary depending upon temperature and nutrient availability.

- Section 3.1.2: It is correct that cyanobacterial/algal blooms were less intense at end of July and August. And it is worth noting that when the blooms of both cyanobacteria and indeed macroalgae start to decay and die (while being exposed to sunlight) they produce more emissions (biogenic SA and IA). Thus, this itself is a reason for speculating why we see most of the NPF events in later summer months. All these observations suggest that there could be a strong link between algal/cyanobacterial emissions and their impact on NPF. However, as stated in our conclusions further studies are definitely needed to confirm these findings in a coastal setting.

  In order to clarify this point we have included the following lines in the conclusion section of the study:

  "In fact, an overall higher impact of biogenic emissions was noted in this semi-urban site particularly during end of July and mid-August when the bloom intensity decreases and the cyanobacteria/macroalage start to decay and die, while being exposed to sunlight, they produce more biogenic emissions biogenic of SA and IA".

**Would Chl-a measurements also measure the contribution from macroalgae?**

Yes the satellite Chl-a measurements can indicate if the contribution is from Macroalgae in case of high values (higher than the average) of Chl-a. This might indicate that there are floating/exposed macroalgae present. However, typically these algae are not present in the open sea areas of the Baltic Sea and the resolution of the satellite Chl-a are not in such high resolution as to permit an interpretation (or differentiation) with higher confidence, hence it's not mentioned in the MS. Nevertheless, the Baltic Sea has a great abundance of macroalgae along its coasts that we speculate could be contributing to the IA signal.

- **L485: The authors indicate that the change in wind direction 'apparently discontinued the precursor vapor source', however I'm not sure why this is apparent? From Figure 5(d), the concentrations of SA, MSA and IA remain relatively constant with the change in wind direction.**

  We thank the reviewer for pointing this misinterpretation in the manuscript. We accept that the changed air-mass just discontinued the growth and not the precursor vapor concentrations. We have clarified this and changed the lines to

  "This shows the particles must be the process of growth mostly elsewhere, which is not evident in the changed air mass, however we still observe almost the same (or even slightly higher) precursor vapor concentrations, since the wind still passed over the bloom areas before entering our study site".

- **Figure 6: Is the green trace called 'particles' the measured particles? Perhaps make that more clear.**

  The figure caption states that these are observed particles (measured through NAIS). However, we have changed the Legend to "measured particles (NAIS)" to make it clearer.

- **L593: How do you know all the I2 was oxidized to IA?**

  The reviewer has correctly pointed out that not all $I_2$ is oxidized to IA. Also, in the present study we cannot give an estimate that how much $I_2$ (from source region) would be converted to IA. Hence the lines 660-661 have been changed to

  "By the time the air mass reached our measurement site from the emission source, a fraction of the emitted $I_2$ could have oxidized to IA".

- **The authors often use parentheses to provide additional details within the text. In some cases, the parentheses are unnecessary and interrupt the flow of the text. I suggest the authors review their use of parentheses for clarity. Some examples:**

    1. **L32: Several studies have investigated New Particle Formation (NPF) events from various sites ranging from pristine locations, including (boreal) forest sites to urban areas.There have been studies of more than just boreal forests, I'm not sure why boreal was specified here. Can remove the parentheses and/or the word boreal.:**

    The parentheses has been removed as per the reviewer's suggestion.

    2. **L101: The parentheses around 'produced from macroalgae' can be removed**.: The parentheses has been removed as per the reviewer's suggestion.
    3. **L499: Can be rewritten as 'The high normalized signals…' to remove the parentheses.:** The parentheses has been removed as per the reviewer's suggestion.
    4. **L355: Can use 'The daily mean' instead of The mean (whole day):** We corrected the statement throughout the MS as per the reviewer's suggestion.

**Technical Corrections:** *All the technical corrections have been incorporated in the revised MS*

**L42: Keep the chemical names in lowercase "sulfuric acid (SA)" to match L78**

- It has been corrected in the MS as suggested by the reviewer.

**L46: Chemical names in lowercase "iodic acid (IA)"**

- It has been corrected in the MS as suggested by the reviewer.

**L150: Use the abbreviation for New Particle Formation (NPF)**

- It has been corrected in MS as suggested by the reviewer.

**L196: I'm not sure what 'mlpm' is, define it?**

- mlpm stands for milliliter per minutes, we have corrected these unit to "mLpm" in the revised MS

**L205: Define HOMs when it is first used**

"Please note that the concentration of highly oxygenated molecules (HOM, monomers and dimers) were calculated from the unit mass resolution data".

**L208: extra 'The'. Don't need to define UMR if you only use it once.**

We have made the correction in the MS as mentioned in the sentence above.

**L263: Don't need to redefine growth rate.**

- It has been corrected in MS as suggested by the reviewer.

**Table S1: $O_3$ instead of $O_2$?**

Yes we mean $O_3$, thanks for pointing out this typo error. We have corrected it now to $O_3$.

**447: Missing a period?**

Period added now in the revised MS. Thanks for pointing the error.

**556: Replace $HIO_3$ with IA**

We replaced HIO3 with "IA".

**Figure 4 caption: Not sure what the yellow circles are for 'all time' – is it just the other time except the morning and evening?**

Yes that is correct, that yellow circles denote "all time". It is mentioned in the Figure caption also, as underlined below.

"Figure 4: Correlation of SA with MSA (a,b), $SO_2$ (c,d) and $NO_x$ (e) for June–July. The black dashed lines for both axis represent the mean of the gas concentration, red dashed line represent the median value the gas concentrations and red solid line represents the linear fit. Spearmann's coefficient ($r_s$) was used to test the correlation, at significance level, 0.001. The circles represent data points at different hours of the day. The upward pointing green triangles represent the morning rush hours (6:00–8:00 h) and the downward pointing blue triangles represent the evening rush hours (15:00–17:00 h). The yellow hollow circles represent all data. $NO_x$ data unavailable of August"

---

## Author Comment (AC2)

**REVIEWER 2**

**Thakur et al. present field measurements of new particle formation events occurring in Helsinki. This site is impacted by air masses from the city and sea. Observations from Helsinki help fill in a critical gap in understanding how marine new particle formation impacts urban air quality. Their observations relate nearby algal and cyanobacteria blooms to marine new particle formation events.**

**Overall, the information presented in the paper is logical but some of the conclusions on which precursors contributed to which nucleation events are not persuasive. The paper is not written as concisely as it could be with many parts repeated and it's difficult to follow. Only some of these instances have been pointed out here. The authors should try to shorten the paper. The study fits well in ACP. Several aspects of the manuscript should be improved prior to acceptance for publication**.

We thank the reviewer for providing the specific and very valuable comments which has considerably improved the quality and clarity of the manuscript. We have answered the queries/comments for each point as detailed below. The corrections has been be incorporated in the revised manuscript. We would try and shorten the MS in the revised version as much as possible, basically modifying statements by replacing them with "short and precise lines" for a better flow and readability of the MS. However we might not be able to considerably shorten the MS, since we may not remove any sections or sub-sections from MS, since we feel it may distort the connectivity of the text. We promise the reviewers that we would do our best possible in this regard.

Also please consider that the number of pages may increase (or perhaps remain the same) in the revised MS since we revised the figures as per reviewer's suggestion (produced stacked time line figures) for each event and then making a separate figure for the trajectory+Chla for each event for clarity.

Major Comments:

**Line 124: Why are more coastal measurements needed? The authors detail out a few studies conducted at the coasts and where they found correlations to coastal seaweed and algal blooms. What does this study add to the scientific field other than more measurements? How do measurements from Helsinki help the scientific field? I am sure these measurements are important but framing the "why" will help the reader better understand the purpose of this study.**

We thank the reviewer for highlighting this point. We have now included a new paragraph and also modified the existing write up in the "Introduction", which is as follows:

Lines 78-90 (new addition) "The measurements of gaseous precursors, meteorology and biogenic influences are important to study the coastal NPF, which may lead to the formation of coastal/marine clouds. Coastal clouds are the drivers of many coastal ecosystem (Carbone et al 2013, Emery et al 2018, Lawson et al 2018). Any impact or fluctuations in the cloud formation may impact several other processes of the fragile coastal ecosystem. These coastal clouds demonstrate a high sensitivity to CCN (He et al., 2021) and they have a significant impact on the radiation budget because they have a high infrared emission and albedo when compared to the dark water bodies down below. In this study we highlight the type of NPF processes and their drivers in a semi-urban-coastal setting where the atmosphere could be a mixture of anthropogenic and biogenic emissions. Unlike the above mentioned previous studies which were mostly carried out in a perfect coastal environments where NPF would be most likely affected by the biogenic emissions (from macroalage), this study in a semi-urban coastal environment helps to evaluate the impact of urban processes Vs coastal emissions on NPF and at large the cloud formation processes.

Lines 137-179 (modified): "The limited NPF studies in the semi-urban coastal regions and the dynamic coastal meteorology drives the motivation of this research. Another motivation for this research is that till date no detailed studies on the impact of biogenic emissions on NPF events were done before in Finland despite the fact that extensive cyanobacteria blooms occur every year in the Baltic Sea region and neighboring water bodies (including Finnish lakes) (Kahru and Elmgren 2014), which could be a significant source of iodine species, SA and MSA. Increasing temperatures and the excessive nutrient load in the Baltic Sea promote algal growth (Kuosa et al., 2017; Suikkanen et al., 2007, 2013). According to HELCOM (Baltic Marine Environment Protection Commission), the Baltic Sea has warmed 0.3° C per decade, however after 1990 significantly faster at 0.6° C per

decade and in Finnish coastal areas the warming is even faster with a 2° C increase since 1990 (Humborg et al. 2019). The amount of blue-green algae (i.e. cyanobacteria) has shown a statistically significant increase in open sea areas in the Gulf of Finland, Sea of Åland and the Sea of Bothnia in the last 40 years (Kahru and Elmgren, 2014). The increase in frequency and intensity of cyanobacterial blooms would increase the potential emission of biogenic gases changing the composition of the overlying atmosphere and the atmosphere of the neighboring sites, depending on the meteorological conditions. In this semi-urban coastal setting the concentration of gaseous precursors and aerosol size distribution may be influenced by the local meteorological parameters such as wind direction, wind speed, (air mass) turbulences especially at the surface layer of the lower atmosphere. Coastal locations are dynamic environments with rapid changes in meteorological parameters, also making the study of NPF more challenging.

In this study, we aim at a thorough evaluation of aerosol precursor molecules with a detailed (NPF events) analysis during the cyanobacterial bloom period, in the coastal-city of Helsinki, Finland, from June to August (summer) 2019. This work evaluates the role of phytoplankton blooms and meteorological parameters in the NPF events observed during the measurement period. We also identify the major precursor vapor(s) and molecular clusters found during the aerosol events. Here, we formulate the hypothesis that gaseous precursors formed from the biogenic emissions from the surrounding marine areas could play an important role in the nucleation processes in Helsinki. Although Helsinki is a coastal area yet the role of marine emissions on NPF processes has not been studied before".

We hope that this is sufficient to bring out the real importance of the coastal measurements and again we thank the reviewer for helping us to make this research work better.

**Line 199: The authors state that the CIAPiToF was calibrated following the procedure detailed in (Kürten et al., 2012). That study only calibrated the CI-APi-ToF for sulfuric acid. How does this calibration constant apply to MSA, iodic acid, and organics with nitrate as the chemical ionization reagent ion? What is the systematic uncertainty associated with using this calibration constant for non-sulfuric acid molecules? Often the authors report 3 significant figures on their precursor concentrations. Is this in-line with their estimated uncertainty?**

The uncertainty range of the measured concentrations reported in this study is estimated to be $-50\%/+100\%$ and the limit of detection, LOD: $4 \cdot 10^4$ molecules $cm^{-3}$ (Jokinen et al., 2012).

HOMs and iodic acid have been estimated to be charged similarly at the kinetic limit as sulfuric acid (Ehn et al., 2014; Sipilä et al., 2016), so the calibration factor for them should be similar, but please note, that the concentration of other compounds than SA can be highly uncertain due to different ionizing efficiencies, sensitivities and other unknown uncertainties. If MSA, IA or HOMs do not ionize at the kinetic limit these concentrations could be underestimated and thus, the concentrations reported in here should be taken as low limit values (These statements have been included in the section 2.1, where we describe CI-APiToF).

Thus to be in line with uncertainty associated with using the same calibration factor for non-sulfuric acid molecules we have corrected the concentrations of all species to 1 significant figure. We thank the reviewer for pointing this out.

**Along these same lines, what was the holdover time of SA, MSA, and IA (and other compounds) in the CIAPiToF inlet? On line 599, the event lasted less than 30 minutes. These compounds are very sticky and likely persist in the sampling lines even if the sampling rate is high. They likely persist at different rates so the order at which each compounds reaches its maximum concentration (and its absolute concentration at the maximum) will vary. Have the authors examined this to better determine if short burst new particle formation events can actually be studied with this instrument setup? How would time dependent wall loss rate impact the calculation of growth rates?**

Hoping that the reviewer meant residence time of the species in the inlet tube, an estimation can be provided. Considering the inlet length of 1 m and ID of 17mm, volume of one meter is 0.23 litres, with 10lpm inlet flow rate residence time is ca. 1.4 sec per meter. In our study we used inlet design as described by Eisele and Tanner (1993) and Kurten et al. (2011) and further used by Jokinen et al., 2012. In this type of inlet (with a inlet flow of 10Lpm) the interaction time between the sample and reagent ions is approximately 200ms. For the bisulphate-DMA cluster the negative free energy is high enough to be detected at this residence time (Ortega et al., 2014). For SA which is detected as HSO4- ion in the CI-APiToF is reliable enough since the evaporating molecule/cluster could be H2SO4/DMA/NH3 leaving behind bisulphate ion to be detected as $HSO_4^-$ + (HNO3)HSO4- (Ortega et al., 2014).

The loss rate is proportional to the square root of the diffusivity for the different molecules (Crump and Seinfeld, 1981). Although we agree, that this instrument is not completely free of the wall losses. Since wall losses are dependent on the flow rate, tube length and the diffusivity of the molecule we

have corrected the final concentration for these losses by considering 50% loss for SA (concentrations corrected by a factor of 2 for 1m inlet length and 10 lpm flow rate).

We take into account the diffusion loses while calculating the calibration factor and this is already mentioned in the MS text (Section 2.1).

"The instrument was calibrated prior to the experiment according to (Kürten et al., 2012) resulting in a calibration factor of $1.45 \times 10^9$ molecule per normalized unit signal including the diffusion losses in the inlet line"

Therefore we do not expect that "time dependent wall losses" to be significant enough to affect the GRs given that other losses have been accounted for prior to estimating the final concentrations.

Wang et al., 2021 determined the the decay rates of $HIO_3$ are 400 s for the Br-MION-CIMS. If we consider this rough estimate for $NO_3^-$ CIMS (present study, where we use Eisele type inlet, Eisele and Tanner (1993)), with the assumption that MION inlet minimally differs from the CIMS inlet (differing basically in the ion source orifice) we can suggest that the residence times was less than the decay rates of the iodine species, hence the instrument can be trusted that it gives close to accurate concentration of these species during a burst event.

To minimize the wall loses of the even the extremely low volatile species the inlet of the CI-APiToF was designed to the use of coaxial sample and sheath flows in order to sample (extremely) low-volatile species which are easily lost to the walls (Riva et al., 2019). So as per the numerous other previous works who almost precisely quantified the ELVOCs through the same design scheme, the authors believe that SA, MSA and IA could be almost precisely quantified by this instrument and flow scheme. Further Sipilä et al., 2016 has quantified the iodine species during an intense burst event at Macehead using a nitrate ion. CI-APiToF (event lasted >1hr). The instrument without chemical ionization is also capable of detecting the initial ions in the burst- NPF (Junninen et al., 2016).

**Line 322: how long did the cyanobacteria bloom last during the measurement campaign? What area did it cover? In line 324, what does lower than normal mean? Lower than June? Some numbers would help put this intensity in perspective. In line 582, the authors comment that the blooms are intense but how does this compare to other periods of time. Is there a correlation of**

**bloom intensity with IA, MSA, and SA concentrations (assuming the air mass is coming from the bloom's direction)?**

As per the SYKE press release (2019) detailing results from the annual monitoring, the northern part of the Baltic Sea's main basin, entrance to the Gulf of Finland and south of the Åland Islands were enriched with blue-green algae (cyanobacteria).The bloom lasted from June-August 2019. In coastal areas, bloom was mostly spotted in the Archipelago Sea, Gulf of Finland, Bothnian Sea and the Quark. The bloom situation was highly variable in space, even over short distances. The fragmented nature of the coastal areas and changing wind and water currents makes the algal situation intensely dynamic.

The information/statement "Lower than normal" here is extracted from the SYKE press release 2019. It refers to the lower mean cyanobacterial biomass as compared to the previous years. We have provided some actual estimates of bloom intensity in the main text, which is as follows:

"However, the weather conditions in July began changing with high winds causing the cyanobacteria to be highly mixed in the water column, which reduced bloom intensity at the sea surface to lower than normal mean cyanobacterial biomass (mean biomass of cyanobacteria, 105 µg $L^{-1}$, Kownacka et al., 2020) by end of July and August (SYKE press release, 2019). However the average biomass of cyanobacteria in 2019 (196 µg $L^{-1}$, Kownacka et al., 2020) was slightly higher than the average."

No we did not find any good correlation (in terms of correlation coefficients) between the gaseous precursors and Bloom intensity mainly because of the following reasons:

1.      The trajectory distance covered by the air mass before entering the study site was quite large to accurately estimate the Chla concentrations along the path (without large uncertainties).
2.      The semi-urban setting of the experimental site may not allow us to accurately estimate the exact biogenic emissions from the source. Particularly if the source is situated in Baltic Sea, Gulf of Finland or Gulf of Bothnia (from where most of the trajectories passed before entering the study site.

For the above mentioned reasons, we opted to analyze the events, emissions and wind direction on a case-by-case basis, where we can provide more accurate estimations.

However, we agree with the reviewer's suggestion of correlating Chla with the precursors, which for this study may not possible. But based on this study and several other publications related to coastal NPF from our research group, we have secured funding for the next few years to establish a coastal

atmospheric research observatory in Finland in collaboration with Tvärminne Zoological Research station, Hanko. Through this collaborative research with the biologists/ecologists we will obtain in situ data of Chl-a to accurately correlate with our gaseous measurements. Further, there would be less interferences in the gaseous precursor's concentration from other sources, which are mainly found in an semi urban/urban setting. However, this current research would serve as the baseline study for this kind of future research in Finland (which we highlight in the discussion).

**Line 331 and paragraph beginning 633: Did the authors actually measure the algae and cyanobacteria types during the blooms that occurred during the measurement period? Bloom composition can easily change based on numerous factors so it may not be a fair conclusion to link these previously measurements bacteria and algae types to what was observed during the measurement campaign. How confident are the authors that they can link these algal species to their new particle formation events?**

No, we did not do any actual measurements of the algal and cyanobacterial blooms. The data is obtained only from the satellite measurements and national monitoring conducted the Finnish Environment Institute. As per the various papers mentioned in MS, it seems that Bloom composition largely remains the same if we consider the blooms that occur either in early or late summer and in different regions. Hence, the bloom composition during summer is different than the Spring time bloom (Kownacka et al., 2020).

We strongly speculate the link between the algal species and NPF events mainly because of the strong relationship of the changes in wind direction and the changes in the precursor vapors specifically IA. However we cannot be 100% sure for this speculation and as stated in the conclusions of the study we need more studies of coastal NPF near/around the coast of Finland to confirm our findings. We would soon start to measure the chemical composition of the NPF forming precursors right at the Hanko, Finland coast (Tvärminne Zoological Research) where the urban influences would be minimal and we can then quantify the algal emission more precisely. However, this baseline study is equally important against which we are have initiated further research in this domain.

**Line 471: From previous NPF campaigns, it seems that sulfuric acid concentration should increase before observed particle concentration? The text suggests the sulfuric acid concentration increased after particle concentration increased. Figure 5 shows that SA was already increasing. Did the authors observed any freshly formed clusters with the CIAPITOF? From line 499, it seems the authors did observe some clusters (and shown in Figure S4). It would**

**be helpful/more logical to mention this earlier. Did the authors measure DMA and ammonia concentrations? If the authors believe SA-DMA is forming neutral clusters, do the authors know where the DMA is originating from? Why does the SA+DMA cluster concentration peak significantly after the new particle formation event (figure S7)? On line 474, what is a local clustering event? Does this refer to clusters observed on the CIAPITOF?**

The reviewer is correct that the all the NPF studies mention that SA starts to increase preceding a NPF. Line 504 mentions "Subsequently, SA concentration doubles from $2 \times 10^6$ to $4 \times 10^6$ molec. $cm^{-3}$".

With this line we meant that SA concentration doubles at the start of NPF, however, it can be clearly seen form Figure 5 that the SA concentrations have been steadily increasing well before the NPF event starts. To avoid the confusion the sentence has been changed to "Preceeding the NPF event the SA concentrations were steadily increasing and Subsequently at 09:00hrs".

As per the reviewer comment we have mentioned the observation of clusters earlier in the text.

We have not made any separate measurements of $NH_3$ or DMA. Our CI-APiToF was not tuned to measure masses lower than 50 a.m.u. And unfortunately, no separate instrument was deployed to make such measurements during the campaign. However, we do report clusters of SA-DMA-SA in our manuscript as normalized signals, since we did not make any separate calibrations to quantify DMA clusters. Moreover the calculations of concentrations of DMA-SA clusters would lead to large uncertainities since they are prone to evaporation losses inside the CI-inlet (Sipilä et al., 2015).

There is currently no reported observation of DMA in Helsinki. Although previous studies have reported , the ambient air concentration of $NH_3$ ranged from 20 pptV to 830 pptV in the forest site (Hyytiäla) (Makkonen et al., 2014) and in the urban station (SMEAR III) from below 450 pptV to 3000 pptV (Makkonen et al., 2012). Dimethylamine concentration of 5 pptV has been shown to enhance atmospheric aerosol formation rate by more than a 1000-fold compared to an NH3 concentration of 250 pptV (Almeida et al., 2013). DMA inclusive of other main methylamines like mono and tri methylamines (Bergman et al., 2015) in the global inventory (Schade and Crutzen, 1995) is contributed through the animal husbandry and other agricultural practices, biomass burning and some contributions from marine and terrestrial sources. Although among these methylamine emissions from the above mentioned sources Trimethylamine dominates (Schade and Crutzen, 1995).Although there are no current estimates of DMA in the Helsinki region, but DMA estimates is available from the boreal forest region of Helsinki. The study of Hemmilä et al., 2018 states that the

median concentration of DMA in July in Hyytiälä region was below the detection limit of the instrument. In this study the amines were detected using an online ion chromatograph (instrument for Measuring AeRosols and Gases in Ambient Air – MARGA) connected to an electrospray ionization quadrupole mass spectrometer (MS), with the detection limit as 0.2–3.1 ng m$^{-3}$. Further Sipilä et al 2015 measured DMA concentrations through a $NO_3^-$ CIMS and found that DMA was below ∼ 150 ppqV in a boreal forest site. Their work also stated that DMA was unlikely the playing an important role in the nucleation process observed at the site.

Figure S7 : shows the increasing signals of the clusters for the event on 11$^{th}$ August through which we speculate this as IA driven event since preceding the burst events we only observe IA (and $IO_3^-$) to be increasing and not SA-DMA. That is why it made us speculate that IA might be playing a more important role here in terms of initiating the particle formation. SA-DMA increase when we see the growth of particles at 15 hrs, probably indicating their more dominant role in the growth of particles along with IA.

Local clustering event here means that the molecules could be transported from elsewhere but the actual clustering could have taken place near the experimental site before the inlet, since it's a very small bump of clusters (with absolutely no growth) seen in the NAIS. It does not mean the clustering happening in the CI-APiToF inlet.

**Line 520: Why does MSA and IA concentrations need to increase in order to demonstrate they could participate in that NPF event? Also, from Figure S5, the concentrations of both are increasing (before the signal cuts off). How do the authors know for certain that these compounds are not participating in the event? Also HOM concentrations do not seem any higher than those in Figure 5. So why is this event SA-HOMs driven? What clusters did the CIAPITOF see? The organic clusters shown in Figure S4 just show a constant and slow increase in organic cluster concentrations throughout the day.**

Yes, we agree with the reviewer that to "participate in the NPF event" the concentrations need not to increase, if they are already significant in concentration. Perhaps as the other NPF studies a significant increase in concentration of IA and MSA (may be comparable or more than SA) is necessary in order for them to "initiate" nucleation (Beck et al., 2021; He et al., 2021).

In the manuscript we have stated:

"Therefore, the precursor gases from the biogenic origin, IA and MSA do not show a significant concentration increase during this event and hence assumed to be contributing insignificantly to this event"

The lines simply meant that they might participate but their contribution in the NPF event may not be as significant as SA.

However, for clarity we have modified the lines to the following:

"Therefore, the precursor gases from the biogenic origin, IA and MSA do not show a significant concentration increase as compared to SA, during this event and hence their contribution towards the initiation of the NPF event may not be as significant as SA"

Case of 30 June 2019 (Figure S4): If we compare the increase of the normalized signals of organics during NPF on 30 June 2019 with that of 11 August 2019, we can see that the cps started to increase well before the NPF on 30 June 2019 and keeps on steadily increasing throughout. With this comparison we can suggest that on 30 June NPF event the organics played an important role. Since this is a highly speculated conclusion, we have modified the sentence as follows: "indicating a SA driven event –with a possible contribution of HOMs"

**Line 558: Isn't the temperature during this campaign much higher than 10C? Higher temperature will still favor SA+amine/ammonia nucleation. Observations of HNO3.IO3- and H2O.IO3- clusters doesn't indicate IA nucleated. Were larger IA clusters seen? Also did the authors calculate how much IA contributes to growth? Is that why the authors are implicitly linking high concentrations of IA to particle growth in line 564?**

Yes the temperature throughout the campaign was higher than 10°C.

The recent work of Xiao et al 2021 demonstrates that in the urban atmosphere NPF is mainly driven by the formation of sulfuric acid–base clusters, which are stabilized by the presence of amines, high ammonia concentrations and lower temperatures. Figure 1 from the work of Xiao et al., 2021 clearly shows that at an SA concentration of 10e6 molec.cm-3 or higher the nucleation rates (J1.7) was mostly > 1 cm3 s-1 at 293 K (magenta) in the presence of DMA (4pptv DMA injection).

[Figure]

Figure 1: Atmospheric nucleation rates( $J_{1.7}$ ) versus SA concentrations (Xiao et al., 2021).

Our study reports the atmospheric nucleation rates (J1.5) mostly below 1 cm3 s-1 at SA concentrations $10^6$ molec.cm-3 (Pls refer to Figure 5 in the MS). In these cases we can speculate that the NPF may be to some extent driven by SA-DMA (Also HOMs) system. However, in the NPF case on $11^{th}$ August 2019, where IA concentrations clearly increase ( $10^7$ *molec. cm$^{-3}$* ) over the SA concentrations (note the SA concentration remains similar to that observed during the event on 30 June 2019, i.e $10^6$ molec.cm-3), we see nucleation rates (J1.5) clearly increasing above 1 cm-3 s-1 (Figure 7).

As mentioned in the manuscript text (Lines 607-612) *"This was the highest observed IA concentration in the entire measurement period. A recent study by He et al., 2021, indicate that HIO$_3$ concentrations above $1 \times 10^7$ molec. cm$^{-3}$ leads to rapid new particle formation at +10° C. At such concentrations the efficacy of iodine oxoacids to form new particles exceeds that of the H$_2$SO$_4$-NH$_3$ system at the same acid concentrations. Thus, the concentration of IA found in this event is capable of initiating nucleation, especially since the concentration of IA being two times higher than SA during the start of the event",* we speculate that this was an IA-driven event given the amount of SA remains unchanged! Further, higher temperature in principle should reduce the formation rates for all systems (SA+amine/ SA+NH$_3$/Iodine oxoacids). However, SA-DMA nucleation is in general much faster than the other two systems (SA+NH$_3$/iodine oxoacids) and if significant amount of DMA is present the nucleation rates should be significantly higher than current values given that we do not have very high SA concentrations (>$3 \times 10^7$ molec. cm-3) in the study area. (Yao et al., 2018)

No larger clusters were seen in the CI-API-TOF measurements. Since to actually detect the large clusters the ambient IA concentrations should be close to or higher than $10^8$ (Sipilä et al., 2016)

We calculated the IA GR for the event 30 June and 30 July through the parameterization methods used in Nieminen et al., 2010. This method was also very recently used to calculate the IA growth rates in the recent work of Beck et al., 2021. However since it was an intense burst event with no proper growth (as seen in "banana" type events) we were not able to calculate the growth rate for this particular event. We speculate it to be an IA driven event due to high concentration of IA seen during the event capable of initiating the ion clustering (He et al., 2021), observation of increase of normalized signals of IA-clusters (Fig S7) and all the above reasons mentioned above why SA may not be the main precursor gas initiating the NPF.

**Line 567 (And figures 5,7,8): The authors comment that the Aitken mode particle concentration increase after a new particle formation event. Why does the concentration drop before+during a new particle formation event? No doubt the decrease in scavenging rates allows nucleation to occur but what is leading to this drop in large particle concentration? This seems just as important as an increase in precursor concentration in producing an event.**

We thank the reviewer for pointing this important observation in the manuscript. In all the cases (Fig 5, 7,8) there is a drop in both accumulation and Aitken accumulation mode particles before NPF. We also see a change in wind direction before start of NPF in all the three cases. Previous study of Vakeva et al., 2000 also suggests that wind direction changes led to decrease in particle concentrations and also change in particle size distributions particularly in urban areas and was considered as the most important factor-affecting particle concentrations. Also, if we carefully examine, during the NPF the accumulation mode particles showed an increase in all the cases, except the one out of two NPFs on 30 June 2019. This was because unlike the other two cases (11 Aug and 15 Aug) a stable wind was observed during NPF on 30 June. Also, its worth noting that cloudiness parameter also affected the nucleation on 11 August, since it was an overall cloudy day with few hours of clear sky conditions (which is already described in the manuscript). Hence, we can say that there is not one meteorological parameter affecting the start of NPF and determining whether the NPF would lead to growth of particles or not. And this observation is consistent with various other NPF studies.

Case 30th June: we included the lines regarding the change in particle concentration as follows:

"However, we also observe a drop in Aitken particles before which continues during NPF. We speculate it could be due to the change in wind direction (Vakeva et al., 2000) before NPF. The wind direction relatively remains constant throughout the NPF so the low concentration of Aitken mode continues. Wind Direction changes abruptly at 12:00h and the Aitken mode particle concentrations increases soon after this change of wind direction. (Fig. 5d)."

Considering this an important observation (and also as per the reviewer's helpful comment) we have included this a statement "Further we also infer that that the wind direction played an important role in determining the particle concentrations at the study site" in the conclusion section of this study.

Minor comments:

**Line 154-160: It would be less confusing if the instrument details in this paragraph moved to the instrument 2.1. Otherwise, the reader will want more details about the instruments before the actual instrument section.**

We accept the reviewer's suggestion and will incorporate the suggested changes in the revised MS.

**Line 170: the paper hypothesis has already been stated in introduction. No need to state it again here**.

This change would be incorporated in the revised MS.

**Paragraph beginning 169: It would be more useful if this paragraph focuses instead on presenting the date+times of the algae and cyanobacteria blooms during the measurement campaign. The background on why there are more blooms should be mentioned in the introduction instead to better motivate this study.**

These lines are incorporated in this section "As per the SYKE press realease (2019) the northern part of the Baltic Sea's main basin, entrance to the Gulf of Finland and south of the Åland Islands.were enriched with blue-green algae (cyanobacteria) .The bloom lasted from June-August 2019. In coastal areas, bloom was mostly spotted in the Archipelago Sea, Gulf of Finland, Bothnian Sea and the Quark. The bloom situation developed rapidly and spatially highly variable, even over short distances. The fragmented nature of the coastal areas and changing wind and water currents makes the algal bloom conditions highly dynamic". And more details on "why there are blooms" has been shifted to

Introduction section (Lines 144-156). The Introduction section has been described in detail in answer 1 of the reviewer's comment.

**Line 189: What type of inlet?**

We have added this text in the Lines 190-192:

"In our study we used inlet design as described by Eisele and Tanner (1993) and Kurten et al. (2011) and further used by Jokinen et al., 2012".

**Line 195: Was the only reagent ion NO3-? Or did it have ligands?**

$NO_3^-$ was the most abundant reagent ion. However its dimer $((HNO_3)NO_3^-)$ and trimer $((HNO3)2NO3-)$ is also found in the spectrum. All the concentrations are normalized against the most dominant reagent ions which are estimated as $NO_3^- + (HNO_3)NO_3^-$. The lines in the MS are modified as follows:

"SA, MSA, IA concentrations are calculated after normalizing them with the reagent ions ($NO_3^-$ and $(HNO_3)NO_3$) using the equation mentioned in Jokinen et al., 2012"

**Line 196: is mlpm milliters per min? The L should be capitalized to make it less confusing. Or ccm which is more commonly used? Or maybe mlpm is fine? But it was initially confusing to me.**

Yes mlpm= mililitres per minute. The units are now written as "mLpm" in the MS.

**Line 208: There is a random The at the end of the line**

It is deleted now. Thanks for pointing that out.

**Line 213: What do the authors mean by two identical DMAs? Have they quantified the transfer functions and transmission efficiencies for both to say they are identical?**

In principle the DMAs are built identical but in the instrument the applied voltages are of opposite sign. But we agree with the reviewer's comments that they are not perfectly identical since as per our knowledge all DMAs are individually characterized (by the manufacturer, in this case Airel) and the electrometers can have different sensitivities, so this results in each DMA having their own transfer function after calibration.

So the statement about identical measurement columns differing in polarity is a statement about principle of operation, not a result that was obtained from calibration.

Therefore we have modified the statement in the revised MS (section 2.1) as follows:

"NAIS consists of two multichannel electrical mobility analyzer columns (DMA's) operating in parallel. The columns differ by the polarity of the ions measured, but are otherwise identical (Mirme and Mirme, 2013) in operation. However they may differ in the transfer functions after calibration.The calibration procedure for the DMAs is presented in Mirme and Mirme, 2013. The ion mode measurements are corrected as in Wagner et al.,2016)".

**Line 280: Are these mobilities diameters?**

Yes it means the mobility diameter and the term "mobility" has been incorporated.

"The formation rate of the total particles of mobility diameter 1.5 nm is calculated using the time derivative of the particle number concentration measured using the PSM in the size range 1.5– 3 nm."

**Section 2.5: Condensation sink spans what particle diameters? (I think it's >6 nm). Is there a reason why CS does not include surface area of smaller particles which could be significant during a new particle formation event?**

We thank the reviewer for their comment, we calculated the condensation sink using the equation mentioned in (*Pirjola et al., 1999*) which uses the Kn (Knudsen number), which is also dependent on the diffusion coefficient and the average absolute velocity of the vapor molecules (Hirshfelder et al., 1954). The equation used for this calculation assumes that condensable vapor does not take part in nucleation (Maso et al., 2002). So assuming that the vapors less than 3nm size (measured by NAIS) mainly participate in nucleation and the condensable vapor under study which is depleted via condensation onto the existing particles, were mainly the particles outside the size range of nucleating particles (>6nm and above), we decided to calculate the condensation sink using DMPS data.

Further to check, whether particles less than 6nm act as condensation sink or not, we calculated the condensation sink from the DMPS system which includes the diameters <6 nm, and we now include this information in our supplementary information of the manuscript as per your suggestion. The results are shown in figure below:

[Figure]

Figure S1 NPF Event- 30 June 2019 (a) Number size distribution of particles (data from PSM, NAIS and DMPS; size range: sub-3–100nm) (b) Particle Size dependent condensation sink variability during the most intense hour (12:00 h) of the event.

The Fig. S1 clearly shows that the size distribution of the CS for an example day (30 June 2019) with a strong nucleation event and a selection during the most intense hour (12:00h). We find that most of the CS is concentrated in the Aitken and accumulation mode rather than the nucleation mode.

Although the concentration of the smallest particles is substantially higher during an NPF event, we find that nucleation mode particles do not provide enough surface area to compete with the larger particles in terms of condensation.

This figure has been included in the supplementary information as Fig. S1 and the other figure numbers both in caption and text has been changed accordingly.

Also the text in section 2.5 has been changed as below:

"The condensation sink (CS) plays an important role in understanding aerosol dynamics. This parameter determines how fast gas molecules will condense on the pre-existing particles (Dal Maso et al., 2002; Kulmala et al., 2005, 2012). In this study, CS has been calculated by using the DMPS data (>6nm particles), according to Pirjola et al., 1999. Further to check, whether particles less than 6nm act as condensation sink or not, we calculated the condensation sink for one nucleation day (30 June 2019) from the DMPS system which includes the diameters <6 nm and found that most of the CS is concentrated in the Aitken and accumulation mode rather than nucleation mode (Fig. S1)".

Also many other NPF studies use DMPS/SMPS (Aalto et al., 2001: Maso et al., 2002; data for CS calculations therefore CS values could be easily compared if the baseline calculations are homogenous.

**Line 340: is open sea microalgae cyanobacteria? Can the authors more clearly show/explain what time periods were for coastal macroalgae and blue green algae? Did they these bloom/exposure events overlap? If so, to what extent?**

The bloom is mostly the microalgae cyanobacteria with a mix of macroalgae which are mainly exposed during the receding tides and are speculated to emit biogenic gases when the they start to decay during the ending phase of cyanobacterial bloom (mid-August 2019: ref, SYKE press release 2019 and Kowanka et al., 2020). At this point we cannot say precisely that when (or if at all) the bloom or exposure events overlap, considering that the bloom was widespread in the Northern Baltic sea, Gulf of Finland, Gulf of Riga and other coastal areas of Finland.

**Line 406: For surface emissions to be carried to the measurement site, the surface wind speed is important. Is this wind velocity at the surface/altitude of the measurement site? Or does it include a component of vertical velocity? In other words, how confident are the authors that the air mass is not vertically mixing downwards which would dilute the surface emissions?**

The wind velocity is measured at almost the same altitude as the measurement site, so we assume that the air mass is not vertically mixing downwards. The wind data was taken from the wind vane installed at the roof of Physicum building (roof of 5th floor) and our CI-APiTOF measuring the gaseous precursors was situated on the floor just below the roof top with the inlet sticking out of the 4th floor window.

However, as stated in the MS also that the measurements for particle size distributions was carried out at SMEAR III, which is 25 m a.m.sl and the wind vane at the Physicum building was situated roughly at 50 m a.m.s.l. So we agree that the particle size distribution data might not be completely free from downward vertical mixing of airmass and this we accept as the limitation of this study.

**Line 424: MSA also originates from agriculture.**

Yes, we agree that apart from marine sources MSA sourced from DMS can be emitted from agricultural practices and also biomass burning. However, in the global scenario the Sulphur emissions (considering DMS to be the most important Sulphur source) ocean contributes to 19% while the terrestrial emissions account only for 0.4% and out of this agriculture accounts for 2.7% of the sulphur emissions in the latitudes 65°N-50°N (Bates et al., 1992).

We have included "agriculture" in the lines mentioned in the manuscript.

"however some emissions could be sourced from agriculture and other terrestrial sources, Bates et al., 1992"

**Line 459: could a burst in sub-3 nm particles be from a suppression in growth and not a local nucleation event?**

We agree with the reviewer that in these local clustering there is no growth of clusters beyond a certain diameter. And we follow the NPF classification of mainly Lubna et al., 2018 which clearly says "The type IB, or ion bursts, is an attempt at NPF, during which clusters form in Hyytiälä; however, they do not grow beyond a few nanometres in diameter". So yes they are kind of ion bursts which call "local clustering" in this work following the description of Lubna et al., 2018.

However for clarity we removed the word "NPF" from the statement:

"These  so-called bursts /spikes appearing at small sizes (sub-3 nm) are indicative of local clustering processes in contrast to regional events, where it is possible to follow the growing particle mode for several hours (Dada et al., 2018; Dal Maso et al., 2005)"

**Figure comments:**

**Figure 2: Is this figure necessary? The manuscript only details specific events that occurred in short periods of time. It would be more helpful to see this data with the event data.**

We have included the wind data along with the event data. We tried to club other meteorological parameters like wind speed with the event data, however the event figures looked crowded. Hence, we decided to include only the most important parameter.

However, we feel it's important to describe the overall meteorological conditions during the study period since it helps the readers to get an understanding that what kind of environment the authors are talking about (hot/humid/windy etc) before they get to read about the new particle formation processes in the atmosphere of the this study site.

**Figure 5,7,8 (and their siblings in the SI) are very difficult to read. The font on the labels is too small to read. It might be easier to have the timeline graphs vertically stacked so it's easier to compare between them. The F panel is strange. Are the maps of the same area? It doesn't seem like it. Why have two panels for F**?

The font size has been increased for all the panels in the figures (5,7and 8). The timeline graphs are stacked as shown below. The Figure 5f (trajectory + Chla data showing bloom) would be made as Fig6 (a, b) as shown below, for more clarity. Accordingly all figure numbers would be changed in the entire MS. Panel F in all figures would be made as a separate figure. Yes, the graph panel "f" is of the same area the zoom percentage is different in the two maps (for clarity). We have two panels as one data(map) is for Chla concentrations taken from GlobColour level-3 as a separate Map, whereas the other figure is plotted through HYSPLIT and data taken from GDAS. Superimposing both the maps may not be possible. However, we have tried to make the Chla map of the same zoom-level as the trajectory map for a better comparison.

[Figure]

Figure 5: NPF Event- 30 June 2019 (a) Number size distribution of particles (data from PSM, NAIS and DMPS; size range: sub-3–100nm). (b) Charged particles number size distribution (negative: upper, positive: lower) obtained from the NAIS. (c) Diurnal variation of formation rates ($J_{1.5}$) of 1.5 nm particles and ions ($J^-_{1.5}$ and $J^+_{1.5}$) on the left axis and particle number concentrations (1.5–3 nm) on the right axis. (d) Diurnal variation of HOMs SA, IA and MSA with wind direction (WD). (e) The diurnal variation of particle concentration in nucleation:3–20 nm; aitken: 25–100 nm and accumulation: >100nm) mode particles during the event (Data from DMPS).

[Figure]

Figure 6: (a) Chl-*a* concentrations (MODIS); Black line shows the trajectory direction and the star point denotes the measurement site. (b) Trajectory frequency plot (100 a.g.l, arrival time of trajectory at the measurement site: 22:00 h) for 24 h back trajectory using GDAS meterological input data (frequency grid resolution: $1.0^\circ \times 1.0^\circ$).

**Figure S3: The labels are too small to read. Units of residence time?**

The font size has been increased. Since the residence time is normalized so it has no units. The term "normalized residence time" has been included in the figures.

Although we have already also mentioned this in the section 2.2 of the MS

"The residence times were normalized for clarity in the all the figures and is shown on a scale of 0 to 1 (Results are included in the supplementary information)"

[Figure]

Figure S4 (previously Figure S3): Normalized Residence times of air masses (3-day backwards) arriving at the experimental site on 30 June 2019. The color bar indicates the normalized residence times for each subplot. The residence time of particles originating 3 days before reaching SMEAR III is shown for 6:00 h, 9:00 h, 12:00 h and 15:00 h. The red shaded areas indicate the latitude/longitude pairs having the maximum residence time.

The Residence times figures S6 and S9 has also been changed accordingly in the revised version of the MS.

**New references added in the MS:**

Aalto, P., Hämeri, K., Becker, E., Weber, R., Salm, J., Mäkelä, J. M., Hoell, C., O'Dowd, C. D., Karlsson, H., Hansson, H. C., Väkevä, M., Koponen, I. K., Buzorius, G., and Kulmala, M.: Physical

Characterization of Aerosol Particles during Nucleation Events. *Tellus B*, 53:344–358, https://doi.org/10.3402/tellusb.v53i4.17127, 2001

Bates, T.S., Lamb, B.K., Guenther, A., Dignon, J., Stoiber,R.E.: Sulfur emissions to the atmosphere from natural sources., J Atmos Chem., 14**,** 315–337 (1992). https://doi.org/10.1007/BF00115242.

Bergman, T., A. Laaksonen, H. Korhonen, J. Malila, E. M. Dunne, T. Mielonen, K. E. J. Lehtinen, T. Kühn, A. Arola. and H. Kokkola.: Geographical and diurnal features of amine-enhanced boundary layer nucleation, J. Geophys. Res. Atmos., 120, 9606–9624, doi:10.1002/ 2015JD023181., 2015.

Carbone, M. S., Park Williams, A., Ambrose, A. R., Boot, C. M., Bradley, E. S., Dawson, T. E., Schaeffer, S. M., Schimel, J. P. and Still, C. J: Cloud shading and fog drip influence the metabolism of a coastal pine ecosystem Global Change Biol., 19, 484–97, 2013.

Crump, J.G. and Seinfeld, J.H.: Turbulent deposition and gravitational sedimentation of an aerosol in a vessel of arbitrary shape, J.Aerosol Sci., 12, 405-415.

Ehn, M., Thornton, J. A., Kleist, E., Sipilä, M., Junninen, H., Pullinen, I., Springer, M., Rubach, F., Tillmann, R., Lee, B., Lopez-Hilfiker, F., Andres, S., Acir, I.-H. H., Rissanen, M., Jokinen, T., Schobesberger, S., Kangasluoma, J., Kontkanen, J., Nieminen, T., Kurtén, T., Nielsen, L. B., Jørgensen, S., Kjaergaard, H. G., Canagaratna, M., Maso, M. D., Berndt, T., Petäjä, T., Wahner, A., Kerminen, V.-M. M., Kulmala, M., Worsnop, D. R., Wildt, J. and Mentel, T. F.: A large source of low-volatility secondary organic 556 aerosol, Nature, 506(7489), 476–479, doi:10.1038/nature13032, 2014.

Emery, N. C., D'Antonio, C .M. and Still, C. J.: Fog and live fuel moisture in coastal California shrublands Ecosphere, 9, e02167, https://doi.org/10.1002/ecs2.2167 ,2018.

Hemmilä, M., Hellén, H., Virkkula, A., Makkonen, U., Praplan, A. P., Kontkanen, J., Ahonen, L., Kulmala, M., and Hakola, H.: Amines in boreal forest air at SMEAR II station in Finland, Atmos. Chem. Phys., 18, 6367–6380, https://doi.org/10.5194/acp-18-6367-2018, 2018.

Hirschfelder, J. 0., Curtiss, C. F., and Bird, R. B. : Molecular Theory of Gases and Liquids,John Wiley & Sons, New York, p 539, 1954.

Lawson, D. M., Clemesha, R.E.S., Vanderplank, S., Gershunov, A. and Cayan, D.: Impacts and influences of coastal low clouds and fog on biodiversity in San Diego California's Fourth Climate Change Assessment CCCA4-EXT-2018-010 69–89, https://www.energy.ca.gov/ sites/default/files/2019-12/Biodiversity_CCCA4-EXT-2018-010_ada_0.pdf, 2018.

Lehtinen, K. E. J. and Kulmala, M.: A model for particle formation and growth in the atmosphere with molecular resolution in size, Atmos. Chem. Phys., 3, 251–257, https://doi.org/10.5194/acp-3-251-2003, 2003.

Makkonen, U., Virkkula, A., Hellén, H., Hemmilä, M., Sund, J., Äijälä,M., Ehn, M., Junninen,H., Keronen,P., Petäjä,P., Worsnop,D.R., Kulmala, M. and Hakola, H: Semi-continuous gas and inorganic aerosol measurements at a boreal forest site: seasonal and diurnal cycles of $NH_3$ , HONO and $HNO_3$., Boreal Environment Research 19 (suppl. B), 311–328, ISSN 1797-2469, 2014.

Makkonen, U., Virkkula, A., Mäntykenttä, J., Hakola, H., Keronen, P., Vakkari, V., and Aalto, P. P.: Semi-continuous gas and inorganic aerosol measurements at a Finnish urban site: comparisons with filters, nitrogen in aerosol and gas phases, and aerosol acidity, Atmos. Chem. Phys., 12, 5617–5631, https://doi.org/10.5194/acp-12-5617-2012, 2012.

Ortega, I. K., Olenius, T., Kupiainen-Määttä, O., Loukonen, V., Kurtén, T., and Vehkamäki, H.: Electrical charging changes the composition of sulfuric acid–ammonia/dimethylamine clusters, Atmos. Chem. Phys., 14, 7995–8007, https://doi.org/10.5194/acp-14-7995-2014, 2014.

Rissanen, M. P., Mikkilä, J., Iyer, S., and Hakala, J.: Multi-scheme chemical ionization inlet (MION) for fast switching of reagent ion chemistry in atmospheric pressure chemical ionization mass spectrometry (CIMS) applications, Atmos. Meas. Tech., 12, 6635–6646, https://doi.org/10.5194/amt-12-6635-2019, 2019.

Schade, G. W., and P. J. Crutzen.: Emission of aliphatic-amines from animal husbandry and their reactions: Potential source of N2o and Hcn, *J. Atmos. Chem.*, **22**(3), 319– 346, doi:10.1007/BF00696641., 1995.

Sipilä, M., Sarnela, N., Jokinen, T., Henschel, H., Junninen, H., Kontkanen, J., Richters, S., Kangasluoma, J., Franchin, A., Peräkylä, O., Rissanen, M. P., Ehn, M., Vehkamäki, H., Kurten, T., Berndt, T., Petäjä, T., Worsnop, D., Ceburnis, D., Kerminen, V.-M. M., Kulmala, M., O'Dowd, C. and O'Dowd, C.: Molecular scale evidence of aerosol particle formation via sequential addition of HIO3, Nature, 537(7621), 532–534, doi:10.1038/nature19314, 2016.

Sipilä, M., Sarnela, N., Jokinen, T., Junninen, H., Hakala, J., Rissanen, M. P., Praplan, A., Simon, M., Kürten, A., Bianchi, F., Dommen, J., Curtius, J., Petäjä, T., and Worsnop, D. R.: Bisulfate – cluster based atmospheric pressure chemical ionization mass spectrometer for high-sensitivity (< 100 ppqV) detection of atmospheric dimethyl amine: proof-of-concept and first ambient data from boreal forest, Atmos. Meas. Tech., 8, 4001–4011, https://doi.org/10.5194/amt-8-4001-2015, 2015.

SYKE press release (29 August 2019), Summary of algal bloom monitoring June-August 2019: Cyanobacteria were mostly mixed in the water in the Finnish sea areas, in lakes the cyanobacteria situation varied a lot. https://www.syke.fi/enUS/Current/Press_releases/Summary_of_algal_bloom_monitoring_JuneAu(51391).

Wang, M., He, X.-C., Finkenzeller, H., Iyer, S., Chen, D., Shen, J., Simon, M., Hofbauer, V., Kirkby, J., Curtius, J., Maier, N., Kurtén, T., Worsnop, D. R., Kulmala, M., Rissanen, M., Volkamer, R., Tham, Y. J., Donahue, N. M., and Sipilä, M.: Measurement of iodine species and sulfuric acid using bromide chemical ionization mass spectrometers, Atmos. Meas. Tech., 14, 4187–4202, https://doi.org/10.5194/amt-14-4187-2021, 2021.

Xiao, M., Hoyle, C. R., Dada, L., Stolzenburg, D., Kürten, A., Wang, M., Lamkaddam, H., Garmash, O., Mentler, B., Molteni, U., Baccarini, A., Simon, M., He, X.-C., Lehtipalo, K., Ahonen, L. R., Baalbaki, R., Bauer, P. S., Beck, L., Bell, D., Bianchi, F., Brilke, S., Chen, D., Chiu, R., Dias, A., Duplissy, J., Finkenzeller, H., Gordon, H., Hofbauer, V., Kim, C., Koenig, T. K., Lampilahti, J., Lee, C. P., Li, Z., Mai, H., Makhmutov, V., Manninen, H. E., Marten, R., Mathot, S., Mauldin, R. L., Nie, W., Onnela, A., Partoll, E., Petäjä, T., Pfeifer, J., Pospisilova, V., Quéléver, L. L. J., Rissanen, M., Schobesberger, S., Schuchmann, S., Stozhkov, Y., Tauber, C., Tham, Y. J., Tomé, A., Vazquez-Pufleau, M., Wagner, A. C., Wagner, R., Wang, Y., Weitz, L., Wimmer, D., Wu, Y., Yan, C., Ye, P., Ye, Q., Zha, Q., Zhou, X., Amorim, A., Carslaw, K., Curtius, J., Hansel, A., Volkamer, R., Winkler, P. M., Flagan, R. C., Kulmala, M., Worsnop, D. R., Kirkby, J., Donahue, N. M., Baltensperger, U., El Haddad, I., and Dommen, J.: The driving factors of new particle formation and growth in the polluted boundary layer, Atmos. Chem. Phys., 21, 14275–14291, https://doi.org/10.5194/acp-21-14275-2021, 2021.

---

## Author Response (AR1)

**REVIEWER 1**

**General Comments from Referee**

This paper by Thakur et al. explores new particle formation events in Helsinki from gaseous precursors of marine (iodic acid, sulfuric acid) and anthropogenic (sulfuric acid) origin. Importantly, this study highlights the complexity of nucleation in a semi-urban location with marine and anthropogenic influence. The authors use a wide range of ground-based instruments to monitor particle size and concentration, in addition to measurements of key gas-phase species. These measurements are paired with meteorological and satellite observations to identify the source of the precursors to NPF. This study fills a measurement gap of nucleation events in coastal urban areas.

This paper has some interesting results that are valuable to the NPF community. With that, I find it suitable for publication in ACP. However, this paper would benefit from clearer explanations of how the conclusions were reached, or perhaps some softening of their conclusions. Furthermore, I believe this work could use some editing for clarity.

**Author's response:** We thank the reviewer for appreciating the work and providing the specific and very valuable comments which has considerably improved the quality and clarity of the manuscript.

**Author's changes in manuscript:**

We have answered the queries/comments for each point as detailed below. The corrections are incorporated in the revised manuscript including softening of the conclusions reached in this work.

We have moved the sections describing "Back Trajectory calculations" and "Meteorological and other supporting data" to the supplementary information for shortening the MS. Also editing of the sentences has been done in various sections for better clarity.

**Specific Comments from the Referee**

- **Comment from Referee:** In the abstract (L49), the authors indicate that the type of phytoplankton species and the intensity of the bloom was one of the most important factors affecting aerosol precursor vapor concentrations (IA and SA). How was this conclusion reached, when the only measurements made in this study to link their gas and particle phase measurements to biological activity were satellite measurements of Chl-$a$, which does not differentiate between species?

**Author's response:** We agree with the reviewer, that we did not make any actual measurements of the algal species neither did we do any species identification for this study. But we speculate that the emissions from the type of phytoplankton species found in a particular area (area selected based on the trajectory of air mass) might be influencing the gas concentrations in the atmosphere. Our interpretations are based on the residence time of air masses in a particular marine region. We made the best possible estimations on the species present in that region based on Baltic-wide monitoring of cyanobacterial blooms from previous studies mentioned in the MS. As per these studies (Knutson et al., 2016; Attard et al., 2019; Kownacka et al., 2020) results show that bloom composition is fairly consistent for different regions and seasons from year to year, which makes it possible for us to make close estimations of the species present during our study in a particular region (from where the airmass travels). Importantly, however, the bloom composition during summer is different from spring time blooms, which we detail in our study and helps us interpret particle formation and their potential sources.

**Author's changes in manuscript:** Accordingly we have added /modified the statements in the section 3.4: L842-847, Pg:34, "In addition, the Bothnian Sea and Gulf of Finland were found to be rich in cyanobacterial genera of *Aphanizomenon* along with *Nodularia* and *Dolichospermum* (Kownacka et al., 2020). As per the previous studies which carried out the Baltic-wide monitoring (Kowancka et al., 2020 and the references mentioned therein) that bloom composition is fairly consistent for different regions and seasons from year to year, which makes it possible for us to make close estimations of the species present during our study in a particular region (from where the airmass travels and the residence time over a particular region)".

- **Comment from Referee:** The authors also conclude on L696 that the type of phytoplankton species, bloom intensity, and distance from the bloom plays an important role.

  1. **How does the phytoplankton species affect the gas-phase concentration in their measurements?**

     **Author's response:** Not all plankton species emit DMS (a precursor for biogenic SA). There are only very few specific species found in some particular areas that may be relevant to account for regarding their contribution to biogenic SA in the atmosphere. Similarly, specific species of macroalgae are responsible for large emissions of $I_2$ which finally oxidizes to IA either at the source or during their transport to the study site. A detailed explanation of the species and their niche is explained in section 3.2.

**Author's changes in manuscript:** The authors have soften the conclusion as follows L905-907, Pg:36, "The type of phytoplankton species, intensity of the bloom and distance of the bloom from the experimental site is speculated to play an  important role in determining the concentrations of precursor gases and thus influence the duration and type of NPF".

2.  **Is there an instance where there was a sea wind with less intense phytoplankton bloom, and no NPF events?**

**Author's response:** Yes, we observed a few days when this occurred; for example, on August 13 and August 17 when there was no event, yet the there was a sea wind and the bloom was less intense as compared to other event days. An example of such a day (August 17) is shown in the figure S13. The figure and this required explanation is incorporated in the supplementary information as reference.

[Figure]

**Figure S13:** No event day, 17August 2019 (a): Satellite map showing Chla concentrations (GlobColour level-3) (b) Trajectory analyis plot (100 a.g.l) for 24 h back trajectory using

GDAS meterological input data (frequency grid resolution: $1.0° \times 1.0°$). (c) Charged particle number size distribution (negative: upper, positive: lower) obtained from the NAIS.

**Author's changes in manuscript:** This figure has been incorporated in the SI as Figure S13. The relevent explanation has been added in the main MS, L824-828, Pg: 34.

"To confirm our findings we checked a day where there was less intense bloom in the Gulf of Finland and Northern Baltic Sea and the dominant airmass did not pass over the bloom patch in Gulf of Finland (Figure S13) before entering our experimental site. We did not observe an NPF event on this day, thereby suggesting that the airmasses passing over the bloom patches before arriving at our study site were capable of bringing in biogenic precursor vapors capable of initiating NPF events".

**Comment from Referee:** I'm also not convinced by the importance of the cyanobacterial blooms on the IA concentration, especially when compared to the other algae and marine sources. The authors timed their study to match with the cyanobacterial blooms that are expected in the Baltic Sea and coastal regions of Finland. In section 3.1.2 however, the authors emphasized that the cyanobacterial blooms were reduced below normal in July and August, which were the time periods in which they observed the NPF events. The authors also point out that the low tide and high irradiance could be a source of macroalgae iodine, as was observed in McFiggans et al., 2010.

**Author's response:** In section 3.3.2 (Case 1) we propose that the contribution of macroalgae to IA could be the dominant source when IA is speculated to play a dominant role in initiating the burst event on 11 August. However, we have based our conclusions only on the high Chl-*a* values in the region from where the air masses originated and where the air mass residence time was the highest. This section does not talk about the dominant contribution of IA from the Cyanobacterial blooms.

In section 3.3.2 (Case 2): However, we speculate that when the wind direction (coinciding with high residence times) was over the bloom areas which are dominated by specific cyanobacteria producing DMS (a precursor to SA), we see an increase in SA followed by a burst event. The interpretation regarding particular species of cyanobacteria in the respective marine and coastal areas is based on ongoing yearly cyanobacteria monitoring coordinated by the Finnish Environment Institute (and cited in the text). No species identification was done for this study. However as stated above the bloom composition is relatively stable for particular seasons and areas between years, but their intensity may vary depending upon temperature and nutrient availability.

Section 3.1.2: It is correct that cyanobacterial/algal blooms were less intense at end of July and August. And it is worth noting that when the blooms of both cyanobacteria and indeed macroalgae start to decay and die (while being exposed to sunlight) they produce more emissions (biogenic SA and IA). Thus, this itself is a reason for speculating why we see most of the NPF events in later summer months. All these observations suggest that there could be a strong link between algal/cyanobacterial emissions and their impact on NPF. However, as stated in our conclusions further studies are definitely needed to confirm these findings in a coastal setting.

**Author's changes in manuscript:**

In order to clarify this point we have included the following lines in the section 3.1.2

L394-L399, Pg:13, "However, the weather conditions in end of July began changing with high winds causing the cyanobacteria to be highly mixed in the water column, which reduced bloom intensity at the sea surface to lower than normal mean cyanobacterial biomass (mean biomass of cyanobacteria, 105 µg L$^{-1}$, Kownacka et al., 2020) in end of July and August (SYKE press release, 2019). However the average biomass of cyanobacteria in 2019 (196 µg L$^{-1}$, Kownacka et al., 2020) was slightly higher than the average".

L406-419, Pg:14 "There is a possibility that reasonably, large extents of coastal macroalgae, including *F. vesiculosus*, were exposed to direct sunlight (in shallow waters or low tide conditions) during the decay of the blooms during mid-August (when the bloom intensity was low, SYKE press release, 2019), hence making this time window favorable for observing potentially high emissions in gas phase from macroalgae, in addition to the emissions from cyanobacterial blooms".

Also the conclusion section of the study has been modified accordingly (L887-890, Pg 36):

"In fact, an overall higher impact of biogenic emissions was noted in this semi-urban site particularly during end of July and mid-August when the bloom intensity decreases and the cyanobacteria/macroalage start to decay and die, while being exposed to sunlight, they produce more biogenic emissions biogenic of SA and IA"

**Comment from Referee:** Would Chl-a measurements also measure the contribution from macroalgae?

**Author's response:** Yes the satellite Chl-a measurements can indicate if the contribution is from Macroalgae in case of high values (higher than the average) of Chl-a. This might indicate that there are floating/exposed macroalgae present. However, typically these algae are not present in the open sea areas of the Baltic Sea and the resolution of the satellite Chl-a are not in such high resolution as to permit an interpretation (or differentiation) with higher confidence, hence it's not mentioned in the MS. Nevertheless, the Baltic Sea has a great abundance of macroalgae along its coasts that we speculate could be contributing to the IA signal.

**Author's changes in manuscript:**

For clarification the following Lines have been added in the supplementary material where the description of Chl*a* measurements are given (L71-74, Pg: 3).

"However this resolution is not high enough to demarcate the contribution of Chla from cyanobacteria and macroalgae in the marine region. Nonetheless, the contribution of macrolagae to Chla still holds a significant place since the Baltic Sea and other regions of Gulf of Finland are abundant in microalgae"

- **Comment from Referee:**

 L485: The authors indicate that the change in wind direction 'apparently discontinued the precursor vapor source', however I'm not sure why this is apparent? From Figure 5(d), the concentrations of SA, MSA and IA remain relatively constant with the change in wind direction.

**Author's response:** We thank the reviewer for pointing this misinterpretation in the manuscript. We accept that the changed air-mass just discontinued the growth and not the precursor vapor concentrations.

**Author's changes in manuscript:** We have clarified this and changed the lines (L624-L632, Pg: 23-24): "However, we also observe a drop in Aitken particles before NPF which also continues during NPF. We speculate it could be due to the change in wind direction (Vakeva et al., 2000) before NPF. The wind direction relatively remains constant throughout the NPF so the low concentration of Aitken mode continues. Wind Direction changes abruptly at 12:00h and the Aitken mode particle concentrations increases soon after this change of wind direction.  (Fig. 5d). This shows the particles must be the process of growth mostly elsewhere, which is not evident in the changed air mass,

however we still observe almost the same (or even slightly higher) precursor vapor concentrations, since the wind still passed over the bloom areas before entering our study site".

- **Comment from Referee:** Figure 6: Is the green trace called 'particles' the measured particles? Perhaps make that more clear.

  **Author's response:** The figure caption states that these are observed particles (measured through NAIS).

  **Author's changes in manuscript:**

  However, we have modified the caption of the figure for clarity, Pg:24, "**Figure 7:** Particle growth rates calculated from the kinetic condensation of gases (data from CI-APi-ToF) and the measured particle GRs (data from NAIS) in different size classes on 30 June 2019."

- **Comment from Referee: L593: How do you know all the I2 was oxidized to IA?**

  Author's response: The reviewer has correctly pointed out that not all $I_2$ is oxidized to IA. Also, in the present study we cannot give an estimate that how much $I_2$ (from source region) would be converted to IA.

  **Author's changes in manuscript:** The lines 769-770, Pg: 30, have been changed to "By the time the air mass reached our measurement site from the emission source, a fraction of the emitted $I_2$ could have oxidized to IA".

- **Comment from Referee:** The authors often use parentheses to provide additional details within the text. In some cases, the parentheses are unnecessary and interrupt the flow of the text. I suggest the authors review their use of parentheses for clarity. Some examples:

  L32: Several studies have investigated New Particle Formation (NPF) events from various sites ranging from pristine locations, including (boreal) forest sites to urban areas. There have been studies of more than just boreal forests, I'm not sure why boreal was specified here. Can remove the parentheses and/or the word boreal.:

  **Author's response:** The parentheses has been removed as per the reviewer's suggestion.

**Author's changes in manuscript:** Line L32-33, Pg: 1 **"**Several studies have investigated New Particle Formation (NPF) events from various sites ranging from pristine locations, including forest sites to urban areas"

**Comment from Referee:** L101: The parentheses around 'produced from macroalgae' can be removed.:

**Author's response:** The parentheses has been removed as per the reviewer's suggestion.

**Author's changes in manuscript:** L115-L117, Pg:4, The study from the Roscoff coast suggests that the daytime emissions of $I_2$ produced by macroalgae during low tides drives the particle formation (McFiggans et al., 2010).

**Comment from Referee: L499: Can be rewritten as 'The high normalized signals…' to remove the parentheses.:**

**Author's response:**  The parentheses has been removed as per the reviewer's suggestion.

**Author's changes in manuscript:** L581-583, Pg: 20: The high normalized signals of DMA-SA cluster seen during the entire event (increasing from the start of NPF event) possibly indicates that SA clusters initiate the event (Fig. S4a).

**Comment from Referee:** L355: Can use 'The daily mean' instead of The mean (whole day).

**Author's response:** We corrected the Line in the MS as per the reviewer's suggestion. Also this correction has been implemented throughout the MS where "mean (whole day)" was mentioned.

**Author's changes in manuscript:** L433-434, Pg:14, "The daily mean concentration of SA in July and August were almost similar,~$3 \times 10^6$ molec. cm$^{-3}$ ".

**Technical Corrections:** *All the technical corrections have been incorporated in the revised MS*

**Comment from Referee: L42: Keep the chemical names in lowercase "sulfuric acid (SA)" to match L78**

**Author's response:** It has been corrected in the MS as suggested by the reviewer.

**Author's changes in manuscript: Lines L43-49, Pg:1-2,** Although the overall anthropogenic influence on sulfuric acid (SA) concentrations was low during the measurement period, we observed that the regional or local NPF events, characterized by SA concentrations in the order of $10^7$ molecules per $cm^{-3}$ occurred mostly when the air mass travelled over the land areas. Interestingly, when the air mass travelled over the Baltic Sea, an area enriched with Algae and cyanobacterial blooms, high iodic acid (IA) concentration coincided with an aerosol burst or a spike event at the measurement site".

We also abbreviated the terms "sulphuric acid" to SA and "iodic acid" to IA in the whole MS, after defining the full term on the first usage.

**Comment from Referee:L46: Chemical names in lowercase "iodic acid (IA)"**

**Author's response:** It has been corrected in the MS as suggested by the reviewer.

**Author's changes in manuscript:** L43-L49,(as mentioned above).

**Comment from Referee:L150: Use the abbreviation for New Particle Formation (NPF)**

**Author's response:** It has been corrected in MS as suggested by the reviewer.

**Comment from Referee:L196: I'm not sure what 'mlpm' is, define it?**

**Author's response:** mlpm stands for milliliter per minutes, we have corrected these unit to "mLpm" in the revised MS.

**Author's changes in manuscript: L249-L252, Pg: 8, "**. In this study, the chemical ionization was done via nitrate ions ($NO_3^-$) through X-ray exposure of nitric acid ($HNO_3$, flow rate: 3 mLpm), saturating the sheath air flow entering the CI (flow rate: 30 Lpm), the inlet flow of 10 Lpm was reached by using a 40 Lpm total flow".

**Comment from Referee:L205: Define HOMs when it is first used**

**Author's response: Thankyou for pointing this out. We made the required correction in the MS**

**Author's changes in manuscript: L270-L272, Pg: 9,** "Please note that the concentration of highly oxygenated molecules (HOM, monomers and dimers) were calculated from the unit mass resolution data".

**Comment from Referee:L208:** extra 'The'. Don't need to define UMR if you only use it once.

**Author's response:** We thank the reviewer for pointing this out. We made the required correction in the MS

**Author's changes in manuscript:** L270-L272, Pg: 9, "Please note that the concentration of highly oxygenated molecules (HOM, monomers and dimers) were calculated from the unit mass resolution data".

**Comment from Referee:L263: Don't need to redefine growth rate.**

**Author's response:** In the revised MS this section has been moved to the supplementary information. Here, in SI, it has been used for the first time, so we still redefine it.

**Author's changes in SI :** L76-79, Pg:3, "The growth rates (GRs) were calculated based on the 50% appearance time method using the NAIS ion data from both polarities, depending on the better quality polarity (Dada et al., 2020a; Dal Maso et al., 2016; Lehtipalo et al., 2014). »

**Comment from Referee: Table S1: $O_3$ instead of $O_2$?**

**Author's response:** Yes we mean $O_3$, thanks for pointing out this typo error. We have corrected it now to $O_3$.

**Author's changes in manuscript:** Table changed as shown below, Pg:4-5 in SI.

| Parameter measured | Technique | Instrument | Resolution and detection limits | Site of Measurement |
|---|---|---|---|---|
| $SO_2$ | UV-fluorescence technique | Horiba APSA 360 | 60 s

detection limit: 0.2 ppb | a |
| $NO_x$ | Chemiluminescence technique + thermal (molybdenum) converter | TEI42S | 60 s

detection limit: 0.2 ppb | a |
| $O_3$ | IR-absorption photometer | TEI 49 | 60 s

detection limit: 0.5 ppb | a |
| Air Temperature | Platinum resistance thermometer | Pt-100 | 60 s | b |
| Wind direction | 2-D ultrasonic anemometer | Thies Clima ver. 2.1x | 10 s | b |
| Wind Speed | Platinum resistance thermometer + thin film polymer sensor | Vaisala DPA500 | 4 min | b |
| Relative humidity | Platinum resistance thermometer + thin film polymer sensor | Vaisala DPA500 | 4 min | b |
| Global Radiation | Net radiometer | Kipp & Zonen CNR1 | 60s | b |
| Tidal Height | wave buoys | | c | Helsinki Suomenlina, Gulf of Bothnia, Northern Baltic Sea |

**Comment from Referee:447:** Missing a period?

**Author's response:** Period added now in the revised MS. Thanks for pointing the error.

**Author's changes in manuscript:**L543-L544, Pg:18, "These sources are indeed the most significant sources of $NO_x$ globally (Meixner and Yang, 2006)."

**Comment from Referee:556:** Replace $HIO_3$ with IA

**Author's response:** We replaced HIO3 with "IA". Also the replacement of HIO3 to IA was done elsewhere as well in the MS to maintain uniformity.

**Author's changes in manuscript:** L708-709, Pg:26, **"**This was the highest observed IA concentration in the entire measurement period. A recent study by He et al., 2021, indicate that IA concentrations above $1 \times 10^7$ molec. cm$^{-3}$ leads to rapid new particle formation at +10° C".

**Comment from Referee:** Figure 4 caption: Not sure what the yellow circles are for 'all time' – is it just the other time except the morning and evening?

**Author's response:** Yes that is correct, that yellow circles denote "all time". It is mentioned in the Figure caption also, as underlined below.

**Author's changes in manuscript:** Pg:17,"Figure 4: Correlation of SA with MSA (a,b), SO$_2$ (c,d) and NO$_x$ (e) for June–July. The black dashed lines for both axis represent the mean of the gas concentration, red dashed line represent the median value the gas concentrations and red solid line represents the linear fit. Spearmann's coefficient ($r_s$) was used to test the correlation, at significance level, 0.001. The circles represent data points at different hours of the day. The upward pointing green triangles represent the morning rush hours (6:00–8:00 h) and the downward pointing blue triangles represent the evening rush hours (15:00–17:00 h). The yellow hollow circles represent all data. NO$_x$ data unavailable of August"

**REVIEWER 2**

**General Comments from Referee**

Thakur et al. present field measurements of new particle formation events occurring in Helsinki. This site is impacted by air masses from the city and sea. Observations from Helsinki help fill in a critical gap in understanding how marine new particle formation impacts urban air quality. Their observations relate nearby algal and cyanobacteria blooms to marine new particle formation events.

Overall, the information presented in the paper is logical but some of the conclusions on which precursors contributed to which nucleation events are not persuasive. The paper is not written as concisely as it could be with many parts repeated and it's difficult to follow. Only some of these

instances have been pointed out here. The authors should try to shorten the paper. The study fits well in ACP. Several aspects of the manuscript should be improved prior to acceptance for publication.

**Author's response:** We thank the reviewer for providing the specific and very valuable comments which has considerably improved the quality and clarity of the manuscript. We have answered the queries/comments for each point as detailed below. The corrections has been be incorporated in the revised manuscript.

**Author's changes in manuscript:** We have tried to shorten the MS in the revised version as much as possible, basically modifying statements by replacing them with "short and precise lines" for a better flow and readability of the MS. Also we have moved the sections describing "Back Trajectory calculations" and "Meteorological and other supporting data" to the supplementary information for shortening the MS. The section describing the condensation sink has been removed from the MS altogether, since this parameter was not used in reaching any of the important conclusions made in this study.

However we accept that we could not considerably shorten the MS, since some new figures were added as per reviewer's suggestion. We made stacked time line figures for each event and made a separate figure for the trajectory analysis +Chl*a* satellite figures for each event for clarity, which increased the number of figures from a total of 8 figures to total 11 figures in the new revised version of the MS.

Major Comments:

**Comment from Referee:** Line 124, Why are more coastal measurements needed? The authors detail out a few studies conducted at the coasts and where they found correlations to coastal seaweed and algal blooms. What does this study add to the scientific field other than more measurements? How do measurements from Helsinki help the scientific field? I am sure these measurements are important but framing the "why" will help the reader better understand the purpose of this study.

**Author's response:** We thank the reviewer for highlighting this point. We have now included a new paragraph and also modified the existing write up in the "Introduction", which is as follows:

**Author's changes in manuscript:** Lines 79-90, Pg: 3 (new addition) "The measurements of gaseous precursors, meteorology and biogenic influences are important to study the coastal NPF, which may lead to the formation of coastal/marine clouds. Coastal clouds are the drivers of many coastal ecosystem (Carbone et al 2013, Emery et al 2018, Lawson et al 2018). Any impact or fluctuations in the cloud formation may impact several other processes of the fragile coastal ecosystem. These coastal clouds demonstrate a high sensitivity to CCN (He et al., 2021) and they have a significant impact on the radiation budget because they have a high infrared emission and albedo when compared to the dark water bodies down below. In this study we highlight the type of NPF processes and their drivers in a semi-urban-coastal setting where the atmosphere could be a mixture of anthropogenic and biogenic emissions. Unlike the above mentioned previous studies which were mostly carried out in a perfect coastal environments where NPF would be most likely affected by the biogenic emissions, this study helps to evaluate the impact of urban emissions Vs coastal emissions on NPF and at large the cloud formation processes.

Lines 137-179 (modified): "The limited NPF studies in the semi-urban coastal regions and the dynamic coastal meteorology drives the motivation of this research. A nother motivation for this research is that till date no detailed studies on the impact of biogenic emissions on NPF events were done before in Finland despite the fact that extensive cyanobacteria blooms occur every year in the Baltic Sea region and neighboring water bodies (including Finnish lakes) (Kahru and Elmgren 2014), which could be a significant source of iodine species, SA and MSA. Increasing temperatures and the excessive nutrient load in the Baltic Sea promote algal growth (Kuosa et al., 2017; Suikkanen et al., 2007, 2013). According to HELCOM (Baltic Marine Environment Protection Commission), the Baltic Sea has warmed $0.3°$ C per decade, however after 1990 significantly faster at $0.6°$ C per decade and in Finnish coastal areas the warming is even faster with a $2°$ C increase since 1990 (Humborg et al. 2019). The amount of blue-green algae (i.e. cyanobacteria) has shown a statistically significant increase in open sea areas in the Gulf of Finland, Sea of Åland and the Sea of Bothnia in the last 40 years (Kahru and Elmgren, 2014). The increase in frequency and intensity of cyanobacterial blooms would increase the potential emission of biogenic gases changing the composition of the overlying atmosphere and the atmosphere of the neighboring sites, depending on the meteorological conditions. In this semi-urban coastal setting the concentration of gaseous precursors and aerosol size distribution may be influenced by the local meteorological parameters such as wind direction, wind speed, (air mass) turbulences especially at the surface layer of the lower atmosphere. Coastal locations are dynamic environments with rapid changes in meteorological parameters, also making the study of NPF more challenging.

In this study, we aim at a thorough evaluation of aerosol precursor molecules with a detailed (NPF events) analysis during the cyanobacterial bloom period, in the coastal-city of Helsinki, Finland, from June to August (summer) 2019. This work evaluates the role of phytoplankton blooms and meteorological parameters in the NPF events observed during the measurement period. We also identify the major precursor vapor(s) and molecular clusters found during the aerosol events. Here, we formulate the hypothesis that gaseous precursors formed from the biogenic emissions from the surrounding marine areas could play an important role in the nucleation processes in Helsinki. Although Helsinki is a coastal area yet the role of marine emissions on NPF processes has not been studied before".

We hope that this is sufficient to bring out the real importance of the coastal measurements and again we thank the reviewer for helping us to make this research work better.

**Comment from Referee:** Line 199: The authors state that the CIAPiToF was calibrated following the procedure detailed in (Kürten et al., 2012). That study only calibrated the CI-APiToF for sulfuric acid. How does this calibration constant apply to MSA, iodic acid, and organics with nitrate as the chemical ionization reagent ion? What is the systematic uncertainty associated with using this calibration constant for non-sulfuric acid molecules? Often the authors report 3 significant figures on their precursor concentrations. Is this in-line with their estimated uncertainty?

**Author's response:** We thank the reviewer for highlighting this point. We have now included a new paragraph and also modified the existing write up in the section 2.1, for clarification, which is as follows:

**Author's changes in manuscript:** L259-266, Pg: 9, "The uncertainty range of the measured concentrations reported in this study is estimated to be −50%/+100% and the limit of detection, LOD: $4 \times 10^4$ molecules cm$^{-3}$ (Jokinen et al., 2012). HOMs and iodic acid have been estimated to be charged similarly at the kinetic limit as SA (Ehn et al., 2014; Sipilä et al., 2016), so the calibration factor for them should be similar, but please note, that the concentration of other compounds than SA can be highly uncertain due to different ionizing efficiencies, sensitivities and other unknown uncertainties. If MSA, IA or HOMs do not ionize at the kinetic limit these concentrations could be underestimated and thus, the concentrations reported in here should be taken as low limit values".

Thus to be in line with uncertainty associated with using the same calibration factor for non-sulfuric acid molecules we have corrected the concentrations of all species to 1 significant figure in the entire MS.

**Comment from Referee:** Along these same lines, what was the holdover time of SA, MSA, and IA (and other compounds) in the CI-APiToF inlet? On line 599, the event lasted less than 30 minutes. These compounds are very sticky and likely persist in the sampling lines even if the sampling rate is high. They likely persist at different rates so the order at which each compounds reaches its maximum concentration (and its absolute concentration at the maximum) will vary. Have the authors examined this to better determine if short burst new particle formation events can actually be studied with this instrument setup? How would time dependent wall loss rate impact the calculation of growth rates?

**Author's response:** Hoping that the reviewer meant residence time of the species in the inlet tube, an estimation can be provided. Considering the inlet length of 1 m and ID of 17mm, volume of one meter is 0.23 litres, with 10lpm inlet flow rate residence time is ca. 1.4 sec per meter. In our study we used inlet design as described by Eisele and Tanner (1993) and Kurten et al. (2011) and further it was used by Jokinen et al., 2012. In this type of inlet (with a inlet flow of 10Lpm) the interaction time between the sample and reagent ions is approximately 200ms. For the bisulphate-DMA cluster the negative free energy is high enough to be detected at this residence time (Ortega et al., 2014). For SA which is detected as $HSO_4^-$ ion in the CI-APiToF is reliable enough since the evaporating molecule/cluster could be $H_2SO_4/DMA/NH_3$ leaving behind bisulphate ion to be detected as $HSO_4^-$ + $(HNO_3)HSO_4^-$ (Ortega et al., 2014).

The loss rate is proportional to the square root of the diffusivity for the different molecules (Crump and Seinfeld, 1981). Although we agree, that this instrument is not completely free of the wall losses. Since wall losses are dependent on the flow rate, tube length and the diffusivity of the molecule we have corrected the final concentration for these losses by considering 50% loss for SA (concentrations corrected by a factor of 2 for 1m inlet length and 10 lpm flow rate). Wang et al., 2021 determined the decay rates of $HIO_3$ are 400 s for the Br-MION-CIMS. If we consider this rough estimate for $NO_3^-$ CIMS (present study, where we use Eisele type inlet, Eisele and Tanner, 1993), with the assumption that MION inlet minimally differs from the CIMS inlet (differing basically in the ion source orifice) we can suggest that the residence times was less than the decay rates of the iodine species, hence the instrument can be trusted that it gives close to accurate concentration of these species during a burst event.

The inlet of the CI-APiToF was designed to the use of coaxial sample and sheath flows in order to sample (extremely) low-volatile species which are easily lost to the walls (Riva et al., 2019). So as per the numerous other previous works who almost precisely quantified the ELVOCs through the same design scheme, the authors believe that SA, MSA and IA could be almost precisely quantified

by this instrument and flow scheme. Further Sipilä et al., 2016 has quantified the iodine species during an intense burst event at Macehead using a nitrate ion. CI-APiToF (event lasted >1hr). The instrument without chemical ionization is also capable of detecting the initial ions in the burst- NPF (Junninen et al., 2016).

**Author's changes in manuscript:** We take into account the diffusion loses while calculating the calibration factor and this is already mentioned in the MS text (Section 2.1). L252-254, Pg8-Pg9, "The instrument was calibrated prior to the experiment according to (Kürten et al., 2012) resulting in a calibration factor of $1.45 \times 10^9$ molecule per normalized unit signal including the diffusion losses in the inlet line"

Therefore we do not expect that "time dependent wall losses" to be significant enough to affect the GRs given that other losses have been accounted for prior to estimating the final concentrations.

**Comment from Referee:** Line 322: how long did the cyanobacteria bloom last during the measurement campaign? What area did it cover? In line 324, what does lower than normal mean? Lower than June? Some numbers would help put this intensity in perspective. In line 582, the authors comment that the blooms are intense but how does this compare to other periods of time. Is there a correlation of bloom intensity with IA, MSA, and SA concentrations (assuming the air mass is coming from the bloom's direction)?

**Author's response:** As per the SYKE press release (2019) the results from the annual monitoring of the Baltic Sea indicates that the  northern part of the Baltic Sea's main basin, entrance to the Gulf of Finland and south of the Åland Islands were enriched with blue-green algae (cyanobacteria). The bloom lasted from June-August 2019. In coastal areas, bloom was mostly spotted in the Archipelago Sea, Gulf of Finland, Bothnian Sea and the Quark. The bloom situation was highly variable in space, even over short distances. The fragmented nature of the coastal areas and changing wind and water currents makes the algal situation intensely dynamic.

The information/statement "Lower than normal" here is extracted from the SYKE press release 2019. It refers to the lower mean cyanobacterial biomass as compared to the previous years.

**Author's changes in manuscript:** We have provided some actual estimates of bloom intensity in the main text, which is as follows:

L211-217, Pg:7 , "As per the SYKE press release (2019) the  northern part of the Baltic Sea's main basin, entrance to the Gulf of Finland and south of the Åland Islands, were enriched with blue-green algae (cyanobacteria). The bloom lasted from June-August 2019. In coastal areas, bloom was mostly spotted in the Archipelago Sea, Gulf of Finland, Bothnian Sea and the Quark. The bloom situation developed rapidly and spatially highly variable, even over short distances. The fragmented nature of the coastal areas and changing wind and water currents makes the algal bloom conditions highly dynamic".

L394-L399, Pg:13, "However, the weather conditions in July began changing with high winds causing the cyanobacteria to be highly mixed in the water column, which reduced bloom intensity at the sea surface to lower than normal mean cyanobacterial biomass (mean biomass of cyanobacteria, $105\,\mu g\,L^{-1}$, Kownacka et al., 2020) by end of July and August (SYKE press release, 2019). However the average biomass of cyanobacteria in 2019 ($196\,\mu g\,L^{-1}$, Kownacka et al., 2020) was slightly higher than the average."

**Author's response:** No we did not find any good correlation (in terms of correlation coefficients) between the gaseous precursors and Bloom intensity mainly because of the following reasons:

1.      The trajectory distance covered by the air mass before entering the study site was quite large to accurately estimate the Chla concentrations along the path (without large uncertainties).

2.      The semi-urban setting of the experimental site may not allow us to accurately estimate the exact biogenic emissions from the source. Particularly if the source is situated in Baltic Sea, Gulf of Finland or Gulf of Bothnia (from where most of the trajectories passed before entering the study site.

For the above mentioned reasons, we opted to analyze the events, emissions and wind direction on a case-by-case basis, where we can provide more accurate estimations.

However, we agree with the reviewer's suggestion of correlating Chla with the precursors, which for this study may not possible. But based on this study and several other publications related to coastal NPF from our research group, we have secured funding for the next few years to establish a coastal atmospheric research observatory in Finland in collaboration with Tvärminne Zoological Research station, Hanko. Through this collaborative research with the biologists/ecologists we will obtain in situ data of Chl-a to accurately correlate with our gaseous measurements. Further, there would be less interferences in the gaseous precursor's concentration from other sources, which are mainly found in

an semi urban/urban setting. However, this current research would serve as the baseline study for this kind of future research in Finland which we have already highlighted in the discussion.

**Comment from Referee:** Line 331 and paragraph beginning 633: Did the authors actually measure the algae and cyanobacteria types during the blooms that occurred during the measurement period? Bloom composition can easily change based on numerous factors so it may not be a fair conclusion to link these previously measurements bacteria and algae types to what was observed during the measurement campaign. How confident are the authors that they can link these algal species to their new particle formation events?

**Author's response:** No, we did not do any actual measurements of the algal and cyanobacterial blooms. The data is obtained only from the satellite measurements and national monitoring conducted the Finnish Environment Institute. As per the various papers mentioned in MS, it seems that Bloom composition largely remains the same if we consider the blooms that occur either in early or late summer and in different regions. Hence, the bloom composition during summer is different than the Spring time bloom (Kownacka et al., 2020).

**Author's changes in manuscript:** L842-L847, Pg: 34, As per the previous studies which were carried out as part of the Baltic-wide monitoring (Kowancka et al., 2020 and the references mentioned therein), bloom composition is fairly consistent for different regions and seasons from year to year, which makes it possible for us to make close estimations of the species present during our study in a particular region (from where the airmass travels and the residence time over a particular region).

**Author's response:** We strongly speculate the link between the algal species and NPF events mainly because of the strong relationship of the changes in wind direction and the changes in the precursor vapors specifically IA. However we cannot be 100% sure for this speculation and as stated in the conclusions of the study we need more studies of coastal NPF near/around the coast of Finland to confirm our findings. We would soon start to measure the chemical composition of the NPF forming precursors right at the Hanko, Finland coast (Tvärminne Zoological Research) where the urban influences would be minimal and we can then quantify the algal emission more precisely. However, this baseline study is equally important against which we are have initiated further research in this domain.

**Author's changes in manuscript:**. We have mentioned this in the MS already that we need more studies to confirm our findings. However we have modified the lines L915-L918, Pg:37, "In order to

resolve these links require more quantitative studies are required, which aims to understand the correlation of quality and quantity of cyanobacterial blooms to the strength of emissions of aerosol precursors. More studies partitioning the influence of pelagic cyanobacterial blooms and influence of coastal macroalgae on new particle formations would need to be undertaken."

**Comment from Referee:** Line 471: From previous NPF campaigns, it seems that sulfuric acid concentration should increase before observed particle concentration? The text suggests the sulfuric acid concentration increased after particle concentration increased. Figure 5 shows that SA was already increasing. Did the authors observed any freshly formed clusters with the CIAPITOF? From line 499, it seems the authors did observe some clusters (and shown in Figure S4). It would be helpful/more logical to mention this earlier. Did the authors measure DMA and ammonia concentrations? If the authors believe SA-DMA is forming neutral clusters, do the authors know where the DMA is originating from? Why does the SA+DMA cluster concentration peak significantly after the new particle formation event (figure S7)? On line 474, what is a local clustering event? Does this refer to clusters observed on the CIAPITOF?

**Author's response:** The reviewer is correct that the all the NPF studies mention that SA starts to increase preceding a NPF. Line 504 mentions "Subsequently, SA concentration doubles from $2\times10^6$ to $4\times10^6$ molec. cm$^{-3}$". With this line we meant that SA concentration doubles at the start of NPF, however, it can be clearly seen form Figure 5 that the SA concentrations have been steadily increasing well before the NPF event starts.

We have not made any separate measurements of $NH_3$ or DMA. Our CI-APiToF was not tuned to measure masses lower than 50 a.m.u. And unfortunately, no separate instrument was deployed to make such measurements during the campaign. However, we do report clusters of SA-DMA-SA in our manuscript as normalized signals, since we did not make any separate calibrations to quantify DMA clusters. Moreover the calculations of concentrations of DMA-SA clusters would lead to large uncertaininties since they are prone to evaporation losses inside the CI-inlet (Sipilä et al., 2015).

There is currently no reported observation of DMA in Helsinki. Although previous studies have reported , the ambient air concentration of $NH_3$ ranged from 20 pptV to 830 pptV in the forest site (Hyytiäla) (Makkonen et al., 2014) and in the urban station (SMEAR III) from below 450 pptV to 3000 pptV (Makkonen et al., 2012). Dimethylamine concentration of 5 pptV has been shown to enhance atmospheric aerosol formation rate by more than a 1000-fold compared to an $NH_3$ concentration of 250 pptV (Almeida et al., 2013). DMA inclusive of other main methylamines like

mono and tri methylamines (Bergman et al., 2015) in the global inventory (Schade and Crutzen, 1995) is contributed through the animal husbandry and other agricultural practices, biomass burning and some contributions from marine and terrestrial sources. Although among these methylamine emissions from the above mentioned sources Trimethylamine dominates (Schade and Crutzen, 1995). Although there are no current estimates of DMA in the Helsinki region, but DMA estimates is available from the boreal forest region of Finland. The study of Hemmilä et al., 2018 states that the median concentration of DMA in July in Hyytiälä region was below the detection limit of the instrument. In this study the amines were detected using an online ion chromatograph (instrument for Measuring AeRosols and Gases in Ambient Air – MARGA) connected to an electrospray ionization quadrupole mass spectrometer (MS), with the detection limit as 0.2–3.1 ng m$^{-3}$. Further Sipilä et al (2015), measured DMA concentrations through a $NO_3^-$ CIMS and found that DMA was below ~ 150 ppqV in a boreal forest site. Their work also stated that DMA was unlikely the playing an important role in the nucleation process observed at the site.

Figure S8: shows the increasing signals of the clusters for the event on 11$^{th}$ August through which we speculate this an IA driven event since preceding the burst events we only observe IA (and $IO_3^-$) to be increasing and not SA-DMA. That is why it made us speculate that IA might be playing a more important role here in terms of initiating the particle formation. SA-DMA increase when we see the growth of particles at 15 hrs, probably indicating their more dominant role in the growth of particles along with IA.

Local clustering event here means that the molecules could be transported from elsewhere but the actual clustering could have taken place near the experimental site before the inlet, since it's a very small bump of clusters (with absolutely no growth) seen in the NAIS. It does not mean the clustering happening in the CI-APiToF inlet.

**Author's changes in manuscript:** To avoid the confusion the sentence has been changed. L572-574, Pg:19, "Preceeding the NPF event the SA concentrations were steadily increasing and subsequently at 09:00hrs" SA concentration doubles from $2\times10^6$ to $4\times10^6$ molec. cm$^{-3}$ (Fig. 5d)..."

As per the reviewer comment we have mentioned the observation of clusters earlier in the text, in the first aparagraph of describing the event (before the figure for the event).

L581-593, Pg.20: "The high normalized signals of DMA-SA cluster seen during the entire event (increasing from the start of NPF event) possibly indicates that SA clusters initiate the event (Fig.

S4a). DMA inclusive of other main methylamines like mono and tri methylamines (Bergman et al., 2015) in the global inventory (Schade and Crutzen, 1995) is contributed through the animal husbandry and other agricultural practices, biomass burning and some contributions from marine and terrestrial sources. Although among these methylamine emissions, generally the trimethylamine dominates (Schade and Crutzen, 1995). Although no estimates of DMA measurements are available from Helsinki region, the DMA in a boreal forest site in Finland has been estimated to be below $\sim$ 150 ppqV (Sipilä et al., 2015), measured through a $NO_3^-$ CIMS. Their work also stated that DMA was unlikely the playing an important role in the nucleation process observed at the site.

The increase of HOMs is also clearly observed during the event Fig. S4b. Therefore we suggest that nucleation and growth of particles was possibly due to SA-organics which ensures that particles reach the CCN and thus climate relevant diameters".

As per the reviewer comment we have added the clarification for local clustering:

L555-558, Pg19: "Local clustering here means that the molecules could be transported from elsewhere but the actual clustering could have taken place near the experimental site, indicated by a small bump of clusters (with absolutely little or no growth) as seen in the NAIS spectra.

**Comment from Referee:** Line 520: Why does MSA and IA concentrations need to increase in order to demonstrate they could participate in that NPF event? Also, from Figure S5, the concentrations of both are increasing (before the signal cuts off). How do the authors know for certain that these compounds are not participating in the event? Also HOM concentrations do not seem any higher than those in Figure 5. So why is this event SA-HOMs driven? What clusters did the CIAPITOF see? The organic clusters shown in Figure S4 just show a constant and slow increase in organic cluster concentrations throughout the day.

**Author's response:** Yes, we agree with the reviewer that to "participate in the NPF event" the concentrations need not to increase, if they are already significant in concentration. Perhaps as the other NPF studies suggest that a significant increase in concentration of IA and MSA (may be comparable or more than SA) is necessary in order for them to "initiate" nucleation (Beck et al., 2021; He et al., 2021). The changes in the manuscript related to other comments have been incorporated with proper explanations as detailed below:

**Author's changes in manuscript:** In the manuscript we have stated:

"Therefore, the precursor gases from the biogenic origin, IA and MSA do not show a significant concentration increase during this event and hence assumed to be contributing insignificantly to this event". The lines simply meant that they might participate but their contribution in the NPF event may not be as significant as SA.

However, for clarity we have modified the lines to the following:

L670-673, Pg: 25,"Since, the precursor gases from the biogenic origin, IA and MSA, do not show a significant concentration increase as compared to SA, at the start of this event, their contribution towards the initiation of the NPF event may not be as significant as SA."

Case of 30 June 2019 (Figure S4): If we compare the increase of the normalized signals of organics during NPF on 30 June 2019 with that of 11 August 2019, we can see that the cps started to increase well before the NPF on 30 June 2019 and keeps on steadily increasing throughout. With this comparison we can suggest that on 30 June NPF event the organics played an important role. Since this is a highly speculated conclusion, we have modified the sentence as follows:

L674-675, Pg:25, "indicating a SA driven event –with a possible contribution of HOMs"

**Comment from Referee:** Line 558: Isn't the temperature during this campaign much higher than 10C? Higher temperature will still favor SA+amine/ammonia nucleation. Observations of HNO3.IO3- and H2O.IO3- clusters doesn't indicate IA nucleated. Were larger IA clusters seen? Also did the authors calculate how much IA contributes to growth? Is that why the authors are implicitly linking high concentrations of IA to particle growth in line 564?

**Author's response:** Yes the temperature throughout the campaign was higher than 10°C.

The recent work of Xiao et al (2021), demonstrates that in the urban atmosphere NPF is mainly driven by the formation of sulfuric acid–base clusters, which are stabilized by the presence of amines, high ammonia concentrations and lower temperatures. Figure 1 from the work of Xiao et al., 2021 clearly shows that at an SA concentration of $10^6$ molec.cm$^{-3}$ or higher, the nucleation rates ($J_{1.7}$) was mostly > 1 cm$^3$ s$^{-1}$ at 293 K (magenta) in the presence of DMA (4pptv DMA injection).

[Figure]

Figure 1: Atmospheric nucleation rates( $J_{1.7}$) versus SA concentrations (Xiao et al., 2021). Note that this figure is only for providing clarity in the Author's response, it has not been included in the manuscript.

Our study reports that the atmospheric nucleation rates ($J_{1.5}$) was mostly below 1 cm$^3$ s$^{-1}$ at SA concentrations $10^6$ molec.cm$^{-3}$ (Please refer to Fig. 5 in the MS). In such a case, we can speculate that the NPF may be to some extent driven by SA-DMA (Also HOMs) system. However, in the NPF case on 11$^{th}$ August 2019, where IA concentrations clearly increase ($10^7$ molec. cm$^{-3}$ ) over the SA concentrations (note the SA concentration remains similar to that observed during the event on 30 June 2019, i.e $10^6$ molec.cm$^{-3}$), we see nucleation rates ($J_{1.5}$) clearly increasing above 1 cm$^{-3}$ s$^{-1}$ (Figure 7), suggesting IA could be an important contributor to this NPF event.

As mentioned in the manuscript text (Lines 607-612) *"This was the highest observed IA concentration in the entire measurement period. A recent study by He et al., 2021, indicate that HIO$_3$ concentrations above $1 \times 10^7$ molec. cm$^{-3}$ leads to rapid new particle formation at +10° C. At such concentrations the efficacy of iodine oxoacids to form new particles exceeds that of the H$_2$SO$_4$-NH$_3$ system at the same acid concentrations. Thus, the concentration of IA found in this event is capable of initiating nucleation, especially since the concentration of IA being two times higher than SA during the start of the event",* we speculate that this was an IA-driven event given the amount of SA remains unchanged! Further, higher temperature in principle should reduce the formation rates for all systems (SA+amine/ SA+NH$_3$/Iodine oxoacids). However, SA-DMA nucleation is in general much faster than the other two systems (SA+NH$_3$/iodine oxoacids) and if significant amount of DMA is

present the nucleation rates should be significantly higher than current values given that we do not have very high SA concentrations ($>3 \times 10^7$ molec. cm$^{-3}$) in the study area. (Yao et al., 2018)

No larger clusters were seen in the CI-ApiTOF measurements. Since to actually detect the large clusters the ambient IA concentrations should be close to or higher than $10^8$ (Sipilä et al., 2016).

We calculated the IA GR for the event 30 June and 30 July through the parameterization methods used in Nieminen et al., 2010. This method was also very recently used to calculate the IA growth rates in the recent work of Beck et al., 2021. However since it was an intense burst event with no proper horizontal growth (as seen in "banana" type events) we were not able to calculate the growth rate for this particular event. We speculate it to be an IA driven event due to high concentration of IA seen during the event capable of initiating the ion clustering (He et al., 2021), observation of increase of normalized signals of IA-clusters (Fig S7) and all the above reasons mentioned above why SA may not be the main precursor gas initiating the NPF.

**Author's changes in manuscript:**

L711-714, Pg:26, "Thus, the concentration of IA found in this event (two times higher than SA during the start of the event), the high formation rates ($>1$ cm$^{-3}$ s$^{-1}$) and an unchanged concentration of SA during the event, as compared to the event on 30 June 2019, strongly suggests that it could be an IA driven-NPF event"

L725-727, Pg: 27, "Since it was an intense burst event with no proper horizontal growth (as seen in "banana" type events), we were not able to calculate the growth rate for this particular event. Therefore we are unable to quantify the contribution of IA towards the growth of particles reaching CCN sizes".

**Comment from Referee:** Line 567 (And figures 5,7,8): The authors comment that the Aitken mode particle concentration increase after a new particle formation event. Why does the concentration drop before+during a new particle formation event? No doubt the decrease in scavenging rates allows nucleation to occur but what is leading to this drop in large particle concentration? This seems just as important as an increase in precursor concentration in producing an event.

**Author's response:** We thank the reviewer for pointing this important observation in the manuscript. In all the cases (Fig 5, 7,8) there is a drop in both accumulation and Aitken accumulation mode particles before NPF. We also see a change in wind direction before start of NPF in all the three cases.

Previous study of Vakeva et al., 2000 also suggests that wind direction changes led to decrease in particle concentrations and also change in particle size distributions particularly in urban areas and was considered as the most important factor-affecting particle concentrations. Also, if we carefully examine, during the NPF the accumulation mode particles showed an increase in all the cases, except the one out of two NPFs on 30 June 2019. This was because unlike the other two cases (11 Aug and 15 Aug) a stable wind was observed during NPF on 30 June. Also, its worth noting that cloudiness parameter also affected the nucleation on 11 August, since it was an overall cloudy day with few hours of clear sky conditions (which is already described in the manuscript). Hence, we can say that there is not one meteorological parameter affecting the start of NPF and determining whether the NPF would lead to growth of particles or not. And this observation is consistent with various other NPF studies.

**Author's changes in manuscript:** Case 30[th] June: we included the lines regarding the change in particle concentration as follows:

L624-632, Pgs: 23-24," However, we also observe a drop in Aitken particles before NPF which also continues during NPF. We speculate it could be due to the change in wind direction (Vakeva et al., 2000) before NPF. The wind direction relatively remains constant throughout the NPF so the low concentration of Aitken mode continues. Wind direction changes abruptly at 12:00h and the Aitken mode particle concentrations increases soon after this change of wind direction (Fig. 5d). This shows the particles must be the process of growth mostly elsewhere, which is not evident in the changed air mass, however we still observe almost the same (or even slightly higher) precursor vapor concentrations, since the wind still passed over the bloom areas before entering our study site."

Considering this an important observation (and also as per the reviewer's helpful comment) we have included this a statement "Further we also infer that that the wind direction played an important role in determining the particle concentrations at the study site" in the conclusion section of this study.

Minor comments:

**Comment from Referee**: Line 154-160: It would be less confusing if the instrument details in this paragraph moved to the instrument 2.1. Otherwise, the reader will want more details about the instruments before the actual instrument section.

**Author's response:** We accept the reviewer's suggestion and have incorporated the suggested changes in the revised MS.

**Author's changes in manuscript:** L233-L239, Pg:8, The lines about the instrument and their installation sites have been incorporated in the section 2.1 "To understand the chemical composition of the precursor vapors emitted from various sources around the site, the Chemical ionization Atmospheric Pressure interface-Time Of Flight mass spectrometer (CI-APiTOF) was operated from the 4th floor laboratory of the Physicum building, Kumpula campus, University of Helsinki (60° 12' N, 24° 58' E ; 49m , a.m.sl). The other aerosol and trace gases instruments were operated at the SMEAR III station which is 180 m away from the mass spectrometric measurement site (Station for Measuring Ecosystem-Atmosphere Relation (SMEAR III), 60.20° N, 24.96° E; 25 m a.s.l.).

**Comment from Referee:** Line 170: the paper hypothesis has already been stated in introduction. No need to state it again here.

**Author's response:** We have deleted the lines stating the hypothesis. Instead a shorter statement has been incorporated.

**Author's changes in manuscript:** L207-L209, Pg:7, "The site and measurement period (25 June 2019–18 August 2019) selected for this particular study are unique since this semi-urban location could be influenced by emissions from the recurring summertime blooms in the Baltic sea and the neighboring coastal regions.

**Comment from Referee:** Paragraph beginning 169: It would be more useful if this paragraph focuses instead on presenting the date+times of the algae and cyanobacteria blooms during the measurement campaign. The background on why there are more blooms should be mentioned in the introduction instead to better motivate this study.

**Author's response:** We accept reviewer's suggestion and have modified the above mentioned paragraph with the following changes

**Author's changes in manuscript:**

L217-229, Pg:7, are deleted: "Increasing temperatures and the excessive nutrient load in the Baltic Sea promote algal growth (Kuosa et al., 2017; Suikkanen et al., 2007, 2013). According to HELCOM (Baltic Marine Environment Protection Commission), the Baltic Sea has warmed 0.3° C per decade, however after 1990 significantly faster at 0.6° C per decade and in Finnish coastal areas the warming is even faster with a 2° C increase since 1990 (Humborg et al. 2019). The amount of blue-green algae (i.e. cyanobacteria) has shown a statistically significant increase in open sea areas in the Gulf of

Finland, Sea of Åland and the Sea of Bothnia in the last 40 years (Kahru and Elmgren, 2014). Although nutrient pollution has showed a decreasing trend (Andersen et al., 2017), growing oxygen deficient waters recirculate nutrients and perpetuate cyanobacterial blooms (Funkey et al., 2014). The increase in frequency and intensity of cyanobacterial blooms would increase the potential emission of biogenic gases changing the composition of the overlying atmosphere and the atmosphere of the neighboring sites, depending on the meteorological conditions.

L212-218, Pg:7, These lines are incorporated in this section "As per the SYKE press release (2019) the northern part of the Baltic Sea's main basin, entrance to the Gulf of Finland and south of the Åland Islands, were enriched with blue-green algae (cyanobacteria) .The bloom lasted from June-August 2019. In coastal areas, bloom was mostly spotted in the Archipelago Sea, Gulf of Finland, Bothnian Sea and the Quark. The bloom situation developed rapidly and spatially highly variable, even over short distances. The fragmented nature of the coastal areas and changing wind and water currents makes the algal bloom conditions highly dynamic".

**Comment from Referee:**Line 189: What type of inlet?

**Author's response:** We used a nitrate based-chemical ionization (CI) inlet. The inlet design (also called Eisele inlet), as described by Eisele and Tanner (1993) and Kurten et al. (2011) and further used by Jokinen et al., 2012.

**Author's changes in manuscript:** We have added this text in the L244-245, Pg:8, "In our study we used inlet design as described by Eisele and Tanner (1993) and Kurten et al. (2011) and further used by Jokinen et al., 2012.".

**Comment from Referee:** Line 195: Was the only reagent ion NO3-? Or did it have ligands?

**Author's response:**$NO_3^-$ was the most abundant reagent ion. However its dimer (($HNO_3$)$NO_3^-$) and trimer (($HNO_3$)$_2NO_3^-$) is also found in the spectrum. All the concentrations are normalized against the most dominant reagent ions which are estimated as $NO_3^-$ + ($HNO_3$)$NO_3^-$.

**Author's changes in manuscript:** The statements in the MS are modified as follows L257-L259 Pg:9, "SA, MSA, IA concentrations are calculated after normalizing them with the reagent ions ($NO_3^-$ and ($HNO_3$)$NO_3$) using the equation mentioned in Jokinen et al., 2012"

**Comment from Referee:** Line 196: is mlpm milliters per min? The L should be capitalized to make it less confusing. Or ccm which is more commonly used? Or maybe mlpm is fine? But it was initially confusing to me.

**Author's response:** Yes mlpm= mililitres per minute. The units are now written as "mLpm" in the MS.

**Author's changes in manuscript:** L249-252, Pg:8 "In this study, the chemical ionization was done via nitrate ions ($NO_3^-$) through X-ray exposure of nitric acid ($HNO_3$, flow rate: 3 mLpm), saturating the sheath air flow entering the CI (flow rate: 30 Lpm), the inlet flow of 10 Lpm was reached by using a 40 Lpm total flow"

**Comment from Referee:** Line 208: There is a random The at the end of the line

**Author's response:** It is deleted now. Thanks for pointing that out.

**Author's changes in manuscript:** L270, Pg:9, "Please note that the concentration of highly oxygenated molecules (HOM monomers and dimers) were calculated from the unit mass resolution data".

**Comment from Referee:** Line 213: What do the authors mean by two identical DMAs? Have they quantified the transfer functions and transmission efficiencies for both to say they are identical?

**Author's response:** In principle the DMAs are built identical but in the instrument the applied voltages are of opposite sign. But we agree with the reviewer's comments that they are not perfectly identical since as per our knowledge all DMAs are individually characterized (by the manufacturer, in this case Airel) and the electrometers can have different sensitivies, so this results in each DMA having their own transfer function after calibration. So the statement about identical measurement columns differing in polarity is a statement about principle of operation, not a result that was obtained from calibration.

**Author's changes in manuscript:** Therefore we have modified the statement in the revised MS (section 2.1) as follows:
L278-L283, Pg:9, "NAIS consists of two multichannel electrical mobility analyzer columns (DMA's) operating in parallel. The columns differ by the polarity of the ions measured, but are otherwise identical (Mirme and Mirme, 2013) in operation. However they may differ in the transfer functions

after calibration.The calibration procedure for the DMAs is presented in Mirme and Mirme, 2013. The ion mode measurements are corrected as in Wagner et al.,2016)".

**Comment from Referee:** Line 280: Are these mobilities diameters?

**Author's response:** Yes it means the mobility diameter and the term "mobility" has been incorporated. This section has now been moved to the supplementary information.

**Author's changes in manuscript (SI):** L92-93, Pg:4,"The formation rate of the total particles of mobility diameter 1.5 nm is calculated using the time derivative of the particle number concentration measured using the PSM in the size range 1.5– 3 nm."

**Comment from Referee:** Section 2.5: Condensation sink spans what particle diameters? (I think it's >6 nm). Is there a reason why CS does not include surface area of smaller particles which could be significant during a new particle formation event?

**Author's response:** We thank the reviewer for their comment, we calculated the condensation sink using the equation mentioned in (*Pirjola et al.*, 1999) which uses the Kn (Knudsen number), which is also dependent on the diffusion coefficient and the average absolute velocity of the vapor molecules (Hirshfelder et al., 1954). The equation used for this calculation assumes that condensable vapor does not take part in nucleation (Maso et al., 2002).

Further to check, whether nucleation mode particles act as condensation sink or not, we calculated the condensation sink from the DMPS system which includes the diameters <3 nm, The results are shown in figure below:

[Figure]

Figure S1 NPF Event- 30 June 2019 (a) Number size distribution of particles (data from PSM, NAIS and DMPS; size range: sub-3–100nm) (b) Particle Size dependent condensation sink variability during the most intense hour (12:00 h) of the event.

The Fig. S1 clearly shows that the size distribution of the CS for an example day (30 June 2019) with a strong nucleation event and a selection during the most intense hour (12:00h). We find that most of the CS is concentrated in the Aitken and accumulation mode rather than the nucleation mode.

Although the concentration of the smallest particles is substantially higher during an NPF event, we find that nucleation mode particles do not provide enough surface area to compete with the larger particles in terms of condensation.

**Author's changes in manuscript:** We have now completely removed the section 2.5 from the revised manuscript as the CS was not adding any extra information or was not aiding in the interpretation of any major observations. This deletion would also help in providing better clarity in the approach of the interpretations and would also shorten the length of the manuscript.

Also please note there was a typo error in L285-286, Pg:9, The DMPS data is considered from particles starting from 3nm and above and not 6nm and above. We have corrected that as follows:

"Larger particles of 3–820 nm were measured using a twin differential mobility particle sizer (DMPS) (Aalto et al., 2001)".

**Comment from Referee:** Line 340: is open sea microalgae cyanobacteria? Can the authors more clearly show/explain what time periods were for coastal macroalgae and blue green algae? Did they these bloom/exposure events overlap? If so, to what extent?

**Author's response:** The bloom is mostly the microalgae cyanobacteria with a mix of macroalgae which are mainly exposed during the receding tides and are speculated to emit biogenic gases when the they start to decay during the ending phase of cyanobacterial bloom (mid-August 2019: ref, SYKE press release 2019 and Kowanka et al., 2020). At this point we cannot say precisely that when (or if at all) the bloom or exposure events overlap, considering that the bloom was widespread in the Northern Baltic sea, Gulf of Finland, Gulf of Riga and other coastal areas of Finland.

**Author's changes in manuscript:** L415-417, Pg:13, "During these conditions, contributors to emissions might be a mix of both coastal macroalgae and open sea microalgae, which are mostly the cyanobacteria.

**Comment from Referee:** Line 406: For surface emissions to be carried to the measurement site, the surface wind speed is important. Is this wind velocity at the surface/altitude of the measurement site? Or does it include a component of vertical velocity? In other words, how confident are the authors that the air mass is not vertically mixing downwards which would dilute the surface emissions?

**Author's response:** The wind velocity is measured at almost the same altitude as the measurement site, so we assume that the air mass is not vertically mixing downwards. The wind data was taken from the wind vane installed at the roof of Physicum building (roof of 5[th] floor) and our CI-APiTOF measuring the gaseous precursors was situated on the floor just below the roof top with the inlet sticking out of the 4[th] floor window. However, as stated in the MS also that the measurements for particle size distributions was carried out at SMEAR III, which is 25 m a.m.sl and the wind vane at the Physicum building was situated roughly at 50 m a.m.s.l. So we agree that the particle size distribution data might not be completely free from downward vertical mixing of airmass and this we accept as the limitation of this study.

**Author's changes in manuscript: L49-57, Pg:2 in SI**

"All the meteorological parameters are measured by sensors installed on the roof of the physicum building (where CI-APi-TOF was housed). Thus we can say that the precursor vapor concentrations measured by the CI-APITOF was not influenced by any vertical mixing of airmasses since the sensors for meterological parameters (installed on the roof of 5th floor, physicum building and CI-APITOF (installed on the 4th floor, physicum building) was almost at the same height. However, the measurements for particle size distributions was carried out at SMEAR III, which is 25 m a.m.sl and the wind vane at the Physicum building was situated roughly at 50 m a.m.s.l. ,we state that the particle size distribution data might not be completely free from downward vertical mixing of airmass and should be treated with certain uncertainty. However, near the SMEAR III station, the mixing usually affected the larger particles, decreasing their number concentration (Järvi et al., 2009)". So we can assume that the uncertainties in the number concentration of nucleation and Aitken mode particles would be negligible in this study.

**Comment from Referee:** Line 424: MSA also originates from agriculture.

**Author's response:** Yes, we agree that apart from marine sources MSA sourced from DMS can be emitted from agricultural practices and also biomass burning. However, in the global scenario the Sulphur emissions (considering DMS to be the most important Sulphur source) ocean contributes to 19% while the terrestrial emissions account only for 0.4% and out of this agriculture accounts for 2.7% of the sulphur emissions in the latitudes 65°N-50°N (Bates et al., 1992).

**Author's changes in manuscript:** We have included "agriculture" in the lines mentioned in the manuscript. Lines 520-521, Pg:18 "However some emissions could be sourced from agriculture and other terrestrial sources, Bates et al., 1992"

**Comment from Referee:** Line 459: could a burst in sub-3 nm particles be from a suppression in growth and not a local nucleation event?

**Author's response:** We agree with the reviewer that in these local clustering there is no growth of clusters beyond a certain diameter. And we follow the NPF classification of mainly Lubna et al., 2018 which clearly says "The type IB, or ion bursts, is an attempt at NPF, during which clusters form in Hyytiälä; however, they do not grow beyond a few nanometres in diameter". So yes they are kind of ion bursts which call "local clustering" in this work following the description of Lubna et al., 2018.

**Author's changes in manuscript:** For clarity we removed the word "NPF" from the statement. Line556, Pg: 19, "These so-called bursts /spikes appearing at small sizes (sub-3 nm) are indicative

of local clustering processes in contrast to regional events, where it is possible to follow the growing particle mode for several hours (Dada et al., 2018; Dal Maso et al., 2005)".

**Figure comments:**

**Comment from Referee:** Figure 2: Is this figure necessary? The manuscript only details specific events that occurred in short periods of time. It would be more helpful to see this data with the event data.

**Author's response:** We feel it's important to describe the overall meteorological conditions during the study period since it helps the readers to get an understanding that what kind of environment the authors are talking about (hot/humid/windy etc) before they get to read about the new particle formation processes in the atmosphere of the this study site. Therefore we would like to keep the Figure 2 in the main text.

We have included the wind data along with the event data. We tried to club other meteorological parameters like wind speed with the event data, however the event figures looked crowded. Hence, we decided to include only the most important meterological parameter in the event figure.

**Author's changes in manuscript:** no changes made, Figure 2 is still in the main text of the MS.

**Comment from Referee:** Figure 5,7,8 (and their siblings in the SI) are very difficult to read. The font on the labels is too small to read. It might be easier to have the timeline graphs vertically stacked so it's easier to compare between them. The F panel is strange. Are the maps of the same area? It doesn't seem like it. Why have two panels for F?

**Author's response:** We accept reviewer's suggestion regarding Figures 5, 7 and 8 and the similar figures in the SI. Regarding the two figures in Panel F: one data(map) is for Chla concentrations taken from GlobColour level-3 as a separate Map, whereas the other figure is plotted through HYSPLIT and data taken from GDAS. Yes, the graph panel "f" is of the same area but we agree that the zoom percentage is different in the two maps. Superimposing both the maps may not be possible. However, we have tried to make the Chla map of the same zoom-level as the trajectory map for a better comparison

**Author's changes in manuscript:** The font size has been increased for all the panels in the figures (5,7and 8). The timeline graphs are stacked for all events in the MS as an example Figure 5 is shown

below. Panel F in all figures is made as a separate figure. Accordingly all figure numbers are changed in the entire MS. The Chla map is corrected for the same zoom-level as the trajectory map for a better comparison (as an example Figure 6a and b is shown below).

[Figure]

Figure 5**:** NPF Event- 30 June 2019 (a) Number size distribution of particles (data from PSM, NAIS and DMPS; size range: sub-3–100nm). (b) Charged particles number size distribution (negative: upper, positive: lower) obtained from the NAIS. (c) Diurnal variation of formation rates ($J_{1.5}$) of 1.5 nm particles and ions ($J^-_{1.5}$ and $J^+_{1.5}$) on the left axis and particle number concentrations (1.5–3 nm) on the right axis. (d) Diurnal variation of HOMs SA, IA and MSA with wind direction (WD). (e) The diurnal variation of particle concentration in nucleation:3–20 nm; aitken: 25–100 nm and accumulation: >100nm) mode particles during the event (Data from DMPS).

[Figure]

Figure 6: (a) Trajectory frequency plot (100 a.g.l, arrival time of trajectories at the meaurement site: 20:00 h) for 24 h back trajectory using GDAS meterological input data (frequency grid resolution: $1.0° \times 1.0°$) (b) Chl-*a* concentrations (GlobColour level-3); Black line shows the trajectory direction and the star point denotes the measurement site.

**Comment from Referee:** Figure S3: The labels are too small to read. Units of residence time?

**Author's response:** We agree with the reviewer's suggestion that the font size is small for the figures. We have made the required changes in all the flexpart figures showing the residence times of the airmasses. The residence time has no units therefore its not mentioned in the figures. Although its been already also mentioned this in the section 2.2 of the MS

"The residence times were normalized for clarity in the all the figures and is shown on a scale of 0 to 1 (Results are included in the supplementary information)"

**Author's changes in manuscript:** The font size has been increased for all the flexpart figures in the SI (Fig.S3, Fig.S7 and Fig.S9). Since the residence time is normalized so it has no units, for clarity

the term "normalized residence time" has been included in the figures. As an example Fig.S3 is shown below.

[Figure]

Figure S3 : Normalized Residence times of air masses (3-day backwards) arriving at the experimental site on 30 June 2019. The color bar indicates the normalized residence times for each subplot. The residence time of particles originating 3 days before reaching SMEAR III is shown for 6:00 h, 9:00 h, 12:00 h and 15:00 h. The red shaded areas indicate the latitude/longitude pairs having the maximum residence time.

**References:**

Aalto, P., Hämeri, K., Becker, E., Weber, R., Salm, J., Mäkelä, J. M., Hoell, C., O'Dowd, C. D., Karlsson, H., Hansson, H. C., Väkevä, M., Koponen, I. K., Buzorius, G., and Kulmala, M.: Physical Characterization of Aerosol Particles during Nucleation Events. *Tellus B*, 53:344–358, https://doi.org/10.3402/tellusb.v53i4.17127, 2001

Bates, T.S., Lamb, B.K., Guenther, A., Dignon, J. and Stoiber, R.E.: Sulfur emissions to the atmosphere from natural sources., J Atmos Chem., 14**,** 315–337, https://doi.org/10.1007/BF00115242, 1992.

Bergman, T., Laaksonen, A., Korhonen, H., Malila, J., Dunne, E. M., Mielonen, T., Lehtinen, K. E. J., Kühn, T., Arola, A. and  Kokkola, H.: Geographical and diurnal features of amine-enhanced boundary layer nucleation, J. Geophys. Res. Atmos., 120, 9606–9624, doi:10.1002/ 2015JD023181, 2015.

Carbone, M. S., Park Williams, A., Ambrose, A. R., Boot, C. M., Bradley, E. S., Dawson, T. E., Schaeffer, S. M., Schimel, J. P. and Still, C. J:  Cloud shading and fog drip influence the metabolism of a coastal pine ecosystem Global Change Biol., 19, 484–97, 2013.

Crump, J.G. and Seinfeld, J.H.: Turbulent deposition and gravitational sedimentation of an aerosol in a vessel of arbitrary shape, J.Aerosol Sci., 12, 405-415.

Ehn, M., Thornton, J. A., Kleist, E., Sipilä, M., Junninen, H., Pullinen, I., Springer, M., Rubach, F., Tillmann, R., Lee, B., Lopez-Hilfiker, F., Andres, S., Acir, I.-H. H., Rissanen, M., Jokinen, T., Schobesberger, S., Kangasluoma, J., Kontkanen, J., Nieminen, T., Kurtén, T., Nielsen, L. B., Jørgensen, S., Kjaergaard, H. G., Canagaratna, M., Maso, M. D., Berndt, T., Petäjä, T., Wahner, A., Kerminen, V.-M. M.,  Kulmala, M., Worsnop, D. R., Wildt, J. and Mentel, T. F.: A large source of low-volatility secondary organic aerosol, Nature, 506(7489), 476–479, doi:10.1038/nature13032, 2014.

Eisele, F. L. and Tanner, D. J.: Measurement of the gas phase concentration of $H_2SO_4$ and methane sulfonic acid and estimates of $H_2SO_4$ production and loss in the atmosphere, J. Geophys. Res.,98, 9001–9010, 1993.

Emery, N. C., D'Antonio, C .M. and Still, C. J.: Fog and live fuel moisture in coastal California shrublands Ecosphere, 9, e02167, https://doi.org/10.1002/ecs2.2167 ,2018.

Hemmilä, M., Hellén, H., Virkkula, A., Makkonen, U., Praplan, A. P., Kontkanen, J., Ahonen, L., Kulmala, M., and Hakola, H.: Amines in boreal forest air at SMEAR II station in Finland, Atmos. Chem. Phys., 18, 6367–6380, https://doi.org/10.5194/acp-18-6367-2018, 2018.

Hirschfelder, J. 0., Curtiss, C. F., and Bird, R. B. : Molecular Theory of Gases and Liquids,John Wiley & Sons, New York, p 539, 1954.

Kurten, T., Petäjä, T., Smith, J., Ortega, I. K., Sipilä, M., Junninen, H., Ehn, M., Vehkamäki, H., Mauldin, L., Worsnop, D. R. and Kulmala, M.: The effect of $H_2SO_4$ – amine clustering on chemical ionization mass spectrometry (CIMS) measurements of gas-phase sulfuric acid, Atmos. Chem. Phys., 11, 3007–3019, doi:10.5194/acp-11-3007-2011, 2011.

Lawson, D. M., Clemesha, R.E.S., Vanderplank, S., Gershunov, A. and Cayan, D.: Impacts and influences of coastal low clouds and fog on biodiversity in San Diego California's Fourth Climate Change Assessment CCCA4-EXT-2018-010 69–89, https://www.energy.ca.gov/sites/default/files/2019-12/Biodiversity_CCCA4-EXT-2018-010_ada_0.pdf, 2018.

Lehtinen, K. E. J. and Kulmala, M.: A model for particle formation and growth in the atmosphere with molecular resolution in size, Atmos. Chem. Phys., 3, 251–257, https://doi.org/10.5194/acp-3-251-2003, 2003.

Makkonen, U., Virkkula, A., Hellén, H., Hemmilä, M., Sund, J., Äijälä,M., Ehn, M., Junninen,H., Keronen,P., Petäjä,P., Worsnop,D.R., Kulmala, M. and Hakola, H: Semi-continuous gas and inorganic aerosol measurements at a boreal forest site: seasonal and diurnal cycles of $NH_3$ , HONO and $HNO_3$., Boreal Environment Research 19 (suppl. B), 311–328, ISSN 1797-2469, 2014.

Makkonen, U., Virkkula, A., Mäntykenttä, J., Hakola, H., Keronen, P., Vakkari, V., and Aalto, P. P.: Semi-continuous gas and inorganic aerosol measurements at a Finnish urban site: comparisons with filters, nitrogen in aerosol and gas phases, and aerosol acidity, Atmos. Chem. Phys., 12, 5617–5631, https://doi.org/10.5194/acp-12-5617-2012, 2012.

Mirme , S. and Mirme, A.: The mathematical principles and design of the NAIS – a spectrometer for the measurement of cluster ion and nanometer aerosol size distribution, Atmos. Meas. Tech., 6, 1061–1071, https://doi.org/10.5194/amt-6-1061-2013, 2013.

Ortega, I. K., Olenius, T., Kupiainen-Määttä, O., Loukonen, V., Kurtén, T., and Vehkamäki, H.: Electrical charging changes the composition of sulfuric acid–ammonia/dimethylamine clusters, Atmos. Chem. Phys., 14, 7995–8007, https://doi.org/10.5194/acp-14-7995-2014, 2014.

Rissanen, M. P., Mikkilä, J., Iyer, S., and Hakala, J.: Multi-scheme chemical ionization inlet (MION) for fast switching of reagent ion chemistry in atmospheric pressure chemical ionization mass spectrometry (CIMS) applications, Atmos. Meas. Tech., 12, 6635–6646, https://doi.org/10.5194/amt-12-6635-2019, 2019.

Schade, G. W. and P. J. Crutzen.: Emission of aliphatic-amines from animal husbandry and their reactions: Potential source of N$_2$O and HCN, *J. Atmos. Chem.*, **22**(3), 319– 346, doi:10.1007/BF00696641, 1995.

Sipilä, M., Sarnela, N., Jokinen, T., Henschel, H., Junninen, H., Kontkanen, J., Richters, S., Kangasluoma, J., Franchin, A., Peräkylä, O., Rissanen, M. P., Ehn, M., Vehkamäki, H., Kurten, T., Berndt, T., Petäjä, T., Worsnop, D., Ceburnis, D., Kerminen, V.-M. M., Kulmala, M., O'Dowd, C. and O'Dowd, C.: Molecular scale evidence of aerosol particle formation via sequential addition of HIO3, Nature, 537(7621), 532–534, doi:10.1038/nature19314, 2016.

Sipilä, M., Sarnela, N., Jokinen, T., Junninen, H., Hakala, J., Rissanen, M. P., Praplan, A., Simon, M., Kürten, A., Bianchi, F., Dommen, J., Curtius, J., Petäjä, T., and Worsnop, D. R.: Bisulfate – cluster based atmospheric pressure chemical ionization mass spectrometer for high-sensitivity (< 100 ppqV) detection of atmospheric dimethyl amine: proof-of-concept and first ambient data from boreal forest, Atmos. Meas. Tech., 8, 4001–4011, https://doi.org/10.5194/amt-8-4001-2015, 2015.

SYKE press release (29 August 2019), Summary of algal bloom monitoring June-August 2019: Cyanobacteria were mostly mixed in the water in the Finnish sea areas, in lakes the cyanobacteria situation varied a lot.
https://www.syke.fi/enUS/Current/Press_releases/Summary_of_algal_bloom_monitoring_JuneAu(5 1391).

Wagner, R., Manninen, H.E., Franchin, A., Lehtipalo, K., Mirme, S., Steiner, G., Petäjä, T., Kulmala, M.: On the accuracy of ion measurements using a Neutral cluster and Air Ion Spectrometer, Boreal Environ. Res., 21, pp. 230-241, 2016

Wang, M., He, X.-C., Finkenzeller, H., Iyer, S., Chen, D., Shen, J., Simon, M., Hofbauer, V., Kirkby, J., Curtius, J., Maier, N., Kurtén, T., Worsnop, D. R., Kulmala, M., Rissanen, M., Volkamer, R., Tham, Y. J., Donahue, N. M., and Sipilä, M.: Measurement of iodine species and sulfuric acid using bromide chemical ionization mass spectrometers, Atmos. Meas. Tech., 14, 4187–4202, https://doi.org/10.5194/amt-14-4187-2021, 2021.

Xiao, M., Hoyle, C. R., Dada, L., Stolzenburg, D., Kürten, A., Wang, M., Lamkaddam, H., Garmash, O., Mentler, B., Molteni, U., Baccarini, A., Simon, M., He, X.-C., Lehtipalo, K., Ahonen, L. R., Baalbaki, R., Bauer, P. S., Beck, L., Bell, D., Bianchi, F., Brilke, S., Chen, D., Chiu, R., Dias,

A., Duplissy, J., Finkenzeller, H., Gordon, H., Hofbauer, V., Kim, C., Koenig, T. K., Lampilahti, J., Lee, C. P., Li, Z., Mai, H., Makhmutov, V., Manninen, H. E., Marten, R., Mathot, S., Mauldin, R. L., Nie, W., Onnela, A., Partoll, E., Petäjä, T., Pfeifer, J., Pospisilova, V., Quéléver, L. L. J., Rissanen, M., Schobesberger, S., Schuchmann, S., Stozhkov, Y., Tauber, C., Tham, Y. J., Tomé, A., Vazquez-Pufleau, M., Wagner, A. C., Wagner, R., Wang, Y., Weitz, L., Wimmer, D., Wu, Y., Yan, C., Ye, P., Ye, Q., Zha, Q., Zhou, X., Amorim, A., Carslaw, K., Curtius, J., Hansel, A., Volkamer, R., Winkler, P. M., Flagan, R. C., Kulmala, M., Worsnop, D. R., Kirkby, J., Donahue, N. M., Baltensperger, U., El Haddad, I., and Dommen, J.: The driving factors of new particle formation and growth in the polluted boundary layer, Atmos. Chem. Phys., 21, 14275–14291, https://doi.org/10.5194/acp-21-14275-2021, 2021.